# Towards Generalization Bounds of GCNs for Adversarially Robust Node Classification

**Wen Wen**[1]**, Han Li**[1,2]*** Tieliang Gong**[4]**, Hong Chen**[1,2,3]

[1]College of Informatics, Huazhong Agricultural University
[2]Engineering Research Center of Intelligent Technology for Agriculture, Ministry of Education
[3]Shenzhen Institute of Nutrition and Health, Huazhong Agricultural University
[4]School of Computer Science and Technology, Xi'an Jiaotong University
{wen190329,adidasgtl}@gmail.com, {lihan125,chenh}@mail.hzau.edu.cn

## Abstract

Adversarially robust generalization of Graph Convolutional Networks (GCNs) has garnered significant attention in various security-sensitive application areas, driven by intrinsic adversarial vulnerability. Albeit remarkable empirical advancement, theoretical understanding of the generalization behavior of GCNs subjected to adversarial attacks remains elusive. To make progress on the mystery, we establish unified high-probability generalization bounds for GCNs in the context of node classification, by leveraging adversarial Transductive Rademacher Complexity (TRC) and developing a novel contraction technique on graph convolution. Our bounds capture the interaction between generalization error and adversarial perturbations, revealing the importance of key quantities in mitigating the negative effects of perturbations, such as low-dimensional feature projection, perturbation-dependent norm regularization, normalized graph matrix, proper number of network layers, etc. Furthermore, we provide TRC-based bounds of popular GCNs with $\ell_r$-norm-additive perturbations for arbitrary $r \geq 1$. A comparison of theoretical results demonstrates that specific network architectures (e.g., residual connection) can help alleviate the cumulative effect of perturbations during the forward propagation of deep GCNs. Experimental results on benchmark datasets validate our theoretical findings.

## 1 Introduction

Node classification, which aims at predicting a particular class for each unlabeled node in an attributed graph given the class labels of a few nodes, has attracted tremendous attention due to its wide real-world applications (Zhou et al., 2019; Hang et al., 2021; Cao et al., 2021). As one of the predominant models for processing graph-structured data, Graph Convolutional Networks (GCNs) (Wu et al., 2020) have demonstrated superior prediction performance on node classification tasks. However, GCNs have been recently shown to be vulnerable to adversarial nodes, where the attacker injects imperceptible perturbations into node features, leading to incorrect predictions (Dai et al., 2018; Zügner et al., 2020; Ju et al., 2023). This has led to a proliferation of research aimed at enhancing the adversarial robustness of the trained models, built upon the min-max optimization principle of adversarial training (Madry et al., 2018; Wang et al., 2019; Kong et al., 2022; Tao et al., 2023; Li et al., 2022). Despite the empirical success, theoretical aspects of adversarially robust generalization of GCNs are not well understood yet. In this paper, we make progress on this goal by developing the generalization analysis of GCNs under node attacks.

Adversarial generalization problems have been widely investigated in recent years via the lens of statistical learning theory, ranging from uniform convergence analysis associated with hypothesis space capacity (e.g., VC-dimension (Cullina et al., 2018; Attias et al., 2022), Rademacher complexity (Awasthi et al., 2020; Yin et al., 2019), covering numbers (Tu et al., 2019; Mustafa et al., 2022)) to algorithmic stability analysis (Xiao et al., 2022; Xing et al., 2021). However, all the aforementioned work is primarily confined to supervised learning with individual samples as input, with extension

---

*Corresponding author.

to the graph learning remaining unexplored to the best of our knowledge. The significant challenge in analyzing the adversarial generalization of GCNs is that the feature information of each node is aggregated from its neighbors through the 'message-passing' mechanism rather than being taken only from itself, leading to the interaction of perturbations between different nodes. The joint effect of different perturbations in a message-passing network invalidates the classic estimation methods in (Yin et al., 2019; Awasthi et al., 2020; Mustafa et al., 2022), resulting in the analytical intractability of the adversarial loss over graph-structured data. Furthermore, the learning approach for node classification no longer corresponds to the supervised learning setting, where the nodes to be predicted are unlabeled and available during training (Li et al., 2018; Oono & Suzuki, 2020; Song et al., 2022). This paradigm is typically formulated within a transductive learning framework.

To overcome these obstacles, we derive strict upper bounds on the original adversarial loss, and then analyze the generalization properties of the surrogate by leveraging the novel contraction technique on graph convolution, which can yield tighter generalization guarantees. The main contributions of this paper are summarized as follows.

- We provide the high-probability generalization bounds of GCNs for adversarially robust binary and multi-class node classification tasks, through the lens of adversarial TRC. The derived bounds establish the connection between adversarial perturbations and generalization error, revealing the role of key factors (e.g., feature dimension, network architecture, graph matrix, etc.) in mitigating the negative impact of perturbations and improving the generalization ability. When the perturbation value is zero, we can recover the generalization bounds for the non-adversarial case, improving the dependence on the number of layers from exponential to the square root term compared to the existing TRC-based bounds.

- Our analysis enjoys broad applicability across a wide range of models and loss functions, necessitating only the characterization of adversarial TRC. As application examples of theoretical analysis, we provide explicit generalization bounds for popular GCN models, encompassing SGC, Residual GCN, and GCNII, demonstrating the importance of specific network architectures for achieving adversarially robust generalization of deep GCN models.

- Extensive experimental results on benchmark datasets demonstrate the effectiveness of our theoretical findings in reducing generalization error and achieving good generalization performance.

## 2 RELATED WORK

**Adversarial attack and defense on GCNs**  Recent research has shown that node features with carefully crafted perturbations can induce GCNs towards making incorrect predictions with high confidence (Dai et al., 2018; Zügner et al., 2020; Ma et al., 2020). To counter such attacks, various defense mechanisms have been developed to enhance the adversarial robustness, including regularizing the input gradient (Jia et al., 2023; Zhang et al., 2024), designing robust network architectures (Cisse et al., 2017; Abbahaddou et al., 2024), adversarial data augmentation (Suresh et al., 2021; Wu et al., 2022; Dong et al., 2024), and adversarial training (Feng et al., 2019; Li et al., 2022; Gosch et al., 2024). Among them, adversarial training has been validated to be the most effective defense strategy, which trains the robust model jointly with clean data and their adversarial counterparts. The resulting robust GCN models have been successfully applied in various fields (Sun et al., 2022). Despite the excellent performance, the generalization properties of GCNs under adversarial attacks are poorly understood.

**Generalization analysis of GCNs**  Scarselli et al. (2018) study the generalization ability of graph neural networks by leveraging the VC-dimension, which grows polynomially with the number of parameters and the number of nodes. Garg et al. (2020) establish the first data-dependent generalization bounds for message passing neural networks through the lens of the Rademacher complexity. Verma & Zhang (2019) develop stability-based generalization bounds and reveal the relationship between the graph size and algorithmic stability. Different from the above work under supervised learning settings, Esser et al. (2021) consider the semi-supervised graph learning setting and provide the generalization bounds by using the transductive Rademacher complexity. Deng et al. (2022) establish generalization guarantees for GCN-based recommendation models under inductive and transductive learning. Tang & Liu (2023) derive high probability bounds of generalization gap for popular graph

models in the transductive setting. Although the aforementioned work cannot be directly extended to adversarial settings due to the outer maximization w.r.t. adversarial perturbations, it provides valuable insights into the generalization analysis of GCNs under adversarial settings.

## 3  NOTATIONS AND PRELIMINARIES

**Notations.**  Let $[L] = \{1, \dots, L\}$. We denote vectors as lowercase bold letters (e.g., $\boldsymbol{w}$). The vector elements are denoted by lowercase letters (e.g., $\boldsymbol{w} = (w_1, \dots, w_n) \in \mathbb{R}^n$). We denote matrices by boldface uppercase letters (e.g., $\boldsymbol{W}$). For a matrix $\boldsymbol{W} \in \mathbb{R}^{m \times n}$, the $(p, q)$-norm is defined as $\|\boldsymbol{W}\|_{p,q} = \|(\|\boldsymbol{W}_{*1}\|_p, \dots, \|\boldsymbol{W}_{*n}\|_p)\|_q$, where $\boldsymbol{W}_{*i}$ is the column of $\boldsymbol{W}$. We use the shorthand notation $\|\cdot\|_p \equiv \|\cdot\|_{p,p}$, and write Hölder conjugates by a star (e.g., $r^*$).

**Preliminaries.**  Let $\mathcal{G} = (\boldsymbol{A}, \boldsymbol{X})$ be an attributed graph with $n$ nodes, where $\boldsymbol{X} = [\boldsymbol{x}_1, \dots, \boldsymbol{x}_n] \in \mathbb{R}^{n \times d}$ denotes the node feature matrix, and $\boldsymbol{A} \in \mathbb{R}^{n \times n}$ denotes the adjacency matrix. In this work, we focus on node classification tasks in a transductive manner, where the goal is to complete node labeling of the given graph with randomly sampled labels (Deng et al., 2022). Let $S = \{\boldsymbol{x}_i, y_i\}_{i=1}^n$ be the set of samples. Without loss of generality, we assume that the selected labels $y_1, \dots, y_m \in \mathbb{R}$ are known, and aim at finding the best predictor $f$ to predict the class labels $y_{m+1}, \dots, y_n$ by minimizing the training error $\mathcal{L}_m(f) := \frac{1}{m} \sum_{i=1}^m \ell(f(\boldsymbol{A}, \boldsymbol{X})_i, y_i)$, where $f(\cdot)_i \in \mathbb{R}$ represents the prediction of node $i$, and $\ell : \mathbb{R} \to \mathbb{R}_+$ denotes a given loss function. The test error is defined as $\mathcal{L}_u(f) := \frac{1}{n-m} \sum_{i=m+1}^n \ell(f(\boldsymbol{A}, \boldsymbol{X})_i, y_i)$. In the transductive learning setting, the training and test nodes are typically determined by a random partition (Ciano et al., 2021; Esser et al., 2021).

However, in the presence of adversaries, imperceptible perturbations on node features can deceive the model to make wrong predictions (Dai et al., 2018; Bojchevski & Günnemann, 2019). Following the previous empirical work (Sun et al., 2020; Jaeckle & Kumar, 2021), we assume that the set of adversarial nodes is generated from the neighborhood $\mathcal{B}_r^\varepsilon(\boldsymbol{X}) = \{\widetilde{\boldsymbol{X}} = [\widetilde{\boldsymbol{x}}_1, \dots, \widetilde{\boldsymbol{x}}_n] : \|\widetilde{\boldsymbol{x}}_i - \boldsymbol{x}_i\|_r \leq \varepsilon, r \geq 1, i \in [n]\}$, where $\varepsilon$ denotes the maximum perturbation bound. Given $\varepsilon > 0$, an attributed graph $(\boldsymbol{A}, \boldsymbol{X})$, the label $y_i$ of node $i$, and the loss function $\ell : \mathbb{R} \to \mathbb{R}_+$, the adversary selects the effective adversarial nodes $\widetilde{\boldsymbol{X}}_* = [\widetilde{\boldsymbol{x}}_{1*}, \dots, \widetilde{\boldsymbol{x}}_{n*}]$ by

$$\widetilde{\boldsymbol{X}}_* = \arg \max_{\widetilde{\boldsymbol{X}} \in \mathcal{B}_r^\varepsilon(\boldsymbol{X})} \ell(f(\boldsymbol{A}, \widetilde{\boldsymbol{X}})_i, y_i),$$

and the *adversarial loss* of $f$ at node $i$ is defined by

$$\widetilde{\ell}(f(\boldsymbol{A}, \boldsymbol{X})_i, y_i) := \max_{\widetilde{\boldsymbol{X}} \in \mathcal{B}_r^\varepsilon(\boldsymbol{X})} \ell(f(\boldsymbol{A}, \widetilde{\boldsymbol{X}})_i, y_i).$$

One of the popular defense methods against adversarial perturbations is adversarial training (Madry et al., 2018; Li et al., 2022), which aims to minimize the *adversarial training error*, i.e.,

$$\widetilde{\mathcal{L}}_m(f) := \frac{1}{m} \sum_{i=1}^m \widetilde{\ell}(f(\boldsymbol{A}, \boldsymbol{X})_i, y_i),$$

which measures the worst-case performance of the predictor under adversarial perturbations. We are interested in the generalization behavior measured by the *adversarial test error*, i.e.,

$$\widetilde{\mathcal{L}}_u(f) := \frac{1}{n-m} \sum_{i=m+1}^n \widetilde{\ell}(f(\boldsymbol{A}, \boldsymbol{X})_i, y_i).$$

We denote the generalization gap by $\mathrm{Gen}(f) = \widetilde{\mathcal{L}}_u(f) - \widetilde{\mathcal{L}}_m(f)$, which could serve as an indicator of the generalization performance of $f \in \mathcal{F}$ and often depends on the capability of the function class $\mathcal{F}$ (Oono & Suzuki, 2020; Deng et al., 2022). This paper introduces the Transductive Rademacher Complexity (TRC) (El-Yaniv & Pechony, 2009) to quantity the complexity of hypothesis classes for deriving the generalization bounds.

**Definition 3.1** (Transductive Rademacher Complexity). Let $\mathcal{F} \subseteq \mathbb{R}^n$, $p \in [0, 0.5]$, and $m$ the number of labeled samples. Let $\boldsymbol{\sigma} = (\sigma_1, \dots, \sigma_n)$ be a vector of i.i.d. random variables, where $\sigma_i$ takes

the value $+1$ or $-1$ with probability $p$, and $0$ with probability $1 - 2p$. Transductive Rademacher Complexity of $\mathcal{F}$ is defined as

$$\mathfrak{R}_{m,n}(\mathcal{F}) \triangleq \left(\frac{1}{m} + \frac{1}{n-m}\right) \mathbb{E}_{\boldsymbol{\sigma}} \left[\sup_{f \in \mathcal{F}} \boldsymbol{\sigma}^T f\right].$$

It is noteworthy that the TRC degenerates to the standard Rademacher complexity (Bartlett & Mendelson, 2002) if $p = 1/2$ and $m = n/2$. For $p < 1/2$, the TRC is beneficial to obtain tighter generalization bounds, where some Rademacher variables will reach zero value. This paper thus considers the probability $p$ of Rademacher variable $\sigma_i = \pm 1$ to be $\frac{m(n-m)}{n^2}$ (El-Yaniv & Pechyony, 2009).

We introduce the following classic result by directly applying Corollary 1 in (El-Yaniv & Pechyony, 2009) to adversarial settings, which shows that the generalization gap can be controlled by adversarial TRC, i.e., $\mathfrak{R}_{m,n}(\widetilde{\ell} \circ \mathcal{F})$.

**Lemma 3.2.** *Suppose that the range of the loss function $\ell$ is $[0, 1]$. Let $Q_1 \triangleq (\frac{1}{m} + \frac{1}{n-m})$, and $Q_2 \triangleq \frac{n}{(n-1/2)(1-1/(2\max(m,n-m)))}$. Then, with probability at least $1 - \delta$ for all $f \in \mathcal{F}$,*

$$\mathrm{Gen}(f) \leq \mathfrak{R}_{m,n}(\widetilde{\ell} \circ \mathcal{F}) + c_0 Q_1 \sqrt{\min(m, n-m)} + \sqrt{\frac{Q_1 Q_2}{2} \ln \frac{1}{\delta}},$$

*where $c_0 < 5.05$ is absolute constant.*

It is noteworthy that TRC-based bounds inherently exhibit monotonic decrease at a rate of $\mathcal{O}(\max\{\frac{1}{\sqrt{m}}, \frac{1}{\sqrt{n-m}}\})$ (El-Yaniv & Pechyony, 2009; Esser et al., 2021; Deng et al., 2022), reflecting the role of the number of labeled node $m$ on the generalization. With Lemma 3.2 as a toolkit, we can establish the generalization bounds for adversarial learning algorithms in the context of transductive inference, and the explicit characterization of $\mathfrak{R}_{m,n}(\widetilde{\ell} \circ \mathcal{F})$ for various models will be the focus of this paper. However, deriving an upper bound on $\mathfrak{R}_{m,n}(\widetilde{\ell} \circ \mathcal{F})$ is often intractable, due to the maximization operator of the adversarial loss over the graph-structured data via the message-passing network. Our approach is to derive a surrogate upper bound on the original adversarial loss, and establish a new risk bound in terms of the TRC of the surrogate by developing the novel usage of contraction inequality on graph convolution.

Let the hypothesis class of GCNs be defined as follows (Garg et al., 2020; Deng et al., 2022):

$$\mathcal{F} = \left\{ \boldsymbol{H}^{(L)} = \phi(g(\boldsymbol{A}) \cdots \phi(g(\boldsymbol{A}) \boldsymbol{X} \boldsymbol{W}^{(1)}) \cdots \boldsymbol{W}^{(L)}) : \|\boldsymbol{W}^{(l)}\|_2, \|\boldsymbol{W}^{(l)}\|_p \leq \omega, l \in [L] \right\}, \quad (1)$$

and the propagation procedure can be written as

$$\boldsymbol{H}^{(0)} = \boldsymbol{X}, \quad \boldsymbol{H}^{(l)} = \phi(g(\boldsymbol{A}) \boldsymbol{H}^{(l-1)} \boldsymbol{W}^{(l)}), \quad l \in [L] \quad (2)$$

where $\omega$ denotes the maximum bound over the $\| \cdot \|_2, \| \cdot \|_p$ of $\boldsymbol{W}^{(l)}$, $\boldsymbol{W}^{(l)} \in \mathbb{R}^{d_{l-1} \times d_l}$ is a layer-specific weight matrix, $d_l$ is the width of $l$-th layer, $d_0 = d$, $\phi(\cdot)$ is the ReLU function (Hahnloser et al., 2000), i.e., $\phi(u) = \max\{0, u\}$, which is monotonically increasing 1-Lipschitz activation function. The graph filter $g(\boldsymbol{A}) : \mathbb{R}^{n \times n} \mapsto \mathbb{R}^{n \times n}$ is a function of the adjacency matrix $\boldsymbol{A}$, such as

$$
\begin{aligned}
g(\boldsymbol{A}) &= \boldsymbol{A} + \boldsymbol{I}_n && \text{the graph with self-loops (Xu et al., 2018)} \\
g(\boldsymbol{A}) &= \boldsymbol{D}^{-1} \boldsymbol{A} && \text{the random-walk graph (Zhang et al., 2019)} \\
g(\boldsymbol{A}) &= \boldsymbol{D}^{-1/2} \boldsymbol{A} \boldsymbol{D}^{-1/2} && \text{the symmetric normalized graph (Kipf \& Welling, 2017)}
\end{aligned}
$$

where $\boldsymbol{I}_n$ is the identity matrix, and $\boldsymbol{D}$ is the degree matrix defined by $\boldsymbol{D}_{i,i} = \sum_{j \in [n]} \boldsymbol{A}_{i,j}$.

## 4 MAIN RESULTS

### 4.1 GENERAL ANALYSIS: BINARY CLASSIFICATION

Let the label $y$ takes values in $\{-1, +1\}$, and $\mathcal{F} : \mathbb{R}^{n \times d} \mapsto \mathbb{R}^n$ be the function class of multi-layer GCNs defined in (1). We predict the label of node $i$ with the sign of $f(\boldsymbol{A}, \widetilde{\boldsymbol{X}})_i$ for any

$f \in \mathcal{F}$. Assume that the loss function $\ell(f(\boldsymbol{A}, \boldsymbol{X})_i, y_i) \equiv \hat{\ell}(y_i f(\boldsymbol{A}, \boldsymbol{X})_i)$ where $\hat{\ell} : \mathbb{R} \to \mathbb{R}_+$ is monotonically nonincreasing and $L_\ell$-Lipschitz, the following equation holds:

$$\widetilde{\ell}(f(\boldsymbol{A}, \boldsymbol{X})_i, y_i) = \max_{\widetilde{\boldsymbol{X}} \in \mathcal{B}_r^\varepsilon(\boldsymbol{X})} \ell(f(\boldsymbol{A}, \widetilde{\boldsymbol{X}})_i, y_i) = \hat{\ell}(\min_{\widetilde{\boldsymbol{X}} \in \mathcal{B}_r^\varepsilon(\boldsymbol{X})} y_i f(\boldsymbol{A}, \widetilde{\boldsymbol{X}})_i).$$

Note that this assumption is a mild condition encompassing some common losses such as the hinge loss and logistic loss, which has been widely used in adversarial learning literature (Awasthi et al., 2020; Xiao et al., 2022) to derive the non-trivial bounds. According to the Ledoux-Talagrand contraction inequality (Ledoux & Talagrand, 2013), we have

$$\mathfrak{R}_{m,n}(\widetilde{\ell} \circ \mathcal{F}) \leq L_\ell \mathfrak{R}_{m,n}(\widetilde{\mathcal{F}}), \tag{3}$$

where

$$\widetilde{\mathcal{F}} := \left\{ (\boldsymbol{A}, \boldsymbol{X}) \mapsto \min_{\widetilde{\boldsymbol{X}} \in \mathcal{B}_r^\varepsilon(\boldsymbol{X})} y_i f(\boldsymbol{A}, \widetilde{\boldsymbol{X}})_i : f \in \mathcal{F} \right\}. \tag{4}$$

The above inequality allows us to bound the TRC of the adversarial loss class $\mathfrak{R}_{m,n}(\widetilde{\ell} \circ \mathcal{F})$ by controlling the adversarial TRC of function class $\mathfrak{R}_{m,n}(\widetilde{\mathcal{F}})$, which is presented in the following theorem. The proof is provided in Appendix B.

**Theorem 4.1.** *Let $\mathcal{F} : \mathbb{R}^{n \times d} \to \mathbb{R}^n$ be the L-layer GCN function class defined in (1), and $\widetilde{\mathcal{F}}$ be its adversarial counterpart with the form of (4). We have*

$$\mathfrak{R}_{m,n}(\widetilde{\mathcal{F}}) \leq Q_{m,n}(\sqrt{2\log(2)L} + 1)\|g(\boldsymbol{A})\|_\infty^L \omega^L \big(B_{p^*}\|\boldsymbol{X}\|_{2,p^*} + \varepsilon s(r^*, p, d)\big),$$

*where $B_{p*} = \sqrt{2\log(2d)}$, if $p = 1$; $B_{p*} = \sqrt{2}[\frac{\Gamma(\frac{1+p^*}{2})}{\sqrt{\pi}}]^{\frac{1}{p^*}}$, if $p \in (1, 2]$; $B_{p*} = 1$, if $p \in [2, +\infty)$, $Q_{m,n} = \sqrt{\frac{2n}{m(n-m)}}$, and $s(r^*, p, d) = d^{\max\{0, \frac{1}{r^*} - \frac{1}{p}\}}$.*

**Remark 4.2.** The adversarial TRC bound above has an unavoidable polynomial dimension dependency, i.e., $s(r^*, p, d)$ as compared to its natural counterpart, which arises from the mismatch between the $p$-norm on the weight $\boldsymbol{W}^{(1)}$ and the $r$-norm in the adversarial node set $\mathcal{B}_r^\varepsilon(\boldsymbol{X})$. One could avoid such a dimension dependency by applying a perturbation-dependent norm regularizer on the weight matrix. Namely, for arbitrary $\ell_r$-norm perturbations and $r \geq 1$, the $\ell_p$-norm regularizer that satisfies $p \in [1, r^*]$ should be chosen such that $s(r^*, p, d) \equiv 1$, where $\frac{1}{r} + \frac{1}{r^*} = 1$. In contrast with related work on adversarial learning (Yin et al., 2019; Awasthi et al., 2020; Mustafa et al., 2022), our theory is the first touch for the generalization analysis of graph-structured data with the $\ell_r$-norm additive perturbations for $r \geq 1$.

**Remark 4.3.** For a $L$-layer GCN, the generalization gap might increase exponentially with the number of layers $L$ leading to a vacuous bound, which explains why stacking too many layers tends to deteriorate the performance of GCN models (Kipf & Welling, 2017; Li et al., 2018). It is worth noting that if $\omega = \mathcal{O}(1/\|g(\boldsymbol{A})\|_\infty)$ or selecting an appropriate graph filter, one can significantly weaken depth dependency and tighten the bound. For the graph with self-loops, $\|g(\boldsymbol{A})\|_\infty = 1 + D_{\max}$, while $\|g(\boldsymbol{A})\|_\infty$ has a maximum value $\sqrt{D_{\max}/D_{\min}}$ for the symmetric normalized graph, and can be equal to 1 for the random-walk graph, where $D_{\max}$ and $D_{\min}$ denote the maximum and minimum degrees, respectively. This also demonstrates the benefit of normalized graph filters for reducing generalization error (Kipf & Welling, 2017; Zhang et al., 2019).

**Remark 4.4.** Taking $\varepsilon = 0$ and applying Lemma 4.5 yield the upper bound of the generalization gap in the non-adversarial setting:

$$\mathcal{O}\Big( \max\{\frac{1}{\sqrt{m}}, \frac{1}{\sqrt{n-m}}\} \times \big(\sqrt{2\log(2)L} + 1\big)\|g(\boldsymbol{A})\|_\infty^L \omega^L B_{p^*}\|\boldsymbol{X}\|_{2,p^*}\Big),$$

which has comparable convergence rate of $\mathcal{O}(\max\{\frac{1}{\sqrt{m}}, \frac{1}{\sqrt{n-m}}\})$ to the existing TRC-based bound (Esser et al., 2021; Tang & Liu, 2023). Notably, our bound improves the existing exponential dependency of the number of layers to a logarithmic term $\mathcal{O}(\sqrt{2\log(2)L})$, facilitating the tighter bound than (Esser et al., 2021; Tang & Liu, 2023), which benefit from the usage of the contraction technique.

## 4.2 GENERAL ANALYSIS: MULTI-CLASS CLASSIFICATION

We turn to the multi-class classification with the standard margin bound framework. In $K$-category classification problems, we define the label $y \in [K]$ and consider the hypothesis class $\mathcal{F} : \mathbb{R}^{n \times d} \mapsto \mathbb{R}^{n \times K}$. For a given $f \in \mathcal{F}$, we carry out prediction for node $i$ by $\arg \max_{y_i' \in [K]} [f(\boldsymbol{A}, \boldsymbol{X})_i]_{y_i'}$. The quality of prediction is measured by the ramp loss defined by

$$\ell_\gamma(\boldsymbol{v}, y) = \begin{cases} 1 & M(\boldsymbol{v}, y) \leq 0 \\ 1 - M(\boldsymbol{v}, y)/\gamma & 0 < M(\boldsymbol{v}, y) < \gamma \\ 0 & M(\boldsymbol{v}, y) \geq \gamma, \end{cases}$$

where $M(\boldsymbol{v}, y) := \boldsymbol{v}_y - \max_{j \neq y} \boldsymbol{v}_j$ denotes the margin operator. It is worth noting that $\ell_\gamma(\boldsymbol{v}, y)$ is $\| \cdot \|_\infty$-Lipschitz with constant $\frac{1}{\gamma}$ and is an upper bound on the zero-one loss (Mustafa et al., 2022). The corresponding adversarial loss is defined by

$$\widetilde{\ell}(f(\boldsymbol{A}, \boldsymbol{X})_i, y_i) = \max_{\widetilde{\boldsymbol{X}} \in \mathcal{B}_r^\varepsilon(\boldsymbol{X})} \ell_\gamma(f(\boldsymbol{A}, \widetilde{\boldsymbol{X}})_i, y_i).$$

Previous work (Yin et al., 2019) considers a surrogate margin loss based on a semidefinite programming (SDP) based relaxation (Raghunathan et al., 2018) to address the outer maximization problem of adversarial losses for multi-class classification. Since the SDP-based approach essentially derives an upper bound of the surrogate of adversarial loss rather than the upper bound on the original adversarial loss, the resulting surrogate often overestimates the adversarial loss, potentially leading to the meaningless bound. In addition, this surrogate is only applicable to one-hidden-layer neural networks. Unlike the aforementioned work, we consider pairwise margin-bound analysis w.r.t. adversarial perturbations, yielding a tighter upper bound on the original adversarial loss and enabling multi-layer network architectures.

**Lemma 4.5.** *Let the robust surrogate loss be defined by*

$$\widehat{\ell}(f(\boldsymbol{A}, \boldsymbol{X})_i, y_i) = \ell_\gamma(M(f(\boldsymbol{A}, \boldsymbol{X})_i, y_i) - \Psi(f(\boldsymbol{A}, \boldsymbol{X})_i)),$$

*where the worst-case error is*

$$\Psi(f(\boldsymbol{A}, \boldsymbol{X})_i) = 2 \max_{k \in [K]} \varepsilon s(r^*, p, d) \| g(\boldsymbol{A}) \|_\infty^L \| \boldsymbol{W}_{*k}^{(L)} \|_1 \prod_{l=2}^{L-1} \| \boldsymbol{W}^{(l)} \|_2 \| \boldsymbol{W}^{(1)} \|_p,$$

*where $s(r^*, p, d) = d^{\max\{0, \frac{1}{r^*} - \frac{1}{p}\}}$. Then, we have*

$$\max_{\widetilde{\boldsymbol{X}} \in \mathcal{B}_r^\varepsilon(\boldsymbol{X})} \mathbb{1}\{y_i \neq \arg \max_{y' \in [K]} [f(\boldsymbol{A}, \widetilde{\boldsymbol{X}})_i]_{y'}\}$$

$$\leq \widetilde{\ell}(f(\boldsymbol{A}, \boldsymbol{X})_i, y_i) \leq \widehat{\ell}(f(\boldsymbol{A}, \boldsymbol{X})_i, y_i) \leq \mathbb{1}\{M(f(\boldsymbol{A}, \boldsymbol{X})_i, y_i) - \Psi(f(\boldsymbol{A}, \boldsymbol{X})_i) \leq \gamma\}.$$

**Remark 4.6.** The proof is provided in Appendix C. The robust surrogate loss explicitly characterizes the standard error $M(f(\boldsymbol{A}, \boldsymbol{X})_i, y_i)$ regarded as an optimization objective in the standard training and the worst-case error $\Psi(f(\boldsymbol{A}, \boldsymbol{X})_i)$ incurred by adversarial perturbations that should be suppressed. The proposed robust loss can thus be used to adversarially train robust models to withstand adversarial perturbations. It is noteworthy that the magnitude of the perturbation applied during training should be controlled such that the worst-case error term is smaller than the standard error term.

With the Ledoux-Talagrand contraction inequality and Lemma 4.5, we obtain the following structural result

$$\mathfrak{R}_{m,n}(\widetilde{\ell} \circ \mathcal{F}) \leq \mathfrak{R}_{m,n}(\widehat{\ell} \circ \mathcal{F}) \leq \frac{1}{\gamma}(\mathfrak{R}_{m,n}(M \circ \mathcal{F}) + \mathfrak{R}_{m,n}(\Psi \circ \mathcal{F})).$$

In the following theorem, we present the TRC-based generalization bound of GCNs for multi-class node classification tasks by applying Lemma 3.2. The proof is provided in Appendix C.

**Theorem 4.7.** *Let $\mathcal{F} : \mathbb{R}^{n \times d} \mapsto \mathbb{R}^{n \times K}$ be the class of $L$-layer GCNs as defined in (1). Consider the robust surrogate loss defined in Lemma 4.5. For any fixed $\gamma > 0$, with probability at least $1 - \delta$,*

$$\frac{1}{n - m} \sum_{i=m+1}^{n} \mathbb{1}\{\exists \widetilde{\boldsymbol{X}} \in \mathcal{B}_r^\varepsilon(\boldsymbol{X}) \ s.t. \ y_i \neq \arg \max_{y' \in [K]} [f(\boldsymbol{A}, \widetilde{\boldsymbol{X}})_i]_{y'}\}$$

$$\leq \frac{1}{m} \sum_{i=1}^{m} \mathbb{1}\{[f(\boldsymbol{A}, \boldsymbol{X})_i]_{y_i'} \leq \gamma + \max_{y_i' \neq y'} [f(\boldsymbol{A}, \boldsymbol{X})_i]_{y'} + \Psi(f(\boldsymbol{A}, \boldsymbol{X})_i)\} + \mathfrak{R}_{m,n}(\widehat{\ell} \circ \mathcal{F}) + O_{m,n},$$

where $O_{m,n} = \mathcal{O}(\max\{\frac{1}{\sqrt{m}}, \frac{1}{\sqrt{n-m}}\})$,

$$\mathfrak{R}_{m,n}(\widehat{\ell} \circ \mathcal{F}) \leq Q_{m,n} \frac{4K}{\gamma}(\sqrt{\log(2)L} + 1)\|g(\boldsymbol{A})\|_\infty^L \omega^L \left(B_{p^*}\|\boldsymbol{X}\|_{2,p^*} + \varepsilon s(r^*, p, d)\right),$$

and $Q_{m,n}$, $B_{p^*}$, $s(r^*, p, d)$ are as given in Theorem 4.1.

**Remark 4.8.** Similarly, the upper bound above suffers from an additional perturbation-relevant term as compared to its non-adversarial counterpart, that is, $\mathcal{O}(\varepsilon s(r^*, p, d))$. As discussed in Remark 4.2 and 4.3, one could confine this complexity term and narrow the generalization gap by applying $p$-norm regularizer on the weight to avoid polynomial dimension dependency in $s(r^*, p, d)$, where $p \in [1, r^*]$ and $\frac{1}{r^*} + \frac{1}{r} = 1$, and choosing the factor $\omega = \mathcal{O}(1/\|g(\boldsymbol{A})\|_\infty)$ or the appropriate graph filter to mitigate depth dependency.

**Remark 4.9.** The convergence rate of $\mathcal{O}(K)$ in the number of classes $K$ is comparable with the existing generalization bounds for traditional multi-class classification tasks (Yin et al., 2019; Tu et al., 2019). In particular, when $K = 2$, the above bound can be viewed as a special case of Theorem 4.1, in which the loss function is fixed to the ramp loss. Letting $\varepsilon = 0$, we obtain the high-probability generalization bound of GCNs for multi-class classification, which fills a theoretical gap in the multi-class node classification task to our knowledge.

## 5   GENERALIZATION GAP FOR GCN VARIANTS

Recently, various variants of GCNs have achieved tremendous success in improving the generalization ability of deep GCNs, encompassing SGC, Residual GCN, and GCNII. In this section, we provide explicit generalization bounds for these popular variants through the extension of our theoretical analysis, elucidating the role of network architectures on the generalization performance of deep GCNs in adversarial settings. Here, we consider the case of the $K$-category classification task.

**SGC.**   Wu et al. (2019) propose Simple Graph Convolution (SGC) by removing nonlinearities in Vanilla GCNs (Kipf & Welling, 2017). The resulting linear model is

$$f(\boldsymbol{A}, \boldsymbol{X}) = \mathrm{Softmax}(g(\boldsymbol{A})^L \boldsymbol{X} \boldsymbol{W}^{(1)} \cdots \boldsymbol{W}^{(L)}),$$

where $\boldsymbol{W}^{(l)} \in \mathbb{R}^{d_{l-1} \times d_l}$ for $l \in [L]$, and $d_l$ is the width of $l$-th layer ($d_0 = d$ and $d_L = K$).

**Proposition 5.1.** *For any $\delta \in (0, 1)$, with probability $1 - \delta$,*

$$\mathrm{Gen}(f) \leq \mathcal{O}(\max\{\frac{1}{\sqrt{m}}, \frac{1}{\sqrt{n-m}}\}) + Q_{m,n} \frac{2K}{\gamma}\|g(\boldsymbol{A})^L\|_\infty \omega^L (B_{p^*}\|\boldsymbol{X}\|_{2,p^*} + \varepsilon s(r^*, p, d)),$$

*where $Q_{m,n}$, $B_{p^*}$, and $s(r^*, p, d)$ are as given in Theorem 4.1.*

**Remark 5.2.** The proof is provided in Appendix D. It is worth noting that $\|g(\boldsymbol{A})^L\|_\infty \leq \|g(\boldsymbol{A})\|_\infty^L$, thereby alleviating the negative impact of perturbation-relevant term and leading to a tighter generalization bound in Proposition 5.1 than in Theorem 4.7. This provides the theoretical understanding of why linear models can achieve comparable and even better generalization performance than nonlinear models. It is natural that if a linear GCN has a small training error, it will also perform well on test samples based on the small generalization gap. Furthermore, for $L$-layer SGC, the aggregated information can contain the feature information of all $L$-hop-away neighbor nodes, thereby significantly improving the representation power of deep GCNs while avoiding over-smoothing (i.e., as depth increases, the representations of nodes are inclined to converge to a certain value, resulting in performance degradation) (Chen et al., 2020a).

**Residual GCN.**   Kipf & Welling (2017) facilitate the training of deep GCNs by adding residual connections (He et al., 2016) between hidden layers that carry information from the previous layer. The forward propagation is defined by

$$\boldsymbol{H}^{(l)} = \phi(g(\boldsymbol{A})\boldsymbol{H}^{(l-1)}W^{(l)}) + \boldsymbol{H}^{(l-1)}, \quad \boldsymbol{H}^{(0)} = \phi(\boldsymbol{X}\boldsymbol{W}^{(0)}),$$

where $\boldsymbol{W}^{(l)} \in \mathbb{R}^{d' \times d'}$ for $l \in [L-1]$, $\boldsymbol{W}^{(0)} \in \mathbb{R}^{d \times d'}$. The final output of the model is expressed by $f(\boldsymbol{A}, \boldsymbol{X}) = \mathrm{Softmax}(\boldsymbol{H}^{(L-1)}\boldsymbol{W}^{(L)})$, where $\boldsymbol{W}^{(L)} \in \mathbb{R}^{d' \times K}$.

Table 1: Dataset statistics.

| Dataset | Classes | Nodes | Edges | Features | Training | Validation | Test |
|---|---|---|---|---|---|---|---|
| Citeseer | 6 | 3,327 | 4,732 | 3,703 | 20 per class | 500 | 1000 |
| Cora | 7 | 2,708 | 5,429 | 1,433 | 20 per class | 500 | 1000 |
| Pubmed | 3 | 19,717 | 44,338 | 500 | 20 per class | 500 | 1000 |
| CS | 15 | 18,333 | 81,894 | 6,805 | 20 per class | 30 per class | Rest |
| Physics | 5 | 34,493 | 247,962 | 8,415 | 20 per class | 30 per class | Rest |
| ogbn-arxiv | 40 | 169,343 | 1,166,243 | 128 | 20 per class | 30 per class | Rest |

**Proposition 5.3.** *For any $\delta \in (0, 1)$, with probability $1 - \delta$,*

$$
\begin{aligned}
\mathrm{Gen}(f) \leq & \mathcal{O}\big( \max\{\frac{1}{\sqrt{m}}, \frac{1}{\sqrt{n-m}}\}\big) \\
& + Q_{m,n} \frac{4K}{\gamma}(\sqrt{\log(2)L} + 1)\|g(\boldsymbol{A})\|_\infty^L \omega(\omega + 1)^L (B_{p^*}\|\boldsymbol{X}\|_{2,p^*} + \varepsilon s(r^*, p, d)).
\end{aligned}
$$

*where $Q_{m,n}$, $B_{p^*}$, and $s(r^*, p, d)$ are as given in Theorem 4.1.*

**Remark 5.4.** The proof is provided in Appendix E. The generalization bound above has a similar dependency on the number of network layers as Theorem 4.7. This implies that as the depth increases, the perturbation term will become the dominant factor and may lead to larger generalization errors. Our analysis thus provides the theoretical understanding that residual connections partially alleviate over-smoothing while degrading performance with increasing depth (Kipf & Welling, 2017).

**GCNII.** Chen et al. (2020b) effectively enhance the prediction performance of deep GCNs by building an initial residual connection to the first layer, motivated by (He et al., 2016; Kipf & Welling, 2017). The propagation process is

$$
\boldsymbol{H}^{(l)} = \phi(((1-\alpha)g(\boldsymbol{A})\boldsymbol{H}^{(l-1)} + \alpha\boldsymbol{H}^{(l)})((1-\beta)\boldsymbol{I}_n + \beta\boldsymbol{W}^{(l)})), \quad \boldsymbol{H}^{(0)} = \phi(\boldsymbol{X}\boldsymbol{W}^{(0)})
$$

where $\alpha, \beta \in (0, 1)$, $\boldsymbol{W}^{(l)} \in \mathbb{R}^{d' \times d'}$ for $l \in [L-1]$, and $\boldsymbol{W}^{(0)} \in \mathbb{R}^{d \times d'}$. The final output is defined by $f(\boldsymbol{A}, \boldsymbol{X}) = \mathrm{Softmax}(\boldsymbol{H}^{(L-1)}\boldsymbol{W}^{(L)})$, where $\boldsymbol{W}^{(L)} \in \mathbb{R}^{d' \times K}$.

**Proposition 5.5.** *For any $\delta \in (0, 1)$, with probability $1 - \delta$,*

$$
\mathrm{Gen}(f) \leq \mathcal{O}(\max\{\frac{1}{\sqrt{m}}, \frac{1}{\sqrt{n-m}}\}) + Q_{m,n}\frac{4K}{\gamma}(\sqrt{\log(2)L} + 1)(B_{p^*}\|\boldsymbol{X}\|_{2,p^*} + \varepsilon s(r^*, p, d))
$$

$$
\times \omega^2\big((1-\alpha)\|g(\boldsymbol{A})\|_\infty^L(1-\beta+\beta\omega)^L + \alpha(1-\alpha)\sum_{l=0}^{L}\|g(\boldsymbol{A})\|_\infty^l(1-\beta+\beta\omega)^l\big).
$$

*where $Q_{m,n}$, $B_{p^*}$, and $s(r^*, p, d)$ are as given in Theorem 4.1.*

**Remark 5.6.** The proof is provided in Appendix F. A comparison of Proposition 5.3 and Proposition 5.5 indicates that the product term of multiple norm bounds can be confined to the sum term via a tunable parameter $\alpha$. This implies that as $\alpha$ increases, the perturbation term will significantly weaken the dependency on depth and be well suppressed, thereby reducing the generalization error. Additionally, it is noteworthy that if $\beta$ approaches zero and $\|g(\boldsymbol{A})\|_\infty \leq 1$, the upper bound in Proposition 5.5 is independent of the number of layers and can be considerably narrowed. The reason behind this behavior is that as $\alpha$ increases and $\beta$ decreases, the network architecture is close to the shallow model, which prevents the layer-by-layer propagation of adversarial perturbations. Hence, we posit that initial residual connection confers greater benefits to the generalization ability of deeper GCNs, being corroborated by certain empirical observations (Chen et al., 2020b; Liu et al., 2021).

## 6 EXPERIMENTS

In this section, we evaluate the impact of some key quantities on the generalization performance of GNC in adversarial settings, such as feature dimension, regularizer, graph filters, the number of layers, etc. Extensive experimental results validate our theoretical findings in Sections 4&5.

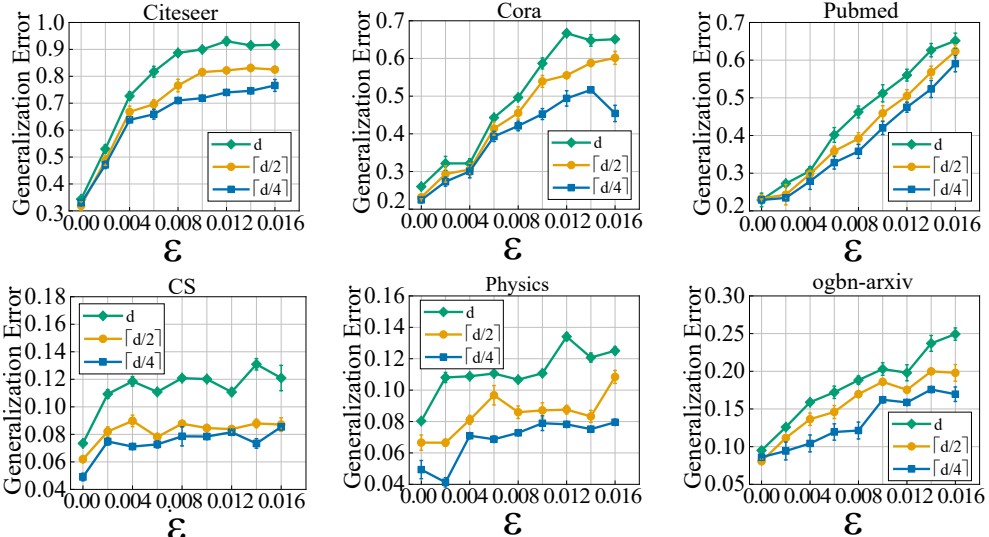

Figure 1: The empirical generalization error (mean value and standard deviation) with different feature dimensions. $\varepsilon$ denotes the maximum allowable perturbation.

## 6.1 EXPERIMENTAL SETUP

We adopt several widely-used benchmark datasets, including Citeseer, Cora, Pubmed, CS, Physics, and ogbn-arxiv (Sen et al., 2008; Yang et al., 2016; Hu et al., 2020). Statistics of the datasets are summarized in Table 1. We adversarially train a robust model by leveraging the following objective:

$$\min_{f \in \mathcal{F}} \max_{\widetilde{\boldsymbol{X}} \in \mathcal{B}_r^\varepsilon(\boldsymbol{X})} \sum_{i=1}^{m} \ell(f(\boldsymbol{A}, \widetilde{\boldsymbol{X}})_i, y_i) + \lambda \|\boldsymbol{W}\|_1, \tag{5}$$

where $\ell(\cdot)$ is cross-entropy loss, $\boldsymbol{W}$ denotes the weight parameter of the first layer, $\lambda \geq 0$ denotes the regularization coefficient, and $\varepsilon$ denotes the maximum allowable perturbation. The training iterations is fixed to 600. During training and testing, the adversarial nodes are generated by the $\ell_\infty$-PGD algorithm (Madry et al., 2018) with the step size $\varepsilon/128$, where adversarial perturbations are added to test nodes after training to avoid a biased evaluation through memorization of the transductive learning setting (Gosch et al., 2024). Similar to previous work (Xiao et al., 2022; Zou & Liu, 2023), we consider an empirical proxy for the generalization gap:

$$\big|\text{Adversarial Training Accuracy} - \text{Adversarial Test Accuracy}\big|$$

that is, the absolute value of the difference between the accuracy on adversarial training and test nodes. Each experiment is independently repeated 10 times and reported with the mean value and standard deviations. We default to present the experimental results of two-layer GCN proposed by (Kipf & Welling, 2017). Please refer to Appendix I for more detailed experimental configurations and experimental results, including different attack methods, SGC, GCNII, and Residual GCN.

## 6.2 NUMERICAL DISCUSSION

**Feature dimension.** We compare the empirical generalization error with different dimensions, including the original dimension $d$, $\lceil d/2 \rceil$, and $\lceil d/4 \rceil$. For convenience, we use a single-layer neural network with ReLU activation to learn a low-dimensional representation of the node features. As shown in Figure 1, the empirical generalization error decreases steadily with the dimension, which implies that low-dimensional feature projection can help reduce the generalization error.

**Regularization.** Following the theoretical findings in Theorem 4.1&4.7, we apply $\ell_1$-norm regularizer on the weight matrix, since $\ell_\infty$-norm attack is used to generate adversarial perturbations. We evaluate the effect of norm regularization on the generalization ability by comparing the empirical

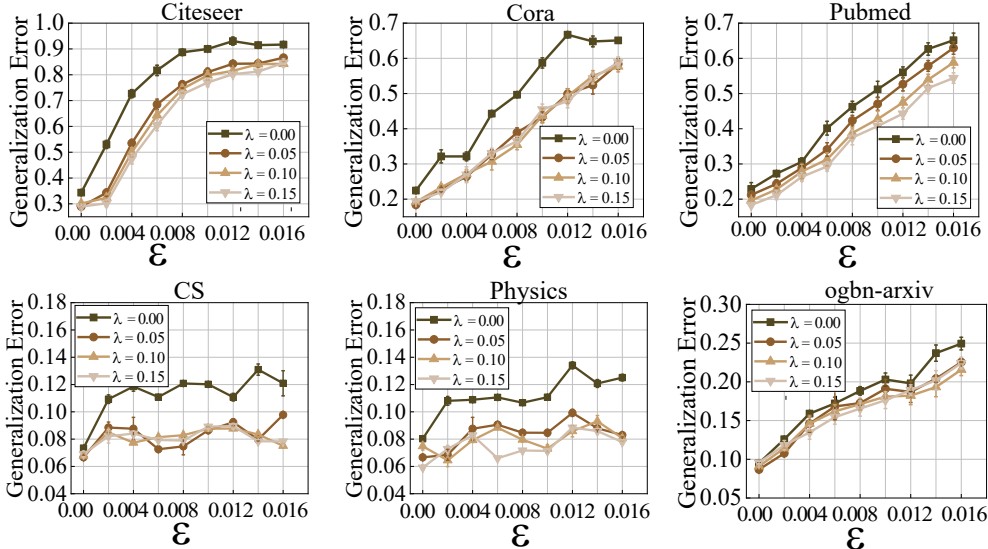

Figure 2: The empirical generalization error (mean value and standard deviation) of models trained with $\ell_1$ regularization for different regularization parameters (i.e., $\lambda$). $\varepsilon$ denotes the maximum allowable perturbation.

generalization error with different regularization coefficients $\lambda$. As shown in Figure 2, the empirical generalization error of the regularized model is smaller than that without (i.e., $\lambda = 0$), which is consistent with our theoretical analysis. Experimental results demonstrate the importance of appropriate regularizer to achieve good generalization performance.

**Graph filter.**  We present the empirical generalization error with different graph filters in Figure 3, where the number of layers is set to 6. As shown in Figure 3, the graph with self-loops has larger empirical generalization errors than the normalized graphs. Hence, we argue that normalizing the graph matrix can facilitate the adversarial generalization of GCNs.

**Model depth.**  We compare the empirical generalization error with different depths in Figure 4. The experimental results show that the generalization error increases as the number of layers increases and tends to be stable or even decreases due to the over-smoothing issue. This suggests that the appropriate number of layers should be determined to balance the representation power and generalization capability.

**Network architecture.**  We investigate the generalization ability of popular GCNs with adversarial perturbations, including Vanilla GCN, SGC, Residual GCN, and GCNII, where the number of layers is set to 6. Figure 5 presents the empirical generalization error of different models with $\ell_\infty$ PGD attacks. Emiprical observations show that GCNII with initial residual connection tends to have smaller generalization error, which demonstrates the effectiveness of the specific network structure in enhancing adversarial robustness. Furthermore, we evaluate the role of the parameter $\alpha$ on the generalization ability of GCNII. As shown in Figure 6, the larger $\alpha$, the smaller empirical generalization error, which is consistent with our theoretical findings in Proposition 5.5.

**Labeled node size.**  We study the effect of the number of labeled nodes on the generalization ability of the learned model in the node classification task. Specifically, we compare the empirical generalization error with different label rates $m/n$, where $m/n$ denotes the number of labeled nodes used for training divided by the total number of nodes. As shown in Figure 7, when label rete $m/n$ is too large or too small, the generalization gap will be at a large level, which is aligned with the general consensus (El-Yaniv & Pechyony, 2009; Esser et al., 2021). This implies that the amount of labeled data should be taken into consideration to achieve excellent prediction performance.

# 7 CONCLUSION

In this paper, we provide a comprehensive generalization analysis for GCNs under perturbation attacks through the lens of the adversarial TRC. The derived bounds provide a theoretical characterization of the interplay between the generalization error, node perturbations, and adversarial robustness. Theoretical results reveal how graph-structured data and model parameters can help improve adversarially robust generalization of GCNs. Furthermore, we develop the generalization bounds for popular variants of GCNs, which implies that specific network architecture (e.g., initial residual connection) is beneficial for enhancing adversarial robustness. Extensive experimental results on benchmark datasets validate our theoretical findings. Interesting directions for future work include analyzing the generalization properties for GCNs under topology attacks.

## ACKNOWLEDGMENTS

This work was supported by the National Natural Science Foundation of China (Nos. 62376104, 12426512, and 11801201), and HZAU-AGIS Cooperation Fund (No. SZYJY2023010).

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

# A  NOTATION

For a matrix $\boldsymbol{W} \in \mathbb{R}^{m \times n}$, the $(p, q)$-norm is defined as $\|\boldsymbol{W}\|_{p,q} = \|(\|\boldsymbol{W}_{*1}\|_p, \ldots, \|\boldsymbol{W}_{*n}\|_p)\|_q$, where $\boldsymbol{W}_{*i}$ is the $i$-column of $\boldsymbol{W}$. We use the shorthand notation $\|\cdot\|_p \equiv \|\cdot\|_{p,p}$. We denote the infinity norm of the matrix by $\|\boldsymbol{W}\|_\infty = \max_{1 \in [m]} \sum_{j=1}^n |\boldsymbol{W}_{i,j}|$. For ease of exposition, we summarize the notations in Table 2.

Table 2: Summary of notations involved in this paper.

| Notations | Meaning |
|---|---|
| $\boldsymbol{x}_i$ | The feature of node $i$, $\boldsymbol{x}_i \in \mathbb{R}^d$. |
| $y_i$ | The label of node $i$. |
| $\boldsymbol{X}$ | The feature matrix of all nodes $\boldsymbol{X} = [\boldsymbol{x}_1, \ldots, \boldsymbol{x}_n] \in \mathbb{R}^{n \times d}$. |
| $\boldsymbol{y}$ | The vector of labels, i.e., $\boldsymbol{y} = (y_1, y_2, \ldots, y_n) \in \mathbb{R}^n$. |
| $g(\boldsymbol{A})$ | The graph filter, i.e., a function of the adjacency matrix $\boldsymbol{A}$. |
| $\varepsilon$ | The adversarial perturbation. |
| $\mathcal{B}_r^\varepsilon(\boldsymbol{X})$ | The set of adversarial node $\{\widetilde{\boldsymbol{X}} = [\widetilde{\boldsymbol{x}}_1, \ldots, \widetilde{\boldsymbol{x}}_n] : \|\widetilde{\boldsymbol{x}}_i - \boldsymbol{x}_i\|_r \leq \varepsilon, r \geq 1, i \in [n]\}$. |
| $f(\boldsymbol{A}, \boldsymbol{X})$ | The function of $L$-layer GCNs. |
| $f(\boldsymbol{A}, \boldsymbol{X})_i$ | The $i$-th element of the hypothesis $f$. |
| $\boldsymbol{H}^{(l)}$ | The feature representation of all nodes at $l$-th layer. |
| $\widetilde{\boldsymbol{H}}^{(l)}$ | The feature representation of all adversarial nodes at $l$-th layer. |
| $\boldsymbol{W}^{(l)}$ | The weight matrix of $l$-th layer. |
| $\omega$ | The norm bounds of the weight matrix. |
| $\mathfrak{R}_{m,n}(\mathcal{F})$ | The TRC of the function class $\mathcal{F}$. |
| $\ell$ | The loss function. |
| $L_\ell$ | The Lipschitz constant of function $\ell$. |
| $\phi(\cdot)$ | The non-decreasing 1-Lipschitz activation function, e.g. ReLU activation. |
| $p^*$ | The Hölder conjugates by a star, e.g. $p^*$, satisfying $\frac{1}{p} + \frac{1}{p^*} = 1$. |
| $[L]$ | The set of positive integers, i.e., $[L] = \{1, \ldots, L\}$. |
| $\|\cdot\|_p, \|\cdot\|_r$ | The $\ell_p$-norm and the $\ell_r$-norm, $p, r \geq 1$. |

Before proceeding to prove main results, we introduce some necessary inequalities.

**Lemma A.1.** *(Awasthi et al., 2020) Let $1 \leq p, r \leq \infty$ and $d$ be the dimension. Then,*

$$\sup_{\|\boldsymbol{w}\|_p \leq 1} \|\boldsymbol{w}\|_{r^*} = s(r^*, p, d),$$

*where $s(r^*, p, d) = d^{\max\{0, \frac{1}{r^*} - \frac{1}{p}\}}$.*

**Lemma A.2.** *Let $\phi$ be a 1-Lipschitz positive-homogeneous activation function. Then for any class of vector-valued functions $\mathcal{F}$, and any convex and monotonically increasing function $\psi : \mathbb{R} \to [0, \infty)$,*

$$\mathbb{E}_{\boldsymbol{\sigma}} \sup_{f \in \mathcal{F}, \|\boldsymbol{W}\|_2 \leq \omega, j \in [n]} \psi \left( \left\| \sum_{i=1}^n \sigma_i \phi \left( \sum_{k \in [n]} g(\boldsymbol{A})_{j,k} f(\boldsymbol{A}, \boldsymbol{X})_k \boldsymbol{W} \right) \right\| \right)$$

$$\leq 2 \mathbb{E}_{\boldsymbol{\sigma}} \sup_{f \in \mathcal{F}, j \in [n]} \psi \left( \|g(\boldsymbol{A})\|_\infty \omega \left\| \sum_{i=1}^n \sigma_i f(\boldsymbol{A}, \boldsymbol{X})_j \right\| \right)$$

*Proof of Lemma A.2.* Let $\boldsymbol{W}_{*1}, \boldsymbol{W}_{*2}, \ldots, \boldsymbol{W}_{*n}$ be the columns of the matrix $\boldsymbol{W}$, we have

$$\max_{j \in [n]} \Big\| \sum_{i=1}^{n} \sigma_i \phi \Big( \sum_{k \in [n]} g(\boldsymbol{A})_{j,k} f(\boldsymbol{A}, \boldsymbol{X})_k \boldsymbol{W} \Big) \Big\|^2$$

$$\leq \max_{t \in [n]} \Big\| \sum_{i=1}^{n} \sigma_i \phi \Big( (\max_{j \in [n]} \sum_{k \in [n]} g(\boldsymbol{A})_{j,k}) f(\boldsymbol{A}, \boldsymbol{X})_t \boldsymbol{W} \Big) \Big\|^2$$

$$\leq \max_{j \in [n]} \Big\| \sum_{i=1}^{n} \sigma_i \phi \Big( \|g(\boldsymbol{A})\|_\infty f(\boldsymbol{A}, \boldsymbol{X})_j \boldsymbol{W} \Big) \Big\|^2$$

$$= \max_{j \in [n]} \sum_{l=1}^{n} \|\boldsymbol{W}_{*l}\|^2 \left( \sum_{i=1}^{n} \sigma_i \phi \Big( \|g(\boldsymbol{A})\|_\infty \Big\langle f(\boldsymbol{A}, \boldsymbol{X})_j, \frac{\boldsymbol{W}_{*l}}{\|\boldsymbol{W}_{*l}\|} \Big\rangle \Big) \right).$$

The supremum of this over all $\boldsymbol{W}_{*1}, \boldsymbol{W}_{*2}, \ldots, \boldsymbol{W}_{*n}$ such that $\|\boldsymbol{W}\|_2^2 = \sum_{l=1}^{n} \|\boldsymbol{W}_{*l}\|^2 \leq \omega^2$ must be obtained when $\|\boldsymbol{W}_{*l}\| = \omega$ for some $l$, and $\|\boldsymbol{W}_{*h}\| = 0$ for all $h \neq l$. Therefore

$$\mathbb{E}_{\boldsymbol{\sigma}} \sup_{f \in \mathcal{F}, \|\boldsymbol{W}\|_2 \leq \omega, j \in [n]} \psi \left( \Big\| \sum_{i=1}^{n} \sigma_i \phi \Big( \sum_{k \in [n]} g(\boldsymbol{A})_{j,k} f(\boldsymbol{A}, \boldsymbol{X})_k \boldsymbol{W} \Big) \Big\| \right)$$

$$\leq \mathbb{E}_{\boldsymbol{\sigma}} \sup_{f \in \mathcal{F}, \|\boldsymbol{W}\|_2 \leq \omega, k \in [n]} \psi \left( \sum_{l=1}^{n} \|\boldsymbol{W}_{*l}\|^2 \left( \sum_{i=1}^{n} \sigma_i \phi \Big( \|g(\boldsymbol{A})\|_\infty \Big\langle f(\boldsymbol{A}, \boldsymbol{X})_k, \frac{\boldsymbol{W}_{*l}}{\|\boldsymbol{W}_{*l}\|} \Big\rangle \Big) \right) \right)$$

$$= \mathbb{E}_{\boldsymbol{\sigma}} \sup_{f \in \mathcal{F}, \|\boldsymbol{W}_{*l}\|_2 = \omega, j \in [n]} \psi \left( \Big| \sum_{i=1}^{n} \sigma_i \phi \Big( \|g(\boldsymbol{A})\|_\infty \langle f(\boldsymbol{A}, \boldsymbol{X})_j, \boldsymbol{W}_{*l} \rangle \Big) \Big| \right)$$

$$\leq 2\mathbb{E}_{\boldsymbol{\sigma}} \sup_{f \in \mathcal{F}, \|\boldsymbol{W}_{*l}\|_2 = \omega, j \in [n]} \psi \left( \sum_{i=1}^{n} \sigma_i \phi \Big( \|g(\boldsymbol{A})\|_\infty \langle f(\boldsymbol{A}, \boldsymbol{X})_j, \boldsymbol{W}_{*l} \rangle \Big) \right) \tag{6}$$

where the last inequality follows from $\psi(|u|) \leq \psi(u) + \psi(-u)$ and the symmetry in the distribution of the random variables $\sigma_i$. For inequality (6), having

$$2\mathbb{E}_{\boldsymbol{\sigma}} \sup_{f \in \mathcal{F}, \|\boldsymbol{W}_{*l}\|_2 = \omega, j \in [n]} \psi \left( \sum_{i=1}^{n} \sigma_i \phi \Big( \|g(\boldsymbol{A})\|_\infty \langle f(\boldsymbol{A}, \boldsymbol{X})_j, \boldsymbol{W}_{*l} \rangle \Big) \right)$$

$$\leq 2\mathbb{E}_{\boldsymbol{\sigma}} \sup_{f \in \mathcal{F}, \|\boldsymbol{W}_{*l}\|_2 = \omega, j \in [n]} \psi \left( \sum_{i=1}^{n} \sigma_i \|g(\boldsymbol{A})\|_\infty \langle f(\boldsymbol{A}, \boldsymbol{X})_j, \boldsymbol{W}_{*l} \rangle \right)$$

$$\leq 2\mathbb{E}_{\boldsymbol{\sigma}} \sup_{f \in \mathcal{F}, \|\boldsymbol{W}_{*l}\|_2 = \omega, j \in [n]} \psi \left( \|g(\boldsymbol{A})\|_\infty \|\boldsymbol{W}_{*l}\|_2 \Big\| \sum_{i=1}^{n} \sigma_i f(\boldsymbol{A}, \boldsymbol{X})_j \Big\| \right)$$

$$= 2\mathbb{E}_{\boldsymbol{\sigma}} \sup_{f \in \mathcal{F}, j \in [n]} \psi \left( \|g(\boldsymbol{A})\|_\infty \omega \Big\| \sum_{i=1}^{n} \sigma_i f(\boldsymbol{A}, \boldsymbol{X})_j \Big\| \right).$$

$\square$

## B  PROOF OF THEOREM 4.1 [BINARY CLASSIFICATION]

*Proof of Theorem 4.1.* Let the function class of $L$-layer GCNs be defined by

$$\mathcal{F} = \Big\{ f(\boldsymbol{A}, \boldsymbol{X}) = g(\boldsymbol{A}) \boldsymbol{H}^{(L-1)} \boldsymbol{W}^{(L)} : \|\boldsymbol{W}^{(l)}\|_2, \|\boldsymbol{W}^{(l)}\|_p \leq \omega, l \in [L] \Big\} : \mathbb{R}^{n \times d} \to \mathbb{R}^n$$

with the update rule:

$$\boldsymbol{H}^{(l)} = \phi(g(\boldsymbol{A}) \boldsymbol{H}^{(l-1)} \boldsymbol{W}^{(l)}) \in \mathbb{R}^{n \times d_l}, \quad \boldsymbol{H}^{(0)} = \boldsymbol{X},$$

where the graph filter $g(\boldsymbol{A}) : \mathbb{R}^{n \times n} \mapsto \mathbb{R}^{n \times n}$, $\boldsymbol{W}^{(l)} \in \mathbb{R}^{d_{l-1} \times d_l}$, $d_l$ is the width of $l$-th layer, $d_0 = d$, $d_L = 1$, and $\phi(\cdot)$ is ReLU activation. The corresponding adversarial counterpart is defined by

$$\widetilde{\mathcal{F}} = \left\{ \inf_{\widetilde{\boldsymbol{X}} \in \mathcal{B}_r^\varepsilon(\boldsymbol{X})} y_i f(\boldsymbol{A}, \widetilde{\boldsymbol{X}})_i : f \in \mathcal{F}, y_i \in \{\pm 1\} \right\}.$$

Let the set of adversarial nodes of $\widetilde{\mathcal{F}}$ be defined by $\widehat{\boldsymbol{X}} = [\widehat{\boldsymbol{x}}_1, \ldots, \widehat{\boldsymbol{x}}_n]$, where each $\widehat{\boldsymbol{x}}_i$ is chosen by

$$\widehat{\boldsymbol{x}}_i = \arg \inf_{\widetilde{\boldsymbol{X}} \in \mathcal{B}_r^\varepsilon(\boldsymbol{X})} y_i f(\boldsymbol{A}, \widetilde{\boldsymbol{X}})_i,$$

for $i = 1, \ldots, n$ and any $f \in \mathcal{F}$. Denote $Q = \frac{1}{m} + \frac{1}{n-m}$. With the definition above, we have the following inequality

$$Q\mathbb{E}_{\boldsymbol{\sigma}}\left[ \sup_{f \in \mathcal{F}} \sum_{i=1}^n \sigma_i \inf_{\widetilde{\boldsymbol{X}} \in \mathcal{B}_r^\varepsilon(\boldsymbol{X})} y_i f(\boldsymbol{A}, \widetilde{\boldsymbol{X}})_i \right] \le Q\mathbb{E}_{\boldsymbol{\sigma}}\left[ \sup_{f \in \widehat{\mathcal{F}}} \sum_{i=1}^n \sigma_i f(\boldsymbol{A}, \widehat{\boldsymbol{X}})_i \right] := \mathfrak{R}_{m,n}(\widehat{\mathcal{F}}), \quad (7)$$

where

$$\widehat{\mathcal{F}} = \left\{ f(\boldsymbol{A}, \widehat{\boldsymbol{X}}) = g(\boldsymbol{A})\widehat{\boldsymbol{H}}^{(L-1)}\boldsymbol{W}^{(L)} : \|\boldsymbol{W}^{(l)}\|_2, \|\boldsymbol{W}^{(l)}\|_p \le \omega, l \in [L] \right\}$$

with update rule:

$$\widehat{\boldsymbol{H}}^{(l)} = \phi(g(\boldsymbol{A})\widehat{\boldsymbol{H}}^{(l-1)}\boldsymbol{W}^{(l)}), \quad \widehat{\boldsymbol{H}}^{(0)} = \widehat{\boldsymbol{X}},$$

where $g(\boldsymbol{A}) \in \mathbb{R}^{n \times n}$, $\boldsymbol{W}^{(l)} \in \mathbb{R}^{d_{l-1} \times d_l}$ for $l \in [L-1]$, and $\boldsymbol{W}^{(L)} \in \mathbb{R}^{d_{L-1} \times 1}$.

We thus turn to bound $\mathfrak{R}_{m,n}(\widehat{\mathcal{F}})$. By the definition of TRC,

$$\mathfrak{R}_{m,n}(\widehat{\mathcal{F}}) = Q\mathbb{E}_{\boldsymbol{\sigma}} \sup_{\|\boldsymbol{W}^{(L)}\|_2 \le \omega} \left[ \sum_{i=1}^n \sigma_i \left( \sum_{j \in [n]} g(\boldsymbol{A})_{i,j}\widehat{\boldsymbol{H}}_{j*}^{(L-1)}\boldsymbol{W}^{(L)} \right) \right]$$

$$\le Q\mathbb{E}_{\boldsymbol{\sigma}} \sup_{\|\boldsymbol{W}^{(L)}\|_2 \le \omega} \left[ \sum_{i=1}^n \sigma_i \left( \max_{i \in [n]} \sum_{j \in [n]} g(\boldsymbol{A})_{i,j} \right) \max_{t \in [n]} \left\langle \widehat{\boldsymbol{H}}_{t*}^{(L-1)}, \boldsymbol{W}^{(L)} \right\rangle \right]$$

$$\le Q\mathbb{E}_{\boldsymbol{\sigma}} \sup_{\|\boldsymbol{W}^{(L)}\|_2 \le \omega, j \in [n]} \|g(\boldsymbol{A})\|_\infty \left[ \sum_{i=1}^n \sigma_i \left\langle \widehat{\boldsymbol{H}}_{j*}^{(L-1)}, \boldsymbol{W}^{(L)} \right\rangle \right]$$

$$\le Q\frac{1}{\lambda} \log \mathbb{E}_{\boldsymbol{\sigma}} \sup_{\|\boldsymbol{W}^{(L)}\|_2 \le \omega, j \in [n]} \exp\left( \lambda\|g(\boldsymbol{A})\|_\infty \left( \sum_{i=1}^n \sigma_i \left\langle \widehat{\boldsymbol{H}}_{j*}^{(L-1)}, \boldsymbol{W}^{(L)} \right\rangle \right) \right)$$

$$\le Q\frac{1}{\lambda} \log \mathbb{E}_{\boldsymbol{\sigma}} \sup_{j \in [n]} \exp\left( \lambda\|g(\boldsymbol{A})\|_\infty \omega \left\| \sum_{i=1}^n \sigma_i \widehat{\boldsymbol{H}}_{j*}^{(L-1)} \right\| \right). \quad (8)$$

We rewrite inequality (8) as

$$Q\frac{1}{\lambda} \log \mathbb{E}_{\boldsymbol{\sigma}} \sup_{\|\boldsymbol{W}^{(L-1)}\|_2 \le \omega, j \in [n]} \exp\left( \lambda\|g(\boldsymbol{A})\|_\infty \omega \left\| \sum_{i=1}^n \sigma_i \phi\left( \sum_{k \in [n]} g(\boldsymbol{A})_{j,k}\widehat{\boldsymbol{H}}_{k*}^{(L-2)}\boldsymbol{W}^{(L-1)} \right) \right\| \right)$$

$$\le Q\frac{1}{\lambda} \log\left( 2\mathbb{E}_{\boldsymbol{\sigma}} \sup_{j \in [n]} \exp\left( \lambda\|g(\boldsymbol{A})\|_\infty^2 \omega^2 \left\| \sum_{i=1}^n \sigma_i \widehat{\boldsymbol{H}}_{j*}^{(L-2)} \right\| \right) \right)$$

where the last inequality follows from Lemma A.2 with $\psi(u) = \exp\{\lambda\|g(\boldsymbol{A})\|_\infty \omega \cdot u\}$. By recursion steps, we obtain

$$\mathfrak{R}_{m,n}(\widehat{\mathcal{F}}) \le Q\frac{1}{\lambda} \log\left( 2^L \mathbb{E}_{\boldsymbol{\sigma}} \sup_{\|\boldsymbol{W}^{(1)}\|_p \le \omega, j \in [n]} \exp\left( \lambda\|g(\boldsymbol{A})\|_\infty^L \omega^{L-1} \left\| \sum_{i=1}^n \sigma_i \widehat{\boldsymbol{X}}_{j*}\boldsymbol{W}^{(1)} \right\| \right) \right)$$

$$\le Q\frac{1}{\lambda} \log\left( 2^L \mathbb{E}_{\boldsymbol{\sigma}} \sup_{j \in [n]} \exp\left( \lambda\|g(\boldsymbol{A})\|_\infty^L \omega^L \left\| \sum_{i=1}^n \sigma_i \widehat{\boldsymbol{x}}_j \right\|_{p*} \right) \right). \quad (9)$$

Let $M = \|g(\boldsymbol{A})\|_\infty^L \omega^L$ and define a random variable

$$Z = M \cdot \sup_{j \in [n]} \Big\| \sum_{i=1}^n \sigma_i \widehat{\boldsymbol{x}}_j \Big\|_{p^*}$$

where random as a function of the random variables $\sigma_1, \ldots, \sigma_n$. Then,

$$\frac{1}{\lambda} \log \Big\{ 2^L \mathbb{E} \exp \lambda Z \Big\} = \frac{L \log(2)}{\lambda} + \frac{1}{\lambda} \log \{ \mathbb{E} \exp \lambda (Z - \mathbb{E}Z) \} + \mathbb{E}Z.$$

By Jensen's inequality and triangle inequality, $\mathbb{E}Z$ can be bounded by

$$
\begin{aligned}
\mathbb{E}Z =& M \cdot \mathbb{E}_{\boldsymbol{\sigma}} \sup_{j \in [n]} \Big\| \sum_{i=1}^n \sigma_i \widehat{\boldsymbol{x}}_j \Big\|_{p^*} = M \cdot \mathbb{E}_{\boldsymbol{\sigma}} \sup_{j \in [n]} \Big\| \sum_{i=1}^n \sigma_i (\widehat{\boldsymbol{x}}_j - \boldsymbol{x}_j + \boldsymbol{x}_j) \Big\|_{p^*} \\
\leq& M \cdot \mathbb{E}_{\boldsymbol{\sigma}} \sup_{j \in [n]} \Big( \Big\| \sum_{i=1}^n \sigma_i \boldsymbol{x}_j \Big\|_{p*} + \Big\| \sum_{i=1}^n \sigma_i (\widehat{\boldsymbol{x}}_j - \boldsymbol{x}_j) \Big\|_{p*} \Big) \\
\leq& M \cdot \mathbb{E}_{\boldsymbol{\sigma}} \sup_{j \in [n]} \Big( \Big\| \sum_{i=1}^n \sigma_i \boldsymbol{x}_j \Big\|_{p*} + s(r^*, p, d) \Big\| \sum_{i=1}^n \sigma_i (\widehat{\boldsymbol{x}}_j - \boldsymbol{x}_j) \Big\|_r \Big) \\
\leq& M \cdot \mathbb{E}_{\boldsymbol{\sigma}} \Big( \Big\| \sum_{i=1}^n \sigma_i \boldsymbol{x}_i \Big\|_{p*} + \varepsilon s(r^*, p, d) \Big| \sum_{i=1}^n \sigma_i \Big| \Big) \\
\leq& M \sqrt{\frac{2m(n-m)}{n}} \left( B_{p^*} \|\boldsymbol{X}\|_{2,p^*} + \varepsilon s(r^*, p, d) \right),
\end{aligned}
$$

where $s(r^*, p, d) = d^{\max\{0, \frac{1}{r^*} - \frac{1}{p}\}}$, $B_{p*} = \sqrt{2\log(2d)}$ if $p = 1$ (Mohri et al., 2018), $B_{p*} = \sqrt{2}[\frac{\Gamma(\frac{1+p^*}{2})}{\sqrt{\pi}}]^{\frac{1}{p^*}}$ if $p \in (1, 2]$, $B_{p*} = 1$ if $p \in [2, +\infty)$ (Awasthi et al., 2020), the second inequality follows from Lemma A.1, and the last inequality is due to the Rademacher variables (El-Yaniv & Pechyony, 2009).

Note that $Z$ is a deterministic function of the i.i.d. random variables $\sigma_1, \ldots, \sigma_n$, and satisfies

$$
\begin{aligned}
Z(\sigma_1, \ldots, \sigma_i, \ldots, \sigma_n) - Z(\sigma_1, \ldots, -\sigma_i, \ldots, \sigma_n) \leq& 2M \sup_{j \in [n]} \|\widehat{\boldsymbol{x}}_j\|_{p^*} \\
\leq& 2M \sup_{j \in [n]} \|\boldsymbol{x}_j + \widehat{\boldsymbol{x}}_j - \boldsymbol{x}_j\|_{p^*} \\
\leq& 2M \sup_{j \in [n]} \|\boldsymbol{x}_j\|_{p^*} + \sup_{j \in [n]} \|\widehat{\boldsymbol{x}}_j - \boldsymbol{x}_j\|_{p^*} \\
\leq& 2M \sup_{j \in [n]} \|\boldsymbol{x}_j\|_{p^*} + \varepsilon s(r^*, p, d),
\end{aligned}
$$

where the last inequality follows from Lemma A.1. This means that $Z$ is sub-Gaussian satisfying a bounded-difference condition with a variance factor

$$\upsilon = \frac{1}{4} \sum_{i=1}^n (2MR)^2 = nM^2 R^2,$$

where $R = \sup_{j \in [n]} \|\boldsymbol{x}_j\|_{p^*} + \varepsilon s(r^*, p, d)$, and satisfies

$$\frac{1}{\lambda} \log \{ \mathbb{E} \exp \lambda (Z - \mathbb{E}Z) \} \leq \frac{1}{\lambda} \frac{\lambda^2 n M^2 R^2}{2} = \frac{\lambda n M^2 R^2}{2}.$$

Letting $\lambda = \frac{\sqrt{2L \log(2)}}{MR\sqrt{n}}$ and combining the above, the inequality (9) can be upper bounded as follows:

$$
\begin{aligned}
Q \frac{1}{\lambda} \log \Big\{ 2^L \mathbb{E} \exp \lambda Z \Big\} \leq& Q(\mathbb{E}Z + \sqrt{2\log(2)LnMR}) \\
\leq& Q_{m,n} M (\sqrt{2\log(2)L} + 1)(B_{p^*} \|\boldsymbol{X}\|_{2,p^*} + \varepsilon s(r^*, p, d)) \\
=& Q_{m,n} \|g(\boldsymbol{A})\|_\infty^L \omega^L (\sqrt{2\log(2)L} + 1)(B_{p^*} \|\boldsymbol{X}\|_{2,p^*} + \varepsilon s(r^*, p, d))
\end{aligned}
$$

where $Q_{m,n} = \sqrt{\frac{2n}{m(n-m)}}$, $s(r^*, p, d) = d^{\max\{0, \frac{1}{r^*} - \frac{1}{p}\}}$, and $B_{p*} = \sqrt{2\log(2d)}$ if $p = 1$, $B_{p*} = \sqrt{2}[\frac{\Gamma(\frac{1+p^*}{2})}{\sqrt{\pi}}]^{\frac{1}{p^*}}$ if $p \in (1, 2]$, $B_{p*} = 1$ if $p \in [2, +\infty)$. Combining inequalities (7) and (9), the proof is completed.

$\square$

## C   Proofs of Lemma 4.5 and Theorem 4.7 [Multi-class Classification]

*Proof of Lemma 4.5.* Let the function class $\mathcal{F} : \mathbb{R}^{n \times d} \to \mathbb{R}^{n \times K}$ be defined in (1). Consider the activation function of the output layer as $\phi(t) = \mathrm{Softmax}(\cdot)$. Let the pairwise class margin of node $i$ be defined as $f^{uv}(\boldsymbol{A}, \boldsymbol{X})_i = [f(\boldsymbol{A}, \boldsymbol{X})_i]_u - [f(\boldsymbol{A}, \boldsymbol{X})_i]_v$, where $[f(\boldsymbol{A}, \boldsymbol{X})_i]_u$ denotes prediction of the class $u$ for node $i$. We would like to observe the relative change in error between any two classes. Specifically, we consider the difference between the set of pairwise margin $f^{uv}(\boldsymbol{A}, \widetilde{\boldsymbol{X}})_i - f^{uv}(\boldsymbol{A}, \boldsymbol{X})_i$. Define $\boldsymbol{H}^{(l)}$ and $\widetilde{\boldsymbol{H}}^{(l)}$ as the feature representation of $\boldsymbol{X}$ and $\widetilde{\boldsymbol{X}}$ at $l$-th layer, respectively. Then,

$$
\begin{aligned}
&f^{uv}(\boldsymbol{A}, \widetilde{\boldsymbol{X}})_i - f^{uv}(\boldsymbol{A}, \boldsymbol{X})_i \\
&= \sum_{j \in [n]} g(\boldsymbol{A})_{i,j} \widetilde{\boldsymbol{H}}_{j*}^{(L-1)} \big(\boldsymbol{W}_{*u}^{(L)} - \boldsymbol{W}_{*v}^{(L)}\big) - \sum_{j \in [n]} g(\boldsymbol{A})_{i,j} \boldsymbol{H}_{j*}^{(L-1)} \big(\boldsymbol{W}_{*u}^{(L)} - \boldsymbol{W}_{*v}^{(L)}\big) \\
&\leq \|g(\boldsymbol{A})\|_\infty \|\boldsymbol{W}_{*u}^{(L)} - \boldsymbol{W}_{*v}^{(L)}\|_1 \max_{j \in [n]} \left\| \phi\Big( \sum_{k \in [n]} g(\boldsymbol{A})_{j,k} \big(\widetilde{\boldsymbol{H}}_{k*}^{(L-2)} - \boldsymbol{H}_{k*}^{(L-2)}\big) \boldsymbol{W}^{(L-1)} \Big) \right\|_\infty \\
&\leq \|g(\boldsymbol{A})\|_\infty \|\boldsymbol{W}_{*u}^{(L)} - \boldsymbol{W}_{*v}^{(L)}\|_1 \max_{t \in [n]} \left\| \big( \max_{j \in [n]} \sum_{k \in [n]} g(\boldsymbol{A})_{j,k} \big) \big(\widetilde{\boldsymbol{H}}_{t*}^{(L-2)} - \boldsymbol{H}_{t*}^{(L-2)}\big) \boldsymbol{W}^{(L-1)} \right\|_\infty \\
&\leq \|g(\boldsymbol{A})\|_\infty^2 \|\boldsymbol{W}_{*u}^{(L)} - \boldsymbol{W}_{*v}^{(L)}\|_1 \|\boldsymbol{W}^{(L-1)}\|_2 \max_{j \in [n]} \|\widetilde{\boldsymbol{H}}_{j*}^{(L-2)} - \boldsymbol{H}_{j*}^{(L-2)}\| \\
&\leq \|g(\boldsymbol{A})\|_\infty^L \|\boldsymbol{W}_{*u}^{(L)} - \boldsymbol{W}_{*v}^{(L)}\|_1 \|\boldsymbol{W}^{(L-1)}\|_2 \cdots \|\boldsymbol{W}^{(2)}\|_2 \max_{j \in [n]} \left\| \big(\widetilde{\boldsymbol{X}}_{j*} - \boldsymbol{X}_{j*}\big) \boldsymbol{W}^{(1)} \right\| \\
&\leq \|g(\boldsymbol{A})\|_\infty^L \|\boldsymbol{W}_{*u}^{(L)} - \boldsymbol{W}_{*v}^{(L)}\|_1 \|\boldsymbol{W}^{(L-1)}\|_2 \cdots \|\boldsymbol{W}^{(2)}\|_2 \|\boldsymbol{W}^{(1)}\|_p \max_{j \in [n]} \|\widetilde{\boldsymbol{x}}_j - \boldsymbol{x}_j\|_{p*} \\
&\leq \|g(\boldsymbol{A})\|_\infty^L \|\boldsymbol{W}_{*u}^{(L)} - \boldsymbol{W}_{*v}^{(L)}\|_1 \prod_{l=2}^{L-1} \|\boldsymbol{W}^{(l)}\|_2 \|\boldsymbol{W}^{(1)}\|_p s(r^*, p, d) \varepsilon. \quad (10)
\end{aligned}
$$

According the definition of the ramp loss and the inequality above, we have

$$
\begin{aligned}
&\min_{\widetilde{\boldsymbol{X}} \in \mathcal{B}_r^\varepsilon(\boldsymbol{X})} \mathbb{1}(y_i \neq \arg \max_{y_i' \in [K]} [f(\boldsymbol{A}, \widetilde{\boldsymbol{X}})_i]_{y_i'}) \\
&\overset{(a)}{\leq} \ell_\gamma \big( \min_{\widetilde{\boldsymbol{X}} \in \mathcal{B}_r^\varepsilon(\boldsymbol{X})} M(f(\boldsymbol{A}, \widetilde{\boldsymbol{X}})_i, y_i) \big) \\
&\overset{(b)}{\leq} \ell_\gamma \big( \min_{y_i' \neq y_i} \min_{\widetilde{\boldsymbol{X}} \in \mathcal{B}_r^\varepsilon(\boldsymbol{X})} [f(\boldsymbol{A}, \widetilde{\boldsymbol{X}})_i]_{y_i} - [f(\boldsymbol{A}, \widetilde{\boldsymbol{X}})_i]_{y_i'} \big) \\
&\overset{(c)}{\leq} \ell_\gamma \big( \min_{y_i' \neq y_i} [f(\boldsymbol{A}, \boldsymbol{X})_i]_{y_i} - [f(\boldsymbol{A}, \boldsymbol{X})_i]_{y_i'} \\
&\qquad - \max_{y_i' \neq y_i} \varepsilon s(r^*, p, d) \|g(\boldsymbol{A})\|_\infty^L \|\boldsymbol{W}_{*u}^{(L)} - \boldsymbol{W}_{*v}^{(L)}\|_1 \prod_{l=2}^{L-1} \|\boldsymbol{W}^{(l)}\|_2 \|\boldsymbol{W}^{(1)}\|_p \big) \\
&\overset{(d)}{\leq} \ell_\gamma \big( M(f(\boldsymbol{A}, \boldsymbol{X})_i, y_i) - 2 \max_{k \in [K]} \varepsilon s(r^*, p, d) \|g(\boldsymbol{A})\|_\infty^L \|\boldsymbol{W}_{*k}^{(L)}\|_1 \prod_{l=2}^{L-1} \|\boldsymbol{W}^{(l)}\|_2 \|\boldsymbol{W}^{(1)}\|_p \big) \\
&\overset{(e)}{\leq} \mathbb{1} \big( M(f(\boldsymbol{A}, \boldsymbol{X})_i, y_i) - 2 \max_{k \in [K]} \varepsilon s(r^*, p, d) \|g(\boldsymbol{A})\|_\infty^L \|\boldsymbol{W}_{*k}^{(L)}\|_1 \prod_{l=2}^{L-1} \|\boldsymbol{W}^{(l)}\|_2 \|\boldsymbol{W}^{(1)}\|_p \leq \gamma \big),
\end{aligned}
$$

where $s(r^*, p, d) = d^{\max\{0, \frac{1}{r^*} - \frac{1}{p}\}}$, the inequality (a) is due to the property of ramp loss, the inequality (b) is due to the definition of margin operator, the inequality (c) follows from inequality (10), the inequality (d) comes from using triangle inequality, and the inequality (e) directly follows from property of ramp loss. This completes the proof of Lemma 4.5. $\qquad\square$

*Proof of Theorem 4.7.* By the Ledoux-Talagrand contraction inequality, we know that

$$\mathfrak{R}_{m,n}(\widehat{\ell} \circ \mathcal{F}) \leq \frac{1}{\gamma}\big(\mathfrak{R}_{m,n}(M \circ \mathcal{F}) + \mathfrak{R}_{m,n}(\Psi \circ \mathcal{F})\big). \tag{11}$$

For the right-hand side of the above inequality, $\mathfrak{R}_{m,n}(\Psi \circ \mathcal{F})$ can be bounded by

$$2\varepsilon s(r^*, p, d)\|g(\boldsymbol{A})\|_\infty^L \sup_{\|\boldsymbol{W}^{(l)}\|_2, \|\boldsymbol{W}^{(l)}\|_p \leq \omega, l \in [L]} \max_{k \in [K]} \|\boldsymbol{W}_{*k}^{(L)}\|_1 \prod_{l=2}^{L-1} \|\boldsymbol{W}^{(l)}\|_2 \|\boldsymbol{W}^{(1)}\|_p$$

$$\times Q\mathbb{E}_{\boldsymbol{\sigma}}\Big|\sum_{i=1}^n \sigma_i\Big|$$

$$\leq 2Q_{m,n}\varepsilon s(r^*, p, d)\|g(\boldsymbol{A})\|_\infty^L \sup_{\|\boldsymbol{W}^{(l)}\|_2, \|\boldsymbol{W}^{(l)}\|_p \leq \omega, l \in [L]} K\|\boldsymbol{W}^{(L)}\|_1 \prod_{l=2}^{L-1} \|\boldsymbol{W}^{(l)}\|_2 \|\boldsymbol{W}^{(1)}\|_p$$

$$\leq 2KQ_{m,n}\|g(\boldsymbol{A})\|_\infty^L \omega^L \varepsilon s(r^*, p, d). \tag{12}$$

where $Q_{m,n} = \sqrt{\frac{2n}{m(n-m)}}$. We turn to prove the upper bound on $\mathfrak{R}_{m,n}(M \circ \mathcal{F})$. Analyzing analogously to the proof of Theorem 4.1, $\mathfrak{R}_{m,n}(M \circ \mathcal{F})$ has the following upper bound

$$Q\mathbb{E}_{\boldsymbol{\sigma}} \sup_{\|\boldsymbol{W}^{(L)}\|_2 \leq \omega} \Big[ \sum_{i=1}^n \sigma_i\Big( \sum_{j \in [n]} g(\boldsymbol{A})_{i,j} \boldsymbol{H}_{j*}^{(L-1)} \boldsymbol{W}_{*y_i}^{(L)} \Big) \Big]$$

$$\leq Q\mathbb{E}_{\boldsymbol{\sigma}} \sup_{\|\boldsymbol{W}^{(L)}\|_2 \leq \omega} \Big[ \sum_{i=1}^n \sigma_i\big( \max_{i \in [n]} \sum_{j \in [n]} g(\boldsymbol{A})_{i,j} \big) \max_{j \in [n]} \Big\langle \boldsymbol{H}_{j*}^{(L-1)}, \boldsymbol{W}_{*y_i}^{(L)} \Big\rangle \Big]$$

$$\leq Q\frac{1}{\lambda} \log \mathbb{E}_{\boldsymbol{\sigma}} \sup_{\|\boldsymbol{W}^{(L)}\|_2 \leq \omega, j \in [n]} \exp\Big( \lambda\|g(\boldsymbol{A})\|_\infty \Big( \sum_{i=1}^n \sigma_i \Big\langle \boldsymbol{H}_{j*}^{(L-1)}, \boldsymbol{W}_{*y_i}^{(L)} \Big\rangle \Big) \Big)$$

$$\leq Q\frac{1}{\lambda} \log \mathbb{E}_{\boldsymbol{\sigma}} \sup_{j \in [n]} \exp\Big( \lambda\|g(\boldsymbol{A})\|_\infty \omega \Big\| \sum_{i=1}^n \sigma_i \boldsymbol{H}_{j*}^{(L-1)} \Big\| \Big)$$

$$= Q\frac{1}{\lambda} \log \mathbb{E}_{\boldsymbol{\sigma}} \sup_{\|\boldsymbol{W}^{(L-1)}\|_2 \leq \omega, j \in [n]} \exp\Big( \lambda\|g(\boldsymbol{A})\|_\infty \omega \Big\| \sum_{i=1}^n \sigma_i \phi\Big( \sum_{k \in [n]} g(\boldsymbol{A})_{j,k} \boldsymbol{H}_{k*}^{(L-2)} \boldsymbol{W}^{(L-1)} \Big) \Big\| \Big)$$

$$\leq Q\frac{1}{\lambda} \log \Big( 2\mathbb{E}_{\boldsymbol{\sigma}} \sup_{j \in [n]} \exp\Big( \lambda\|g(\boldsymbol{A})\|_\infty^2 \omega^2 \Big\| \sum_{i=1}^n \sigma_i \boldsymbol{H}_{j*}^{(L-2)} \Big\| \Big) \Big).$$

Repeating the process, having

$$Q\frac{1}{\lambda} \log \Big( 2^L \mathbb{E}_{\boldsymbol{\sigma}} \sup_{\|\boldsymbol{W}^{(1)}\|_p \leq \omega, j \in [n]} \exp\Big( \lambda\|g(\boldsymbol{A})\|_\infty^L \omega^{L-1} \Big\| \sum_{i=1}^n \sigma_i \boldsymbol{X}_{j*}, \boldsymbol{W}^{(1)} \Big\| \Big) \Big)$$

$$\leq Q\frac{1}{\lambda} \log \Big( 2^L \mathbb{E}_{\boldsymbol{\sigma}} \sup_{j \in [n]} \exp\Big( \lambda\|g(\boldsymbol{A})\|_\infty^L \omega^L \Big\| \sum_{i=1}^n \sigma_i \boldsymbol{x}_j \Big\|_{p*} \Big) \Big)$$

$$\leq Q\frac{1}{\lambda} \log \Big( 2^L \mathbb{E}_{\boldsymbol{\sigma}} \exp\Big( \lambda\|g(\boldsymbol{A})\|_\infty^L \omega^L \Big\| \sum_{i=1}^n \sigma_i \boldsymbol{x}_i \Big\|_{p*} \Big) \Big). \tag{13}$$

Denote $M = \|g(\boldsymbol{A})\|_\infty^L \omega^L$ and define the random function of the random variables $\sigma_1, \ldots, \sigma_n$ as follows

$$Z = M \cdot \Big\| \sum_{i=1}^n \sigma_i \boldsymbol{x}_i \Big\|_{p*}.$$

Then,

$$\frac{1}{\lambda}\log\left\{2^L \mathbb{E}\exp\lambda Z\right\} = \frac{L\log(2)}{\lambda} + \frac{1}{\lambda}\log\{\mathbb{E}\exp\lambda(Z-\mathbb{E}Z)\} + \mathbb{E}Z.$$

According to well-known bounds on the Rademacher complexity (Haagerup, 1981; Mohri et al., 2018; Awasthi et al., 2020), having

$$\mathbb{E}_{\boldsymbol{\sigma}}\Big\|\sum_{i=1}^n \sigma_i \boldsymbol{x}_i\Big\|_{p^*} \leq \begin{cases} \sqrt{2\log(2d)}\|\boldsymbol{X}\|_{2,p^*} & if \quad p=1 \\ \sqrt{2}[\frac{\Gamma(\frac{1+p^*}{2})}{\sqrt{\pi}}]^{\frac{1}{p^*}}\|\boldsymbol{X}\|_{2,p^*} & if \quad 1 < p \leq 2 \\ \|\boldsymbol{X}\|_{2,p^*} & if \quad p \geq 2 \end{cases}$$

where $\boldsymbol{X} = (\boldsymbol{x}_1, \ldots, \boldsymbol{x}_n) \in \mathbb{R}^{n\times d}$. We thus have

$$\mathbb{E}Z = M\cdot\mathbb{E}_{\boldsymbol{\sigma}}\Big\|\sum_{i=1}^n \sigma_i \boldsymbol{x}_i\Big\|_{p^*} \leq MB_{p^*}\|\boldsymbol{X}\|_{2,p^*} \tag{14}$$

where $B_{p*} = \sqrt{2\log(2d)}$ if $p=1$; $B_{p*} = \sqrt{2}[\frac{\Gamma(\frac{1+p^*}{2})}{\sqrt{\pi}}]^{\frac{1}{p^*}}$ if $p \in (1,2]$; $B_{p*} = 1$ if $p \in [2,+\infty)$ (Mohri et al., 2018; Awasthi et al., 2020). Since $Z$ is a deterministic function of $\sigma_1, \ldots, \sigma_n$, and satisfies

$$Z(\sigma_1, \ldots, \sigma_i, \ldots, \sigma_n) - Z(\sigma_1, \ldots, -\sigma_i, \ldots, \sigma_n) \leq 2M\|\boldsymbol{x}_i\|_{p^*}, \tag{15}$$

then $Z$ satisfies a bounded-difference property and is sub-Gaussian with the variance factor

$$\upsilon = \frac{1}{4}\sum_{i=1}^n (2M\|\boldsymbol{x}_i\|_{p^*})^2 = M^2\sum_{i=1}^n \|\boldsymbol{x}_i\|_{p^*}^2,$$

and satisfies

$$\frac{1}{\lambda}\log\{\mathbb{E}\exp\lambda(Z-\mathbb{E}Z)\} \leq \frac{1}{\lambda}\frac{\lambda^2 M^2\sum_{i=1}^n \|\boldsymbol{x}_i\|_{p^*}^2}{2} = \frac{\lambda M^2\sum_{i=1}^n \|\boldsymbol{x}_i\|_{p^*}^2}{2}.$$

Letting $\lambda = \frac{\sqrt{2L\log(2)}}{M\sqrt{\sum_{i=1}^n \|\boldsymbol{x}_i\|_{p^*}^2}}$ and with the above, the inequality (13) can be upper bounded by

$$Q\frac{1}{\lambda}\log\left\{2^L\mathbb{E}\exp\lambda Z\right\} \leq Q\Big(\mathbb{E}Z + \sqrt{2\log(2)L}M\sqrt{\sum_{i=1}^n \|\boldsymbol{x}_i\|_{p^*}^2}\Big)$$

$$\leq Q_{m,n}M(\sqrt{2\log(2)L}+1)B_{p^*}\|\boldsymbol{X}\|_{2,p^*}$$

$$= Q_{m,n}\|g(\boldsymbol{A})\|_\infty^L \omega^L(\sqrt{2\log(2)L}+1)B_{p^*}\|\boldsymbol{X}\|_{2,p^*}$$

where $Q = \frac{1}{m} + \frac{1}{u}$ and $Q_{m,n} = \sqrt{\frac{2n}{m(n-m)}}$. Combining the above, we obtain

$$\mathfrak{R}_{m,n}(M\circ\mathcal{F}) \leq Q_{m,n}\|g(\boldsymbol{A})\|_\infty^L \omega^L(\sqrt{2\log(2)L}+1)B_{p^*}\|\boldsymbol{X}\|_{2,p^*}. \tag{16}$$

Putting inequalities (12) and (16) backs into (11), this completes the proof. $\qquad\square$

## D  PROOF OF PROPOSITION 5.1 [SGC]

**Lemma D.1** (SGC). *Let the robust surrogate loss be defined by*

$$\widehat{\ell}(f(\boldsymbol{A},\boldsymbol{X})_i, y_i) = \ell_\gamma(M(f(\boldsymbol{A},\boldsymbol{X})_i, y_i) - \Psi(f(\boldsymbol{A},\boldsymbol{X})_i)),$$

*where the worst-case error is*

$$\Psi(f(\boldsymbol{A},\widetilde{\boldsymbol{X}})_i) = 2\max_{k\in[K]}\varepsilon s(r^*,p,d)\|g(\boldsymbol{A})^L\|_\infty\|\boldsymbol{W}_{*k}^{(L)}\|_1\prod_{l=1}^{L-1}\|\boldsymbol{W}^{(l)}\|_p,$$

*and $s(r^*,p,d) = d^{\max\{0,\frac{1}{r^*}-\frac{1}{p}\}}$. Then, we have*

$$\widetilde{\ell}(f(\boldsymbol{A},\boldsymbol{X})_i, y_i) \leq \widehat{\ell}(f(\boldsymbol{A},\boldsymbol{X})_i, y_i) \leq \mathbb{1}\{M(f(\boldsymbol{A},\boldsymbol{X})_i, y_i) - \Psi(f(\boldsymbol{A},\widetilde{\boldsymbol{X}})_i) \leq \gamma\}.$$

*Proof of Lemma D.1.* Analyzing analogously to the proof of Lemma 4.5. Consider the pairwise class margin of node $i$ be defined as $f^{uv}(\boldsymbol{A}, \boldsymbol{X})_i = [f(\boldsymbol{A}, \boldsymbol{X})_i]_u - [f(\boldsymbol{A}, \boldsymbol{X})_i]_v$, we then have

$$
\begin{aligned}
&f^{uv}(\boldsymbol{A}, \widetilde{\boldsymbol{X}})_i - f^{uv}(\boldsymbol{A}, \boldsymbol{X})_i \\
=& \Big( \sum_{j \in [n]} g(\boldsymbol{A})_{i,j}^L \widetilde{\boldsymbol{X}}_{j*} \boldsymbol{W}^{(1)} \cdots \big( \boldsymbol{W}_{*u}^{(L)} - \boldsymbol{W}_{*v}^{(L)} \big) \Big) - \Big( \sum_{j \in [n]} g(\boldsymbol{A})_{i,j}^L \boldsymbol{X}_{j*} \boldsymbol{W}^{(1)} \cdots \big( \boldsymbol{W}_{*u}^{(L)} - \boldsymbol{W}_{*v}^{(L)} \big) \Big) \\
\leq& \Big( \max_{i \in [n]} \sum_{j \in [n]} g(\boldsymbol{A})_{ij}^L \Big) \max_{j \in [n]} \Big\langle (\widetilde{\boldsymbol{X}}_{j*} - \boldsymbol{X}_{j*}), \boldsymbol{W}^{(1)} \cdots \big( \boldsymbol{W}_{*u}^{(L)} - \boldsymbol{W}_{*v}^{(L)} \big) \Big\rangle \\
\leq& \|g(\boldsymbol{A})^L\|_\infty \|\boldsymbol{W}^{(1)} \cdots \big( \boldsymbol{W}_{*u}^{(L)} - \boldsymbol{W}_{*v}^{(L)} \big)\|_p \max_{j \in [n]} \|\widetilde{\boldsymbol{x}}_j - \boldsymbol{x}_j\|_{p*} \\
\leq& \|g(\boldsymbol{A})^L\|_\infty \|\boldsymbol{W}^{(1)} \cdots \big( \boldsymbol{W}_{*u}^{(L)} - \boldsymbol{W}_{*v}^{(L)} \big)\|_p s(r^*, p, d) \max_{j \in [n]} \|\widetilde{\boldsymbol{x}}_j - \boldsymbol{x}_j\|_r \\
\leq& \|g(\boldsymbol{A})^L\|_\infty \|\boldsymbol{W}_{*u}^{(L)} - \boldsymbol{W}_{*v}^{(L)}\|_1 \prod_{l=1}^{L-1} \|\boldsymbol{W}^{(l)}\|_p s(r^*, p, d)\varepsilon, \quad\quad (17)
\end{aligned}
$$

where the third inequality follows from Lemma A.2. According to the property of ramp loss, we have

$$
\begin{aligned}
&\ell_\gamma \Big( \min_{\widetilde{\boldsymbol{X}} \in \mathcal{B}_r^\varepsilon(\boldsymbol{X})} M(f(\boldsymbol{A}, \widetilde{\boldsymbol{X}})_i, y_i) \Big) \\
\leq& \ell_\gamma \Big( \min_{y_i' \neq y_i} \min_{\widetilde{\boldsymbol{X}} \in \mathcal{B}_r^\varepsilon(\boldsymbol{X})} [f(\boldsymbol{A}, \widetilde{\boldsymbol{X}})_i]_{y_i} - [f(\boldsymbol{A}, \widetilde{\boldsymbol{X}})_i]_{y_i'} \Big) \\
\leq& \ell_\gamma \Big( \min_{y_i' \neq y_i} [f(\boldsymbol{A}, \boldsymbol{X})_i]_{y_i} - [f(\boldsymbol{A}, \boldsymbol{X})_i]_{y_i'} \\
&\quad - \max_{y_i' \neq y_i} \varepsilon s(r^*, p, d)\|g(\boldsymbol{A})^L\|_\infty \|\boldsymbol{W}_{*u}^{(L)} - \boldsymbol{W}_{*v}^{(L)}\|_1 \prod_{l=1}^{L-1} \|\boldsymbol{W}^{(l)}\|_p \Big) \\
\leq& \ell_\gamma \Big( M(f(\boldsymbol{A}, \boldsymbol{X})_i, y_i) - 2 \max_{k \in [K]} \varepsilon s(r^*, p, d)\|g(\boldsymbol{A})^L\|_\infty \|\boldsymbol{W}_{*k}^{(L)}\|_1 \prod_{l=1}^{L-1} \|\boldsymbol{W}^{(l)}\|_p \Big) \\
\leq& \mathbb{1} \Big( M(f(\boldsymbol{A}, \boldsymbol{X})_i, y_i) - 2 \max_{k \in [K]} \varepsilon s(r^*, p, d)\|g(\boldsymbol{A})^L\|_\infty \|\boldsymbol{W}_{*k}^{(L)}\|_1 \prod_{l=1}^{L-1} \|\boldsymbol{W}^{(l)}\|_p \leq \gamma \Big).
\end{aligned}
$$

$\square$

**Theorem D.2** (restate Proposition 5.1). *For any $\gamma > 0$, with probability at least $1 - \delta$, we have for all $f \in \mathcal{F}$,*

$$
\begin{aligned}
&\frac{1}{n-m} \sum_{i=m+1}^n \mathbb{1}\{\exists \widetilde{\boldsymbol{X}} \in \mathcal{B}_r^\varepsilon(\boldsymbol{X}) \; s.t. \; y_i \neq \arg \max_{y' \in [K]} [f(\boldsymbol{A}, \widetilde{\boldsymbol{X}})_i]_{y'}\} \\
\leq& \frac{1}{m} \sum_{i=1}^m \mathbb{1}\{[f(\boldsymbol{A}, \boldsymbol{X})_i]_{y_i'} \leq \gamma + \max_{y_i' \neq y'} [f(\boldsymbol{A}, \boldsymbol{X})_i]_{y'} + \Psi(f(\boldsymbol{A}, \widetilde{\boldsymbol{X}})_i)\} \\
&+ Q_{m,n} \frac{2K}{\gamma} \|g(\boldsymbol{A})^L\|_\infty \omega^L (B_{p*}\|\boldsymbol{X}\|_{2,p*} + \varepsilon s(r^*, p, d)) + \mathcal{O}(\max\{\frac{1}{\sqrt{m}}, \frac{1}{\sqrt{n-m}}\}),
\end{aligned}
$$

*where $s(r^*, p, d) = d^{\max\{0, \frac{1}{r^*} - \frac{1}{p}\}}$, $B_{p*} = \sqrt{2 \log(2d)}$ if $p = 1$, $B_{p*} = \sqrt{2}[\frac{\Gamma(\frac{1+p^*}{2})}{\sqrt{\pi}}]^{\frac{1}{p*}}$ if $p \in (1, 2]$, $B_{p*} = 1$ if $p \in [2, +\infty)$, $Q_{m,n} = \sqrt{\frac{2n}{m(n-m)}}$.*

*Proof of Theorem D.2.* Let the hypothesis class of SGC be defined by

$$
\mathcal{F} = \Big\{ f(\boldsymbol{A}, \boldsymbol{X}) = g(\boldsymbol{A})^L \boldsymbol{X} \boldsymbol{W}^{(1)} \cdots \boldsymbol{W}^{(L)} : \|\boldsymbol{W}^{(l)}\|_2, \|\boldsymbol{W}^{(l)}\|_p \leq \omega, l \in [L] \Big\} \quad (18)
$$

where $\boldsymbol{W}^{(l)} \in \mathbb{R}^{d_{l-1} \times d_l}$, and $d_l$ is the width of $l$-th layer with $d_L = K$ and $d_0 = d$. According to Lemma D.1 and the Ledoux-Talagrand contraction inequality, we have

$$
\mathfrak{R}_{m,n}(\widehat{\ell} \circ \mathcal{F}) \leq \frac{1}{\gamma} \big( \mathfrak{R}_{m,n}(M \circ \mathcal{F}) + \mathfrak{R}_{m,n}(\Psi \circ \mathcal{F}) \big). \quad (19)
$$

Then, $\mathfrak{R}_{m,n}(M \circ \mathcal{F})$ can be bounded by

$$Q\mathbb{E}_{\boldsymbol{\sigma}} \sup_{\|\boldsymbol{W}^{(l)}\|_p \leq \omega, l=1,\ldots,L} \sum_{i=1}^{n} \sigma_i \Big( \max_{i\in[n]} \sum_{j\in[n]} g(\boldsymbol{A})_{i,j}^L \Big) \max_{j\in[n]} \Big\langle \boldsymbol{X}_{j*}, \boldsymbol{W}^{(1)} \cdots \boldsymbol{W}_{*y_i}^{(L)} \Big\rangle$$

$$\leq Q\|g(\boldsymbol{A})^L\|_\infty \omega^L \max_{j\in[n]} \mathbb{E}_{\boldsymbol{\sigma}} \Big\| \sum_{i=1}^{n} \sigma_i \boldsymbol{x}_j \Big\|_{p^*}$$

$$\leq Q\|g(\boldsymbol{A})^L\|_\infty \omega^L \mathbb{E}_{\boldsymbol{\sigma}} \Big\| \sum_{i=1}^{n} \sigma_i \boldsymbol{x}_i \Big\|_{p^*}$$

$$\leq Q_{m,n}\|g(\boldsymbol{A})^L\|_\infty \omega^L B_{p^*}\|\boldsymbol{X}\|_{2,p^*}, \tag{20}$$

where $Q_{m,n} = \sqrt{\frac{2n}{m(n-m)}}$, $B_{p*} = \sqrt{2\log(2d)}$ if $p = 1$; $B_{p*} = \sqrt{2}[\frac{\Gamma(\frac{1+p^*}{2})}{\sqrt{\pi}}]^{\frac{1}{p^*}}$ if $p \in (1,2]$; $B_{p*} = 1$ if $p \in [2, +\infty)$.

For the second term on the right side of the inequality (19), $\mathfrak{R}_{m,n}(\Psi \circ \mathcal{F})$ is bounded by

$$2\max_{k\in[K]} \varepsilon s(r^*,p,d)\|g(\boldsymbol{A})^L\|_\infty \sup_{\|\boldsymbol{W}^{(l)}\|_p \leq \omega, l=1,\ldots,L} \|\boldsymbol{W}_{*k}^{(L)}\|_1 \prod_{l=1}^{L-1} \|\boldsymbol{W}^{(l)}\|_p Q\mathbb{E}_{\boldsymbol{\sigma}} \Big| \sum_{i=1}^{n} \sigma_i \Big|$$

$$\leq 2K\varepsilon s(r^*,p,d)\|g(\boldsymbol{A})^L\|_\infty \omega^L Q_{m,n}. \tag{21}$$

where $Q_{m,n} = \sqrt{\frac{2n}{m(n-m)}}$. Combining Theorem 3.2 with inequality (19), we complete the proof. $\quad\square$

## E    PROOF OF PROPOSITION 5.3 [RESIDUAL GCN]

**Lemma E.1** (Residual GCN). *Let the robust surrogate loss be defined by*

$$\widehat{\ell}(f(\boldsymbol{A},\boldsymbol{X})_i, y_i) = \ell_\gamma(M(f(\boldsymbol{A},\boldsymbol{X})_i, y_i) - \Psi(f(\boldsymbol{A},\boldsymbol{X})_i)),$$

*where the worst-case error is*

$$\Psi(f(\boldsymbol{A},\widetilde{\boldsymbol{X}})_i) = 2\max_{k\in[K]} \varepsilon s(r^*,p,d)\|g(\boldsymbol{A})\|_\infty^L \|\boldsymbol{W}_{*k}^{(L)}\|_1 \prod_{l=1}^{L-1} (\|\boldsymbol{W}^{(l)}\|_2 + 1)\|\boldsymbol{W}^{(0)}\|_p.$$

*where $s(r^*,p,d) = d^{\max\{0, \frac{1}{r^*} - \frac{1}{p}\}}$. Then, we have*

$$\widetilde{\ell}(f(\boldsymbol{A},\boldsymbol{X})_i, y_i) \leq \widehat{\ell}(f(\boldsymbol{A},\boldsymbol{X})_i, y_i) \leq \mathbb{1}\{M(f(\boldsymbol{A},\boldsymbol{X})_i, y_i) - \Psi(f(\boldsymbol{A},\widetilde{\boldsymbol{X}})_i) \leq \gamma\}.$$

*Proof of Lemma E.1.* Recall the output of Residual GCNs: $f(\boldsymbol{A},\boldsymbol{X}) = \text{Softmax}(\boldsymbol{H}^{(L-1)}\boldsymbol{W}^{(L)})$ with the update rule

$$\boldsymbol{H}^{(l)} = \phi(g(\boldsymbol{A})\boldsymbol{H}^{(l-1)}\boldsymbol{W}^{(l)}) + \boldsymbol{H}^{(l-1)}, \quad \boldsymbol{H}^{(0)} = \boldsymbol{X}\boldsymbol{W}^{(0)}$$

where $\boldsymbol{W}^{(0)} \in \mathbb{R}^{d\times d'}$, $\boldsymbol{W}^{(L)} \in \mathbb{R}^{d'\times K}$, $\boldsymbol{W}^{(l)} \in \mathbb{R}^{d'\times d'}$ for $l = 1,\ldots,L-1$.

Let $\boldsymbol{H}^{(l)}$ and $\widetilde{\boldsymbol{H}}^{(l)}$ denote the feature representation of $\boldsymbol{X}$ and $\widetilde{\boldsymbol{X}}$ at $l$-th layer. We first analyze the difference between set of pairwise margin $f^{uv}(\boldsymbol{A},\widetilde{\boldsymbol{X}})_i - f^{uv}(\boldsymbol{A},\boldsymbol{X})_i$ for node $i$, where $f^{uv}(\boldsymbol{A},\boldsymbol{X})_i = [f(\boldsymbol{A},\boldsymbol{X})_i]_u - [f(\boldsymbol{A},\boldsymbol{X})_i]_v$, having

$$\Big(\phi\Big( \sum_{j\in[n]} g(\boldsymbol{A})_{i,j}\widetilde{\boldsymbol{H}}_{j*}^{(L-2)}\boldsymbol{W}^{(L-1)} \Big) + \widetilde{\boldsymbol{H}}_{j*}^{(L-2)}\Big)\Big(\boldsymbol{W}_{*u}^{(L)} - \boldsymbol{W}_{*v}^{(L)}\Big) -$$

$$\Big(\phi\Big( \sum_{j\in[n]} g(\boldsymbol{A})_{i,j}\boldsymbol{H}_{j*}^{(L-2)}\boldsymbol{W}^{(L-1)} \Big) + \boldsymbol{H}_{j*}^{(L-2)}\Big)\Big(\boldsymbol{W}_{*u}^{(L)} - \boldsymbol{W}_{*v}^{(L)}\Big)$$

$$\leq \Big( \sum_{j\in[n]} g(\boldsymbol{A})_{i,j}(\widetilde{\boldsymbol{H}}_{j*}^{(L-2)} - \boldsymbol{H}_{j*}^{(L-2)})(\boldsymbol{W}^{(L-1)} + \boldsymbol{I}_n)\Big)\Big(\boldsymbol{W}_{*u}^{(L)} - \boldsymbol{W}_{*v}^{(L)}\Big)$$

$$\leq \Big( \max_{i\in[n]} \sum_{j\in[n]} g(\boldsymbol{A})_{i,j}\Big) \max_{j\in[n]} \Big\langle (\widetilde{\boldsymbol{H}}_{j*}^{(L-2)} - \boldsymbol{H}_{j*}^{(L-2)})(\boldsymbol{W}^{(L-1)} + \boldsymbol{I}_n), \boldsymbol{W}_{*u}^{(L)} - \boldsymbol{W}_{*v}^{(L)} \Big\rangle$$

$$\leq \|\boldsymbol{W}_{*u}^{(L)} - \boldsymbol{W}_{*v}^{(L)}\|_1 \|g(\boldsymbol{A})\|_\infty (\|\boldsymbol{W}^{(L-1)}\|_2 + 1) \max_{j\in[n]} \|\widetilde{\boldsymbol{H}}_{j*}^{(L-2)} - \boldsymbol{H}_{j*}^{(L-2)}\|$$

where $\boldsymbol{I}_n$ denotes the Identity matrix and the second inequality is due to the Lipschitzness of the activation function $\phi$. Applying recursive steps, we further obtain

$$\|\boldsymbol{W}_{*u}^{(L)} - \boldsymbol{W}_{*v}^{(L)}\|_1 \|g(\boldsymbol{A})\|_\infty^L \prod_{l=1}^{L-1} (\|\boldsymbol{W}^{(l)}\|_2 + 1) \max_{j\in[n]} \left\|(\widetilde{\boldsymbol{X}}_{j*} - \boldsymbol{X}_{j*})\boldsymbol{W}^{(0)}\right\|$$

$$\leq \|\boldsymbol{W}_{*u}^{(L)} - \boldsymbol{W}_{*v}^{(L)}\|_1 \|g(\boldsymbol{A})\|_\infty^L \prod_{l=1}^{L-1} (\|\boldsymbol{W}^{(l)}\|_2 + 1)\|\boldsymbol{W}^{(0)}\|_p s(r^*, p, d) \max_{j\in[n]} \|\widetilde{\boldsymbol{x}}_j - \boldsymbol{x}_j\|_r$$

$$\leq \|\boldsymbol{W}_{*u}^{(L)} - \boldsymbol{W}_{*v}^{(L)}\|_1 \|g(\boldsymbol{A})\|_\infty^L \prod_{l=1}^{L-1} (\|\boldsymbol{W}^{(l)}\|_2 + 1)\|\boldsymbol{W}^{(0)}\|_p s(r^*, p, d)\varepsilon,$$

where $s(r^*, p, d) = d^{\max\{0, \frac{1}{r^*} - \frac{1}{p}\}}$ and the second inequality follows from Lemma A.2.

By the property of ramp loss, the following inequality holds:

$$\ell_\gamma(\min_{\widetilde{\boldsymbol{X}}\in\mathcal{B}_r^\varepsilon(\boldsymbol{X})} M(f(\boldsymbol{A}, \widetilde{\boldsymbol{X}})_i, y_i))$$

$$\leq \ell_\gamma(\min_{y_i'\neq y_i} \min_{\widetilde{\boldsymbol{X}}\in\mathcal{B}_r^\varepsilon(\boldsymbol{X})} [f(\boldsymbol{A}, \widetilde{\boldsymbol{X}})_i]_{y_i} - [f(\boldsymbol{A}, \widetilde{\boldsymbol{X}})_i]_{y_i'})$$

$$\leq \ell_\gamma(\min_{y_i'\neq y_i} [f(\boldsymbol{A}, \boldsymbol{X})_i]_{y_i} - [f(\boldsymbol{A}, \boldsymbol{X})_i]_{y_i'}$$

$$- \max_{y_i'\neq y_i} \varepsilon s(r^*, p, d)\|\boldsymbol{W}_{*u}^{(L)} - \boldsymbol{W}_{*v}^{(L)}\|_1 \|g(\boldsymbol{A})\|_\infty^L \prod_{l=1}^{L-1} (\|\boldsymbol{W}^{(l)}\|_2 + 1)\|\boldsymbol{W}^{(0)}\|_p)$$

$$\leq \ell_\gamma(M(f(\boldsymbol{A}, \boldsymbol{X})_i, y_i) - 2 \max_{k\in[K]} \varepsilon s(r^*, p, d)\|g(\boldsymbol{A})\|_\infty^L \|\boldsymbol{W}_{*k}^{(L)}\|_1 \prod_{l=1}^{L-1} (\|\boldsymbol{W}^{(l)}\|_2 + 1)\|\boldsymbol{W}^{(0)}\|_p)$$

$$\leq \mathbb{1}(M(f(\boldsymbol{A}, \boldsymbol{X})_i, y_i) - 2 \max_{k\in[K]} \varepsilon s(r^*, p, d)\|g(\boldsymbol{A})\|_\infty^L \|\boldsymbol{W}_{*k}^{(L)}\|_1 \prod_{l=1}^{L-1} (\|\boldsymbol{W}^{(l)}\|_2 + 1)\|\boldsymbol{W}^{(0)}\|_p \leq \gamma).$$

$\square$

**Theorem E.2** (restate Proposition 5.3). *for any $\gamma > 0$, with probability at least $1 - \delta$, we have for all $f \in \mathcal{F}$,*

$$\frac{1}{n - m} \sum_{i=m+1}^n \mathbb{1}\{\exists \widetilde{\boldsymbol{X}} \in \mathcal{B}_r^\varepsilon(\boldsymbol{X}) \; s.t. \; y_i \neq \arg\max_{y'\in[K]} [f(\boldsymbol{A}, \widetilde{\boldsymbol{X}})_i]_{y'}\}$$

$$\leq \frac{1}{m} \sum_{i=1}^m \mathbb{1}\{[f(\boldsymbol{A}, \boldsymbol{X})_i]_{y_i'} \leq \gamma + \max_{y_i'\neq y'} [f(\boldsymbol{A}, \boldsymbol{X})_i]_{y'} + \Psi(f(\boldsymbol{A}, \widetilde{\boldsymbol{X}})_i)\} + O_{m,n}$$

$$+ Q_{m,n} \frac{4K}{\gamma} \|g(\boldsymbol{A})\|_\infty^L \omega(\omega + 1)^L (\sqrt{\log(2)L} + 1)(B_{p^*}\|\boldsymbol{X}\|_{2,p^*} + \varepsilon s(r^*, p, d)).$$

*where $O_{m,n} = \mathcal{O}(\max\{\frac{1}{\sqrt{m}}, \frac{1}{\sqrt{n-m}}\})$, $s(r^*, p, d) = d^{\max\{0, \frac{1}{r^*} - \frac{1}{p}\}}$, $B_{p*} = \sqrt{2\log(2d)}$ if $p = 1$, $B_{p*} = \sqrt{2}[\frac{\Gamma(\frac{1+p^*}{2})}{\sqrt{\pi}}]^{\frac{1}{p^*}}$ if $p \in (1, 2]$, $B_{p*} = 1$ if $p \in [2, +\infty)$, $Q_{m,n} = \sqrt{\frac{2n}{m(n-m)}}$.*

*Proof of Theorem E.2.* With Lemma E.1 and the Ledoux-Talagrand contraction inequality, we have

$$\mathfrak{R}_{m,n}(\widehat{\ell} \circ \mathcal{F}) \leq \frac{1}{\gamma}\big(\mathfrak{R}_{m,n}(M \circ \mathcal{F}) + \mathfrak{R}_{m,n}(\Psi \circ \mathcal{F})\big). \tag{22}$$

Let the hypothesis class of Residual GCNs be defined by

$$\mathcal{F} = \{f(\boldsymbol{A}, \boldsymbol{X}) = \text{Softmax}(\boldsymbol{H}^{(L-1)}\boldsymbol{W}^{(L)}) : \|\boldsymbol{W}^{(l)}\|_2, \|\boldsymbol{W}^{(l)}\|_p \leq \omega, l \in [L]\} \tag{23}$$

with the update rule

$$\boldsymbol{H}^{(l)} = \phi(g(\boldsymbol{A})\boldsymbol{H}^{(l-1)}\boldsymbol{W}^{(l)}) + \boldsymbol{H}^{(l-1)}, \quad \boldsymbol{H}^{(0)} = \boldsymbol{X}\boldsymbol{W}^{(0)}$$

where $\boldsymbol{W}^{(L)} \in \mathbb{R}^{d' \times K}$, $\boldsymbol{W}^{(0)} \in \mathbb{R}^{d \times d'}$, $\boldsymbol{W}^{(l)} \in \mathbb{R}^{d' \times d'}$ for $l = 1, \ldots, L-1$.

Applying the triangle inequality and Lemma A.2, we have the following upper bound on $\mathfrak{R}_{m,n}(M \circ \mathcal{F})$

$$Q\mathbb{E}_{\boldsymbol{\sigma}} \sup_{\|\boldsymbol{W}^{(L)}\|_2 \leq \omega} \Big[ \sum_{i=1}^{n} \sigma_i \Big\langle \boldsymbol{H}_{i*}^{(L-1)}, \boldsymbol{W}_{*y_i}^{(L)} \Big\rangle \Big]$$

$$\leq Q\omega\mathbb{E}_{\boldsymbol{\sigma}} \sup_{\|\boldsymbol{W}^{(L-1)}\|_2 \leq \omega} \Big\| \sum_{i=1}^{n} \sigma_i \Big( \phi\Big( \sum_{j \in [n]} g(\boldsymbol{A})_{i,j}\boldsymbol{H}_{j*}^{(L-2)}\boldsymbol{W}^{(L-1)} \Big) + \boldsymbol{H}_{i*}^{(L-2)} \Big) \Big\|$$

$$\leq Q\omega\mathbb{E}_{\boldsymbol{\sigma}} \sup_{\|\boldsymbol{W}^{(L-1)}\|_2 \leq \omega} \Big\| \sum_{i=1}^{n} \sigma_i \phi\Big( \sum_{j \in [n]} g(\boldsymbol{A})_{i,j}\boldsymbol{H}_{j*}^{(L-2)}\boldsymbol{W}^{(L-1)} \Big) \Big\| + \sup \Big\| \sum_{i=1}^{n} \sigma_i \boldsymbol{H}_{i*}^{(L-2)} \Big\|$$

$$\leq Q\frac{1}{\lambda} \log 2 \mathbb{E}_{\boldsymbol{\sigma}} \sup_{j \in [n]} \exp\Big( \lambda\omega(\|g(\boldsymbol{A})\|_\infty\omega + 1) \Big\| \sum_{i=1}^{n} \sigma_i \boldsymbol{H}_{j*}^{(L-2)} \Big\| \Big)$$

$$\leq Q\frac{1}{\lambda} \log \Big( 2^L \mathbb{E}_{\boldsymbol{\sigma}} \sup_{j \in [n]} \exp\Big( \lambda\omega(\|g(\boldsymbol{A})\|_\infty\omega + 1)^{L-1} \Big\| \sum_{i=1}^{n} \sigma_i \boldsymbol{H}_{j}^{(0)} \Big\|_{p*} \Big) \Big)$$

$$\leq Q\frac{1}{\lambda} \log \Big( 2^L \mathbb{E}_{\boldsymbol{\sigma}} \sup_{j \in [n]} \exp\Big( \lambda\omega^2(\|g(\boldsymbol{A})\|_\infty\omega + 1)^{L-1} \Big\| \sum_{i=1}^{n} \sigma_i \boldsymbol{X}_{j} \Big\|_{p*} \Big) \Big)$$

$$\leq Q\frac{1}{\lambda} \log \Big( 2^L \mathbb{E}_{\boldsymbol{\sigma}} \exp\Big( \lambda\omega(\|g(\boldsymbol{A})\|_\infty\omega + 1)^{L} \Big\| \sum_{i=1}^{n} \sigma_i \boldsymbol{x}_{i} \Big\|_{p*} \Big) \Big). \tag{24}$$

Let $M = \omega(\|g(\boldsymbol{A})\|_\infty\omega + 1)^L$ and consider a random variable

$$Z = M \cdot \Big\| \sum_{i=1}^{n} \sigma_i \boldsymbol{x}_i \Big\|_{p*}.$$

We then have

$$\frac{1}{\lambda} \log \Big\{ 2^L \mathbb{E} \exp \lambda Z \Big\} = \frac{L \log(2)}{\lambda} + \frac{1}{\lambda} \log\{ \mathbb{E} \exp \lambda(Z - \mathbb{E}Z) \} + \mathbb{E}Z.$$

According to the inequalities (14) and (15), we know that $\mathbb{E}Z \leq MB_{p*}\|\boldsymbol{X}\|_{2,p*}$, and $Z$ is sub-Gaussian with the variance

$$\upsilon = \frac{1}{4} \sum_{i=1}^{n} (2M\|\boldsymbol{x}_i\|_{p*})^2 = M^2 \sum_{i=1}^{n} \|\boldsymbol{x}_i\|_{p*}^2,$$

and satisfies

$$\frac{1}{\lambda} \log\{ \mathbb{E} \exp \lambda(Z - \mathbb{E}Z) \} \leq \frac{\lambda M^2 \sum_{i=1}^{n} \|\boldsymbol{x}_i\|_{p*}^2}{2}.$$

Letting $\lambda = \frac{\sqrt{2L \log(2)}}{M\sqrt{\sum_{i=1}^{n} \|\boldsymbol{x}_i\|_{p*}^2}}$, the inequality (24) is bounded by

$$Q\frac{1}{\lambda} \log \Big\{ 2^L \mathbb{E} \exp \lambda Z \Big\} \leq Q\Big( \mathbb{E}Z + \sqrt{2 \log(2)L} M \sqrt{\sum_{i=1}^{n} \|\boldsymbol{x}_i\|_{p*}^2} \Big)$$

$$\leq Q_{m,n}\omega(\|g(\boldsymbol{A})\|_\infty\omega + 1)^L (\sqrt{2 \log(2)L} + 1)B_{p*}\|\boldsymbol{X}\|_{2,p*}$$

where $Q = \frac{1}{m} + \frac{1}{u}$, $Q_{m,n} = \sqrt{\frac{2n}{m(n-m)}}$, and $B_{p*} = \sqrt{2 \log(2d)}$ if $p = 1$; $B_{p*} = \sqrt{2}[\frac{\Gamma(\frac{1+p^*}{2})}{\sqrt{\pi}}]^{\frac{1}{p^*}}$ if $p \in (1, 2]$; $B_{p*} = 1$ if $p \in [2, +\infty)$. Thus,

$$\mathfrak{R}_{m,n}(M \circ \mathcal{F}) \leq Q_{m,n}\omega(\|g(\boldsymbol{A})\|_\infty\omega + 1)^L (\sqrt{2 \log(2)L} + 1)B_{p*}\|\boldsymbol{X}\|_{2,p*}.$$

For the TRC of the worse-case, by $\|\boldsymbol{W}^{(l)}\|_2, \|\boldsymbol{W}^{(l)}\|_p \leq \omega$ for $l \in [L]$, we have

$$\mathfrak{R}_{m,n}(\Psi \circ \mathcal{F}) \leq 2K\varepsilon s(r^*, p, d)\omega\|g(\boldsymbol{A})\|_\infty^L(\omega+1)^L \times Q\mathbb{E}_{\boldsymbol{\sigma}}\Big|\sum_{i=1}^n \sigma_i\Big|$$

$$\leq Q_{m,n} 2K\varepsilon s(r^*, p, d)\omega\|g(\boldsymbol{A})\|_\infty^L(\omega+1)^L.$$

where $Q_{m,n} = \sqrt{\frac{2n}{m(n-m)}}$. Putting the above estimation back into the inequality (22) and combining Theorem 3.2, this completes the proof. $\qquad\square$

## F   PROOF OF PROPOSITION 5.5 [GCNII]

**Lemma F.1** (GCNII). *Let the robust surrogate loss be defined by*
$$\widehat{\ell}(f(\boldsymbol{A}, \boldsymbol{X})_i, y_i) = \ell_\gamma(M(f(\boldsymbol{A}, \boldsymbol{X})_i, y_i) - \Psi(f(\boldsymbol{A}, \boldsymbol{X})_i)),$$
*where $\Psi(f(\boldsymbol{A}, \widetilde{\boldsymbol{X}})_i)$ is $2\max_{k\in[K]}\|\boldsymbol{W}_{*k}^{(L)}\|_1\|\boldsymbol{W}^{(0)}\|_p\varepsilon s(r^*, p, d)\Lambda$ and $\Lambda$ is defined by*

$$\prod_{l=1}^{L-1}(1-\alpha)\|\widehat{\boldsymbol{W}}^{(l)}\|_2\|g(\boldsymbol{A})\|_\infty^l + \alpha\Big(\sum_{l=1}^{L-1}\Big((1-\alpha)\|g(\boldsymbol{A})\|_\infty^l\prod_{k=L-1-l}^{L-1}\|\widehat{\boldsymbol{W}}^{(l)}\|_2\Big)+1\Big),$$

*$\widehat{\boldsymbol{W}}^{(l)} = (1-\beta)\boldsymbol{I}_n + \beta\boldsymbol{W}^{(l)}$, and $s(r^*, p, d) = d^{\max\{0, \frac{1}{r^*} - \frac{1}{p}\}}$.*

*Proof of Lemma F.1.* The propagation of the GCNII is $f(\boldsymbol{A}, \boldsymbol{X}) = \mathrm{Softmax}(\boldsymbol{H}^{(L-1)}\boldsymbol{W}^{(L)})$ with the update rule
$$\boldsymbol{H}^{(l)} = \phi\big(((1-\alpha)g(\boldsymbol{A})\boldsymbol{H}^{(l-1)} + \alpha\boldsymbol{H}^{(0)})((1-\beta)\boldsymbol{I}_n + \beta\boldsymbol{W}^{(l)})\big), \quad \boldsymbol{H}^{(0)} = \boldsymbol{X}\boldsymbol{W}^{(0)}$$

where $\boldsymbol{I}_n$ is the identity matrix, $\boldsymbol{W}^{(0)} \in \mathbb{R}^{d\times d'}$, $\boldsymbol{W}^{(L)} \in \mathbb{R}^{d'\times K}$, $\boldsymbol{W}^{(l)} \in \mathbb{R}^{d'\times d'}$ for $l \in [L-1]$.

Denote $\widehat{\boldsymbol{W}}^{(l)} = (1-\beta)\boldsymbol{I}_n + \beta\boldsymbol{W}^{(l)}$. Let $\boldsymbol{H}^{(l)}$ and $\widetilde{\boldsymbol{H}}^{(l)}$ denote the feature representation of $\boldsymbol{X}$ and $\widetilde{\boldsymbol{X}}$ at $l$-th layer, respectively. Consider the difference between set of pairwise margin $f^{uv}(\boldsymbol{A}, \widetilde{\boldsymbol{X}})_i - f^{uv}(\boldsymbol{A}, \boldsymbol{X})_i$ of node $i$, where $f^{uv}(\boldsymbol{A}, \boldsymbol{X})_i = [f(\boldsymbol{A}, \boldsymbol{X})_i]_u - [f(\boldsymbol{A}, \boldsymbol{X})_i]_v$, having

$$\widetilde{\boldsymbol{H}}_{i*}^{(L-1)}(\boldsymbol{W}_{*u}^{(L)} - \boldsymbol{W}_{*v}^{(L)}) - \boldsymbol{H}_{i*}^{(L-1)}(\boldsymbol{W}_{*u}^{(L)} - \boldsymbol{W}_{*v}^{(L)})$$

$$\leq\|\boldsymbol{W}_{*u}^{(L)} - \boldsymbol{W}_{*v}^{(L)}\|_1\|\widetilde{\boldsymbol{H}}_{i*}^{(L-1)} - \boldsymbol{H}_{i*}^{(L-1)}\|_\infty$$

$$\leq\|\boldsymbol{W}_{*u}^{(L)} - \boldsymbol{W}_{*v}^{(L)}\|_1\Big\|\Big((1-\alpha)\sum_{j\in[n]}g(\boldsymbol{A})_{i,j}(\widetilde{\boldsymbol{H}}_{j*}^{(L-2)} - \boldsymbol{H}_{j*}^{(L-2)}) + \alpha(\widetilde{\boldsymbol{H}}_{i*}^{(0)} - \boldsymbol{H}_{i*}^{(0)})\Big)\widehat{\boldsymbol{W}}^{(L-1)}\Big\|$$

$$\leq\|\boldsymbol{W}_{*u}^{(L)} - \boldsymbol{W}_{*v}^{(L)}\|_1\|\widehat{\boldsymbol{W}}^{(L-1)}\|_2\Big(\Big\|(1-\alpha)\sum_{j\in[n]}g(\boldsymbol{A})_{i,j}(\widetilde{\boldsymbol{H}}_{j*}^{(L-2)} - \boldsymbol{H}_{j*}^{(L-2)})\Big\|+$$

$$\Big\|\alpha(\widetilde{\boldsymbol{H}}_{i*}^{(0)} - \boldsymbol{H}_{i*}^{(0)})\Big\|\Big)$$

$$\leq\|\boldsymbol{W}_{*u}^{(L)} - \boldsymbol{W}_{*v}^{(L)}\|_1\|\widehat{\boldsymbol{W}}^{(L-1)}\|_2\Big((1-\alpha)\max_{t\in[n]}\Big\|\Big(\max_{i\in[n]}\sum_{j\in[n]}g(\boldsymbol{A})_{i,j}\Big)(\widetilde{\boldsymbol{H}}_{t*}^{(L-2)} - \boldsymbol{H}_{t*}^{(L-2)})\Big\|+$$

$$\alpha\|\boldsymbol{W}^{(0)}\|_p\|\widetilde{\boldsymbol{x}}_i - \boldsymbol{x}_i\|_{p^*}\Big)$$

$$\overset{(a)}{\leq}\|\boldsymbol{W}_{*u}^{(L)} - \boldsymbol{W}_{*v}^{(L)}\|_1\|\widehat{\boldsymbol{W}}^{(L-1)}\|_2\Big((1-\alpha)\|g(\boldsymbol{A})\|_\infty\max_{j\in[n]}\Big\|\widetilde{\boldsymbol{H}}_{j*}^{(L-2)} - \boldsymbol{H}_{j*}^{(L-2)}\Big\|+$$

$$\alpha\|\boldsymbol{W}^{(0)}\|_p s(r^*, p, d)\|\widetilde{\boldsymbol{x}}_i - \boldsymbol{x}_i\|_r\Big)$$

$$\overset{(b)}{\leq}\|\boldsymbol{W}_{*u}^{(L)} - \boldsymbol{W}_{*v}^{(L)}\|_1\|\widehat{\boldsymbol{W}}^{(L-1)}\|_2\Big((1-\alpha)\|g(\boldsymbol{A})\|_\infty\max_{j\in[n]}\Big\|\widetilde{\boldsymbol{H}}_{j*}^{(L-2)} - \boldsymbol{H}_{j*}^{(L-2)}\Big\|+$$

$$\alpha\|\boldsymbol{W}^{(0)}\|_p s(r^*, p, d)\varepsilon\Big),$$

where $s(r^*, p, d) = d^{\max\{0, \frac{1}{r^*} - \frac{1}{p}\}}$, the inequality (a) follows from Lemma A.2 and the inequality (b) is due to $\|\widetilde{\boldsymbol{x}} - \boldsymbol{x}\|_r \leq \varepsilon$.

By recursive steps, we further obtain

$$\|\boldsymbol{W}_{*u}^{(L)} - \boldsymbol{W}_{*v}^{(L)}\|_1 \Big( \prod_{l=1}^{L-1} (1-\alpha)^l \|\widehat{\boldsymbol{W}}^{(l)}\|_2 \|g(\boldsymbol{A})\|_\infty^l \max_{j \in [n]} \|\widetilde{\boldsymbol{H}}_{j*}^{(0)} - \boldsymbol{H}_{j*}^{(0)}\| +$$

$$\alpha \|\boldsymbol{W}^{(0)}\|_p s(r^*, p, d)\varepsilon \Big( \sum_{l=1}^{L-1} \big( (1-\alpha)^l \|g(\boldsymbol{A})\|_\infty^l \prod_{k=L-1-l}^{L-1} \|\widehat{\boldsymbol{W}}^{(l)}\|_2 \big) + 1 \Big) \Big)$$

$$\leq \|\boldsymbol{W}_{*u}^{(L)} - \boldsymbol{W}_{*v}^{(L)}\|_1 \Big( \prod_{l=1}^{L-1} (1-\alpha) \|\widehat{\boldsymbol{W}}^{(l)}\|_2 \|g(\boldsymbol{A})\|_\infty^l \|\boldsymbol{W}^{(0)}\|_p \max_{j \in [n]} \|\widetilde{\boldsymbol{x}}_j - \boldsymbol{x}_j\|_{p^*} +$$

$$\alpha \|\boldsymbol{W}^{(0)}\|_p s(r^*, p, d)\varepsilon \Big( \sum_{l=1}^{L-1} \big( (1-\alpha) \|g(\boldsymbol{A})\|_\infty^l \prod_{k=L-1-l}^{L-1} \|\widehat{\boldsymbol{W}}^{(l)}\|_2 \big) + 1 \Big) \Big)$$

$$\leq \|\boldsymbol{W}_{*u}^{(L)} - \boldsymbol{W}_{*v}^{(L)}\|_1 \Big( \prod_{l=1}^{L-1} (1-\alpha) \|\widehat{\boldsymbol{W}}^{(l)}\|_2 \|g(\boldsymbol{A})\|_\infty^l \|\boldsymbol{W}^{(0)}\|_p s(r^*, p, d)\varepsilon +$$

$$\alpha \|\boldsymbol{W}^{(0)}\|_p s(r^*, p, d)\varepsilon \Big( \sum_{l=1}^{L-1} \big( (1-\alpha) \|g(\boldsymbol{A})\|_\infty^l \prod_{k=L-1-l}^{L-1} \|\widehat{\boldsymbol{W}}^{(l)}\|_2 \big) + 1 \Big) \Big)$$

$$:= \|\boldsymbol{W}_{*u}^{(L)} - \boldsymbol{W}_{*v}^{(L)}\|_1 \|\boldsymbol{W}^{(0)}\|_p \varepsilon s(r^*, p, d)\Lambda,$$

where $\Lambda$ is defined by

$$\prod_{l=1}^{L-1} (1-\alpha) \|\widehat{\boldsymbol{W}}^{(l)}\|_2 \|g(\boldsymbol{A})\|_\infty^l + \alpha \Big( \sum_{l=1}^{L-1} \big( (1-\alpha) \|g(\boldsymbol{A})\|_\infty^l \prod_{k=L-1-l}^{L-1} \|\widehat{\boldsymbol{W}}^{(l)}\|_2 \big) + 1 \Big).$$

According to the property of ramp loss, we have

$$\ell_\gamma \big( \min_{\widetilde{\boldsymbol{X}} \in \mathcal{B}_r^\varepsilon(\boldsymbol{X})} M(f(\boldsymbol{A}, \widetilde{\boldsymbol{X}})_i, y_i) \big)$$

$$\leq \ell_\gamma \big( \min_{y_i' \neq y_i} \min_{\widetilde{\boldsymbol{X}} \in \mathcal{B}_r^\varepsilon(\boldsymbol{X})} [f(\boldsymbol{A}, \widetilde{\boldsymbol{X}})_i]_{y_i} - [f(\boldsymbol{A}, \widetilde{\boldsymbol{X}})_i]_{y_i'} \big)$$

$$\leq \ell_\gamma \big( \min_{y_i' \neq y_i} [f(\boldsymbol{A}, \boldsymbol{X})_i]_{y_i} - [f(\boldsymbol{A}, \boldsymbol{X})_i]_{y_i'} - \max_{y_i' \neq y_i} \|\boldsymbol{W}_{*u}^{(L)} - \boldsymbol{W}_{*v}^{(L)}\|_1 \|\boldsymbol{W}^{(0)}\|_p \varepsilon s(r^*, p, d)\Lambda \big)$$

$$\leq \ell_\gamma \big( M(f(\boldsymbol{A}, \boldsymbol{X})_i, y_i) - 2 \max_{k \in [K]} \|\boldsymbol{W}_{*k}^{(L)}\|_1 \|\boldsymbol{W}^{(0)}\|_p \varepsilon s(r^*, p, d)\Lambda \big)$$

$$\leq \mathbb{1} \big( M(f(\boldsymbol{A}, \boldsymbol{X})_i, y_i) - 2 \max_{k \in [K]} \|\boldsymbol{W}_{*k}^{(L)}\|_1 \|\boldsymbol{W}^{(0)}\|_p \varepsilon s(r^*, p, d)\Lambda \leq \gamma \big).$$

$\square$

**Theorem F.2** (restate Proposition 5.5). *for any $\gamma > 0$, with probability at least $1 - \delta$, we have for all $f \in \mathcal{F}$,*

$$\frac{1}{n-m} \sum_{i=m+1}^{n} \mathbb{1}\{\exists \widetilde{\boldsymbol{X}} \in \mathcal{B}_r^\varepsilon(\boldsymbol{X}) \text{ s.t. } y_i \neq \arg\max_{y' \in [K]} [f(\boldsymbol{A}, \widetilde{\boldsymbol{X}})_i]_{y'}\}$$

$$\leq \frac{1}{m} \sum_{i=1}^{m} \mathbb{1}\{[f(\boldsymbol{A}, \boldsymbol{X})_i]_{y_i'} \leq \gamma + \max_{y_i' \neq y'} [f(\boldsymbol{A}, \boldsymbol{X})_i]_{y'} + \Psi(f(\boldsymbol{A}, \widetilde{\boldsymbol{X}})_i)\}$$

$$+ \mathcal{O}\big( \max\{\frac{1}{\sqrt{m}}, \frac{1}{\sqrt{n-m}}\} \big) + \frac{4KQ_{m,n}}{\gamma} (\sqrt{\log(2)L} + 1)(B_{p^*} \|\boldsymbol{X}\|_{2,p^*} + 2K\varepsilon s(r^*, p, d))$$

$$\times \omega^2 \Big( (1-\alpha) \|g(\boldsymbol{A})\|_\infty^L (1 - \beta + \beta\omega)^L + \alpha(1-\alpha) \sum_{l=0}^{L} \|g(\boldsymbol{A})\|_\infty^l (1 - \beta + \beta\omega)^l \Big).$$

where $s(r^*, p, d) = d^{\max\{0, \frac{1}{r^*} - \frac{1}{p}\}}$, $B_{p*} = \sqrt{2\log(2d)}$ if $p = 1$, $B_{p*} = \sqrt{2}[\frac{\Gamma(\frac{1+p^*}{2})}{\sqrt{\pi}}]^{\frac{1}{p^*}}$ if $p \in (1, 2]$, $B_{p*} = 1$ if $p \in [2, +\infty)$, $Q_{m,n} = \sqrt{\frac{2n}{m(n-m)}}$.

*Proof of Theorem F.2.* With Lemma F.1 and the Ledoux-Talagrand contraction inequality, we have

$$\mathfrak{R}_{m,n}(\widehat{\ell} \circ \mathcal{F}) \le \frac{1}{\gamma}\big(\mathfrak{R}_{m,n}(M \circ \mathcal{F}) + \mathfrak{R}_{m,n}(\Psi \circ \mathcal{F})\big). \tag{25}$$

Let the hypothesis class of GCNII be defined by

$$\mathcal{F} = \big\{ f(\boldsymbol{A}, \boldsymbol{X}) = \text{Softmax}(\boldsymbol{H}^{(L-1)}\boldsymbol{W}^{(L)}) : \|\boldsymbol{W}^{(l)}\|_2, \|\boldsymbol{W}^{(l)}\|_p \le \omega, l \in [L] \big\} \tag{26}$$

with layer-wise update rule:

$$\boldsymbol{H}^{(l)} = \phi\big(((1-\alpha)g(\boldsymbol{A})\boldsymbol{H}^{(l-1)} + \alpha\boldsymbol{H}^{(0)})((1-\beta)\boldsymbol{I}_n + \beta\boldsymbol{W}^{(l)})\big), \quad \boldsymbol{H}^{(0)} = \boldsymbol{X}\boldsymbol{W}^{(0)}$$

where $\boldsymbol{I}_n$ is the identity matrix, $\boldsymbol{W}^{(0)} \in \mathbb{R}^{d \times d'}$, $\boldsymbol{W}^{(L)} \in \mathbb{R}^{d' \times K}$ and $\boldsymbol{W}^{(l)} \in \mathbb{R}^{d' \times d'}$ for $l \in [L-1]$. Denote $\widehat{\boldsymbol{W}}^{(l)} = (1-\beta)\boldsymbol{I}_n + \beta\boldsymbol{W}^{(l)}$ for $l \in [L-1]$. We then have $\|\widehat{\boldsymbol{W}}^{(l)}\|_2 \le (1-\beta) + \beta\omega$.

For the right-hand side of inequality (25), $\mathfrak{R}_{m,n}(M \circ \mathcal{F})$ is bounded by

$$Q\mathbb{E}_{\boldsymbol{\sigma}} \sup_{\|\boldsymbol{W}^{(L)}\|_2 \le \omega} \sum_{i=1}^n \sigma_i \left\langle \boldsymbol{H}_{i*}^{(L-1)}, \boldsymbol{W}_{*y_i}^{(L)} \right\rangle$$

$$\le Q\mathbb{E}_{\boldsymbol{\sigma}} \omega \sup \Big\| \sum_{i=1}^n \sigma_i \boldsymbol{H}_{i*}^{(L-1)} \Big\|$$

$$\le Q\omega\mathbb{E}_{\boldsymbol{\sigma}} \sup_{\|\widehat{\boldsymbol{W}}^{(L-1)}\|_2 \le \widehat{\omega}, t \in [n]} \Big\| \sum_{i=1}^n \sigma_i \phi\Big(\big((1-\alpha)\big(\max_{i \in [n]} \sum_{j \in [n]} g(\boldsymbol{A})_{i,j}\big)\boldsymbol{H}_{t*}^{(L-2)} + \alpha\boldsymbol{H}_{t*}^{(0)}\big)\widehat{\boldsymbol{W}}^{(L-1)}\Big) \Big\|$$

$$\le Q\omega\mathbb{E}_{\boldsymbol{\sigma}} \sup_{\|\widehat{\boldsymbol{W}}^{(L-1)}\|_2 \le \widehat{\omega}, j \in [n]} \Big\| \sum_{i=1}^n \sigma_i \phi\Big(\big((1-\alpha)\|g(\boldsymbol{A})\|_\infty \boldsymbol{H}_{j*}^{(L-2)} + \alpha\boldsymbol{H}_{j*}^{(0)}\big)\widehat{\boldsymbol{W}}^{(L-1)}\Big) \Big\|$$

$$\overset{(a)}{\le} Q\frac{1}{\lambda} \log 2\mathbb{E}_{\boldsymbol{\sigma}} \sup_{j \in [n]} \exp\Big(\lambda\omega\widehat{\omega}\Big\| \sum_{i=1}^n \sigma_i\big((1-\alpha)\|g(\boldsymbol{A})\|_\infty \boldsymbol{H}_{j*}^{(L-2)} + \alpha\boldsymbol{H}_{j*}^{(0)}\big)\Big\|\Big)$$

$$\le Q\frac{1}{\lambda} \log 2\mathbb{E}_{\boldsymbol{\sigma}} \sup_{\|\boldsymbol{W}^{(0)}\|_p \le \omega, j \in [n]} \exp\Big(\lambda\omega\widehat{\omega}\big((1-\alpha)\|g(\boldsymbol{A})\|_\infty\Big\| \sum_{i=1}^n \sigma_i \boldsymbol{H}_{j*}^{(L-2)}\Big\| + \alpha\Big\| \sum_{i=1}^n \sigma_i \boldsymbol{x}_i \boldsymbol{W}^{(0)}\Big\|\big)\Big)$$

$$\overset{(b)}{\le} Q\frac{1}{\lambda} \log 2\mathbb{E}_{\boldsymbol{\sigma}} \sup_{j \in [n]} \exp\Big(\lambda\omega\widehat{\omega}\big((1-\alpha)\|g(\boldsymbol{A})\|_\infty\Big\| \sum_{i=1}^n \sigma_i \boldsymbol{H}_{j*}^{(L-2)}\Big\| + \alpha\omega B_{p*}\|\boldsymbol{X}\|_{2,p^*}\big)\Big)$$

where $\widehat{\omega} = (1-\beta) + \beta\omega$, $B_{p*} = \sqrt{2\log(2d)}$ if $p = 1$; $B_{p*} = \sqrt{2}[\frac{\Gamma(\frac{1+p^*}{2})}{\sqrt{\pi}}]^{\frac{1}{p^*}}$ if $p \in (1, 2]$; $B_{p*} = 1$ if $p \in [2, +\infty)$, the inequality (a) follows from Lemma 1 of (Golowich et al., 2018), and the inequality (b) follows from the bounds on the Rademacher complexity given by Awasthi et al. (2020).

Applying recursive steps, $\mathfrak{R}_{m,n}(M \circ \mathcal{F})$ can be further bounded by

$$Q\frac{1}{\lambda} \log 2^L \mathbb{E}_{\boldsymbol{\sigma}} \sup_{\|\boldsymbol{W}^{(0)}\|_p \le \omega, j \in [n]} \exp\Big(\lambda\big(\omega\widehat{\omega}^{L-1}(1-\alpha)\|g(\boldsymbol{A})\|_\infty^{L-1}\Big\| \sum_{i=1}^n \sigma_i \boldsymbol{X}_{j*}\boldsymbol{W}^{(0)}\Big\| +$$

$$\alpha(1-\alpha)\omega^2 B_{p*}\|\boldsymbol{X}\|_{2,p^*} \sum_{l=0}^{L-1} \|g(\boldsymbol{A})\|_\infty^l \widehat{\omega}^l\big)\Big)$$

$$\le Q\frac{1}{\lambda} \log 2^L \mathbb{E}_{\boldsymbol{\sigma}} \exp\Big(\lambda\big(\omega\widehat{\omega}^L(1-\alpha)\|g(\boldsymbol{A})\|_\infty^L\Big\| \sum_{i=1}^n \sigma_i \boldsymbol{x}_i\Big\|_{p^*} +$$

$$\alpha(1-\alpha)\omega B_{p*}\|\boldsymbol{X}\|_{2,p^*} \sum_{l=0}^{L} \|g(\boldsymbol{A})\|_\infty^l \widehat{\omega}^l\big)\Big),$$

where $\widehat{\omega} = (1 - \beta) + \beta\omega$. Denote $U = \alpha(1 - \alpha)\omega B_{p^*}\|\boldsymbol{X}\|_{2,p^*}\sum_{l=0}^{L}\|g(\boldsymbol{A})\|_{\infty}^{l}\widehat{\omega}^{l}$ and $M = \omega^2\widehat{\omega}^{L-1}(1 - \alpha)\|g(\boldsymbol{A})\|_{\infty}^{L}$. We define the following random variable

$$Z = M \cdot \Big\|\sum_{i=1}^{n}\sigma_i\boldsymbol{x}_i\Big\|_{p^*} + U.$$

Then we have

$$\frac{1}{\lambda}\log\Big\{2^L\mathbb{E}\exp\lambda Z\Big\} = \frac{L\log(2)}{\lambda} + \frac{1}{\lambda}\log\{\mathbb{E}\exp\lambda(Z - \mathbb{E}Z)\} + \mathbb{E}Z.$$

With the inequality (14), it is clear that

$$\mathbb{E}Z = M \cdot \mathbb{E}_{\boldsymbol{\sigma}}\Big\|\sum_{i=1}^{n}\sigma_i\boldsymbol{x}_i\Big\|_{p^*} + U \leq MB_{p^*}\|\boldsymbol{X}\|_{2,p^*} + U$$

Note that $Z$ is a deterministic function of $\sigma_1,\ldots,\sigma_n$, and satisfies

$$Z(\sigma_1,\ldots,\sigma_i,\ldots,\sigma_n) - Z(\sigma_1,\ldots,-\sigma_i,\ldots,\sigma_n) \leq 2M\|\boldsymbol{x}_i\|_{p^*}, \tag{27}$$

which implies $Z$ satisfies a bounded-difference property and is sub-Gaussian with the variance factor

$$\upsilon = \frac{1}{4}\sum_{i=1}^{n}(2M\|\boldsymbol{x}_i\|_{p^*})^2 = M^2\sum_{i=1}^{n}\|\boldsymbol{x}_i\|_{p^*}^2.$$

Therefore, the following inequality holds

$$\frac{1}{\lambda}\log\{\mathbb{E}\exp\lambda(Z - \mathbb{E}Z)\} \leq \frac{\lambda M^2\sum_{i=1}^{n}\|\boldsymbol{x}_i\|_{p^*}^2}{2}.$$

Letting $\lambda = \dfrac{\sqrt{2L\log(2)}}{M\sqrt{\sum_{i=1}^{n}\|\boldsymbol{x}_i\|_{p^*}^2}}$, the following inequality holds

$$Q\frac{1}{\lambda}\log\Big\{2^L\mathbb{E}\exp\lambda Z\Big\} \leq Q\Big(\mathbb{E}Z + \sqrt{2\log(2)L}M\sqrt{\sum_{i=1}^{n}\|\boldsymbol{x}_i\|_{p^*}^2}\Big)$$

$$\leq Q_{m,n}(\sqrt{2\log(2)L} + 1)(MB_{p^*}\|\boldsymbol{X}\|_{2,p^*} + U)$$

where $Q_{m,n} = \sqrt{\dfrac{2n}{m(n-m)}}$, and $B_{p*} = \sqrt{2\log(2d)}$ if $p = 1$; $B_{p*} = \sqrt{2}[\frac{\Gamma(\frac{1+p^*}{2})}{\sqrt{\pi}}]^{\frac{1}{p^*}}$ if $p \in (1, 2]$; $B_{p*} = 1$ if $p \in [2, +\infty)$. Thus, we have the following upper bound on $\Re_{m,n}(M \circ \mathcal{F})$,

$$Q_{m,n}(\sqrt{2\log(2)L} + 1)B_{p^*}\|\boldsymbol{X}\|_{2,p^*}\omega^2\Big((1 - \alpha)\|g(\boldsymbol{A})\|_{\infty}^{L}\widehat{\omega}^{L} + \alpha(1 - \alpha)\sum_{l=0}^{L}\|g(\boldsymbol{A})\|_{\infty}^{l}\widehat{\omega}^{l}\Big),$$

where $\widehat{\omega} = (1 - \beta) + \beta\omega$. We proceed to derive an upper bound on $\Re_{m,n}(\Psi \circ \mathcal{F})$, that is

$$Q\mathbb{E}_{\boldsymbol{\sigma}}\sup_{\|\boldsymbol{W}^{(l)}\|_p,\|\boldsymbol{W}^{(l)}\|_2\leq\omega,l\in[L]}\sum_{i=1}^{n}\sigma_i\Psi(f(\boldsymbol{A}, \widetilde{\boldsymbol{X}})_i)$$

$$\leq 2K\varepsilon s(r^*, p, d)\omega^2\Big((1 - \alpha)\|g(\boldsymbol{A})\|_{\infty}^{L}\widehat{\omega}^{L} + \alpha(1 - \alpha)\sum_{l=0}^{L}\|g(\boldsymbol{A})\|_{\infty}^{l}\widehat{\omega}^{l}\Big) \times Q\mathbb{E}_{\boldsymbol{\sigma}}\Big|\sum_{i=1}^{n}\sigma_i\Big|$$

$$\leq 2Q_{m,n}K\varepsilon s(r^*, p, d)\omega^2\Big((1 - \alpha)\|g(\boldsymbol{A})\|_{\infty}^{L}\widehat{\omega}^{L} + \alpha(1 - \alpha)\sum_{l=0}^{L}\|g(\boldsymbol{A})\|_{\infty}^{l}\widehat{\omega}^{l}\Big)$$

where $Q_{m,n} = \sqrt{\dfrac{2n}{m(n-m)}}$, $\widehat{\omega} = (1 - \beta) + \beta\omega$, and $s(r^*, p, d) = d^{\max\{0,\frac{1}{r^*} - \frac{1}{p}\}}$. $\qquad\square$

## G    LIMITATIONS

We outline a few limitations of the current adversarial generalization analysis for GCNs. Since Rademacher complexity-based analysis methods are used in this paper, our bounds suffer from the exponential depth dependence similar to (Bartlett et al., 2017; Golowich et al., 2018). Nonetheless, the derived high-probability generalization bounds for GCNs in adversarial settings are highly non-trivial, which provides theoretical guidance for improving the adversarial robustness. Moreover, this paper does not involve topology attacks, which leads to the limitation of theoretical results. However, our analysis can provide valuable theoretical insights into the generalization of topology attacks. Given the similar settings of topology attacks and node attacks, where adversarial perturbations are measured within the norm space, the methodology (e.g., Lemma B.2) developed in this paper could be expanded upon the topology attack to address the outer maximization of the adversarial loss and tighten the generalization bounds. It is worth noting that unlike node attacks, one can perturb the graph topology at the first layer, the last layer, the arbitrary intermediate layer, and all the layers of GCNs, thus posing additional analysis challenges. We will leave this interesting study as future work.

## H    ADDITIONAL RELATED WORK

**Adversarial Generalization Theory.**    Adversarial robust generalization in Euclidean space has been studied extensively in recent years. Yin et al. (2019) investigate the adversarial generalization problem under $\ell_\infty$-norm attacks via the lens of Rademacher complexity. They derive the high-probability generalization bounds for linear classifiers and one-hidden-layer neural networks, indicating the additional perturbation term that should be suppressed seriously affect the generalization performance. Awasthi et al. (2020) establish the generalization bounds for one-hidden-layer additive neural networks under general perturbation attacks by extending the work of (Yin et al., 2019) to the $\ell_r$-norm attacks for arbitrary $r \geq 1$. Several recent work (Khim & Loh, 2018; Tu et al., 2019; Mustafa et al., 2022) has aimed to analyze the generalization properties of deep neural networks in adversarial settings. Khim & Loh (2018) provide the upper bounds of the adversarial risk by leveraging a tree based decomposition. Their results depend on the assumption that each propagation path in the network can be optimized independently w.r.t. the perturbation, leading to vacuous bounds in (Khim & Loh, 2018). Tu et al. (2019) introduce a transport map between distributions and develop a new risk bound for multi-layer neural networks by means of covering numbers under the Lipschitz condition. Mustafa et al. (2022) establish the adversarial risk bounds for deep neural networks under both additive-perturbation attacks and transform attacks via a novel usage of covering numbers. Xing et al. (2021) and Xiao et al. (2022) investigate the adversarially robust generalization of learning algorithms from the perspective of algorithmic stability, showing that the maximization process w.r.t. adversarial perturbations causes the worse algorithmic stability than natural training. Although all the aforementioned work provides valuable insights into the adversarial generalization problem, it is difficult to apply them to adversarial graph learning due to discrepancies in learning paradigms and data characteristics. This work makes towards this by providing generalization guarantees of GCNs under node perturbation attacks, which is the first touch for adversarial graph learning to the best of our knowledge. Related work is summarized in Table 3.

Table 3: Summary of generalization analysis for adversarial learning (NNs-Neural Networks; $k$-the number of classes; $m$-the number of unlabeled samples; $n$-the number of samples).

| Reference | Model Structure | Attack Type | Analysis Tool | Convergence Rate | Learning Setting |
|---|---|---|---|---|---|
| Yin et al. (2019) | One-layer NNs | $\ell_\infty$ norm | Rademacher complexity | $\mathcal{O}(k/\sqrt{n})$ | Inductive |
| Awasthi et al. (2020) | | $\ell_r$ norm | | | Inductive |
| Khim and Loh (2018) | | $\ell_\infty$ norm | Rademacher complexity | $\mathcal{O}(1/\sqrt{n})$ | Inductive |
| Tu et al. (2019) | | $\ell_r$ norm | Covering number | $\mathcal{O}(1/\sqrt{n})$ | Inductive |
| Mustafa et al. (2022) | Deep NNs | $\ell_r$ norm | Covering number | $\mathcal{O}(\log(k)/\sqrt{n})$ | Inductive |
| | | Transformation | Local Rademacher complexity | $\mathcal{O}(1/n)$ | Inductive |
| Xing et al. (2021) | | $\ell_r$ norm | Algorithmic stability | $\mathcal{O}(1/n)$ | Inductive |
| Xiao et al. (2022) | | | | | Inductive |
| **Ours** | **Deep GCNs** | **$\ell_r$ norm** | **Transductive Rademacher Complexity** | $\mathcal{O}(k \max\{\frac{1}{m}, \frac{1}{n-m}\})$ | **Transductive** |

# I ADDITIONAL EXPERIMENTS FOR NUMERICAL DISCUSSION

**Experiment details.** Unless otherwise specified, we apply a two-layer network architecture for GCN, SGC, GCNII, and Residual GCN (Kipf & Welling, 2017; Wu et al., 2019; Chen et al., 2020b), where the number of hidden units for each layer is fixed to 16 or 64. For GCNII, the parameter $\alpha$ is set by default to 0.5, $\beta$ is set to $\log(\frac{\theta}{l} + 1)$, where $\theta = 0.1$ and $l$ is the number of layers. We use ReLU function as activation function. The Adam optimizer (Kingma & Ba, 2015) with the learning rate 0.01 is used in the training process. The implement is GeForce RTX 3080 GPU. The runtime of each experiment is about 200s∼2h.

**Attack methods.** We evaluate the generalization performance under different attacks including FGSM, BIM, and PGD. Figures 8-13 present the empirical generalization errors with dimensions of adopted datasets under attacks. As we can see, the experimental results for different attacks have the same overall trend (i.e., the smaller the dimension, the smaller the generalization error), which can not be affected by the choice of attack methods.

**Feature dimension.** We study the effect of feature dimension on the generalization performance of SGC, GCNII, and Residual GCN in adversarial settings. We compare the empirical generalization error with different dimensions, including the original dimension $d$, $\lceil d/2 \rceil$, and $\lceil d/4 \rceil$. For convenience, a single-layer neural network with ReLU activation is used to learn a low-dimensional feature representation. As shown in Figures 14-16, the empirical generalization error decreases with node features, which suggests that low-dimensional features help improve the adversarial generalization.

**Regularization.** We also evaluate the effect of regularization on the generalization ability for SGC, GCNII, and Residual GCN under adversarial attacks. Similarly, we consider $\ell_1$-norm regularized model and observe the empirical generalization error with different regularization coefficients $\lambda$, where $\lambda$ is set as 0, 0.05, 0.10, and 0.15. Figure 17-19 show that the empirical generalization error of the regularized model is smaller than that without. Hence, the regularization is beneficial to improve the generalization performance.

**Model depth.** We study the effect of the number of layers on the adversarial generalization for SGC, GCNII, and Residual GCN. Figures 20-22 present the empirical generalization errors with different depths. As we can see, the generalization performance of SGC deteriorates as depth increases, which is limited by its representation power. The generalization ability of GCNII and Residual GCN almost can not be affected by the number of layers. Hence, we suggest that the choice of model architecture is important for achieving the adversarial generalization.

**Tightness.** To evaluate the tightness between theory and experimental observations, we compare the generalization bounds in theory and empirical generalization errors w.r.t. feature dimensions in Figure 23, where the bounds are quantitatively computed based on the adopted dataset and the model parameters. The experimental results show that both theoretical and empirical generalization errors decrease as the feature dimension decreases, which is consistent with our theoretical findings. Hence, our results provide meaningful practical guidance for improving the adversarial generalization of the learning models in a certain sense.

**Generalize to other models.** To evaluate the scalability of the theoretical findings, we further investigate the impact of the feature dimension on the generalization performance of S2GC (Zhu & Koniusz, 2021) and GIN (Xu et al., 2018) under PGD attacks. The empirical errors with different dimensions for the adopted datasets are plotted in Figure 24, demonstrating the low-dimensional feature mapping can enhace the adversarial robustness of the learning models. This observation aligns with our theoretical findings.

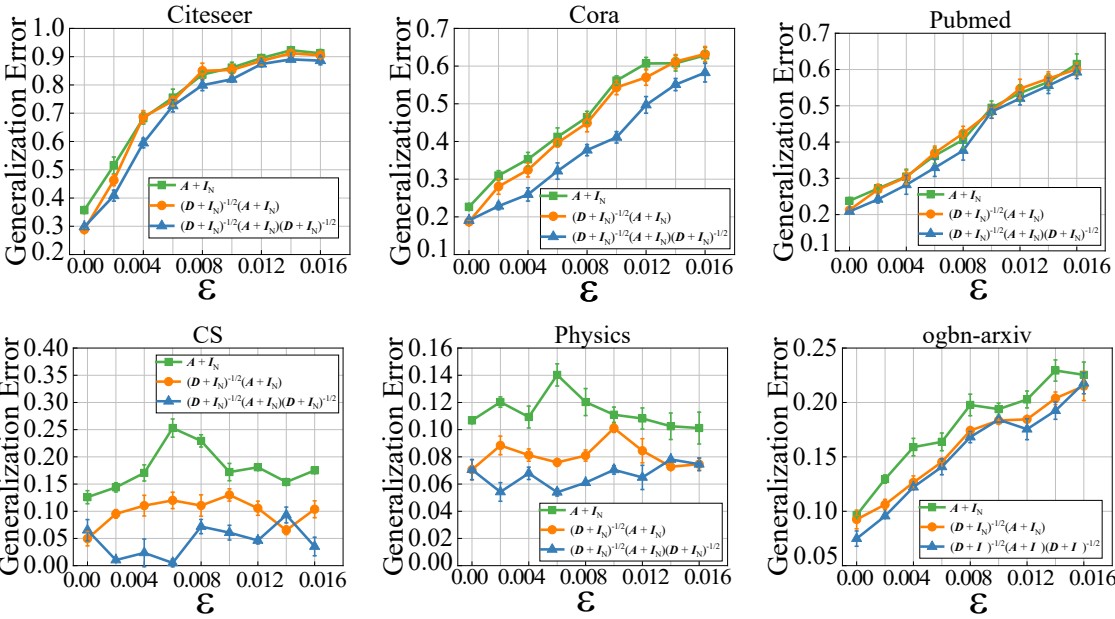

Figure 3: The empirical generalization error (mean value and standard deviation) with graph filters, where depth is set to 6. $\varepsilon$ denotes the maximum allowable perturbation.

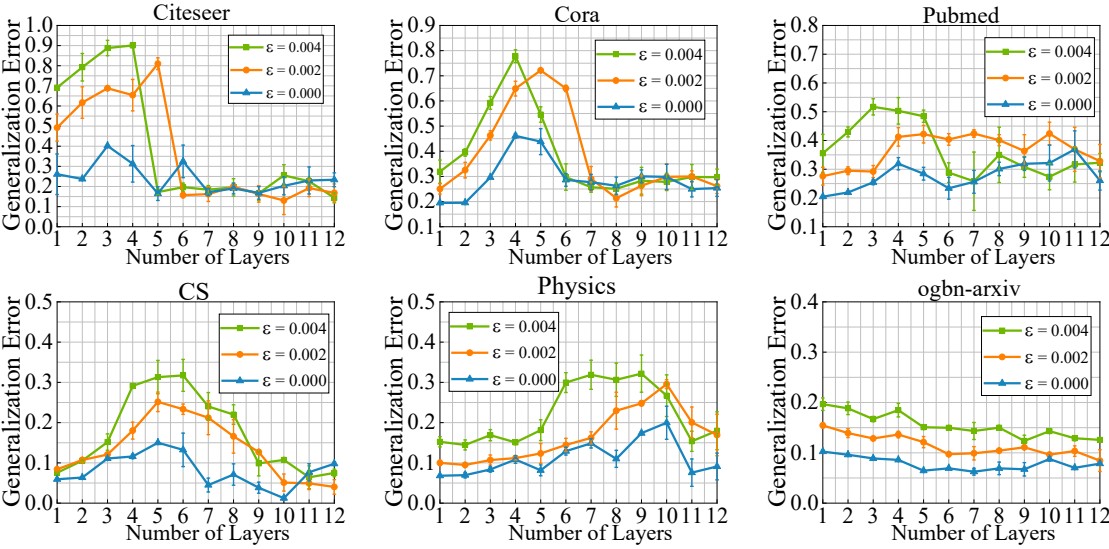

Figure 4: The empirical generalization error (mean value and standard deviation) with different depths. $\varepsilon$ denotes the maximum allowable perturbation.

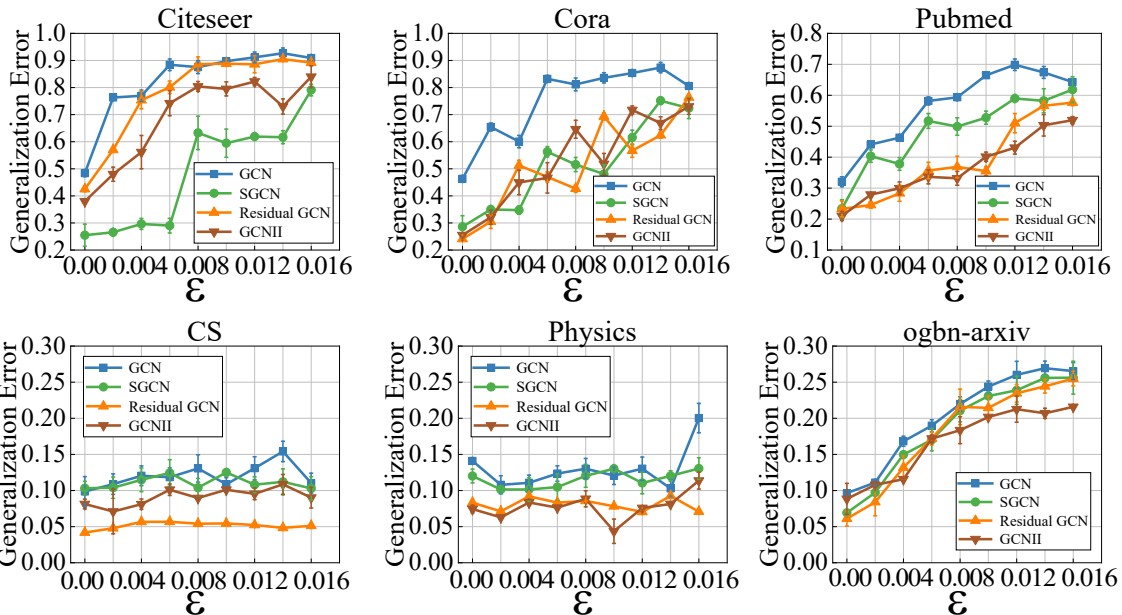

Figure 5: The empirical generalization error (mean value and standard deviation) for different model architectures, where depth is set to 6. $\varepsilon$ denotes the maximum allowable perturbation.

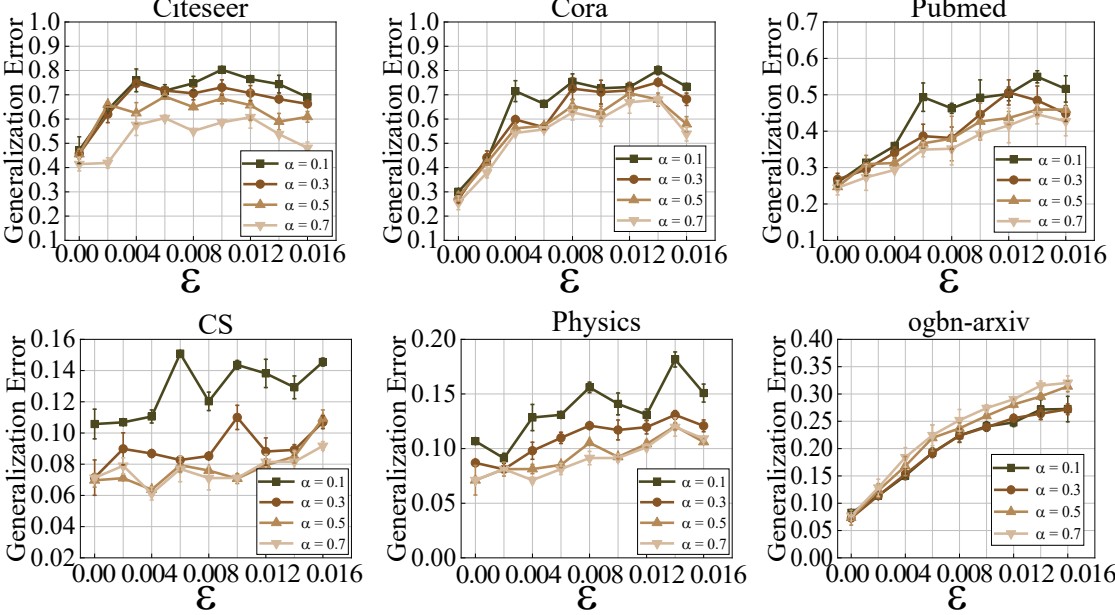

Figure 6: The empirical generalization error (mean value and standard deviation) of GCNII with the parameter $\alpha$, where depth is set to 6. $\varepsilon$ denotes the maximum allowable perturbation.

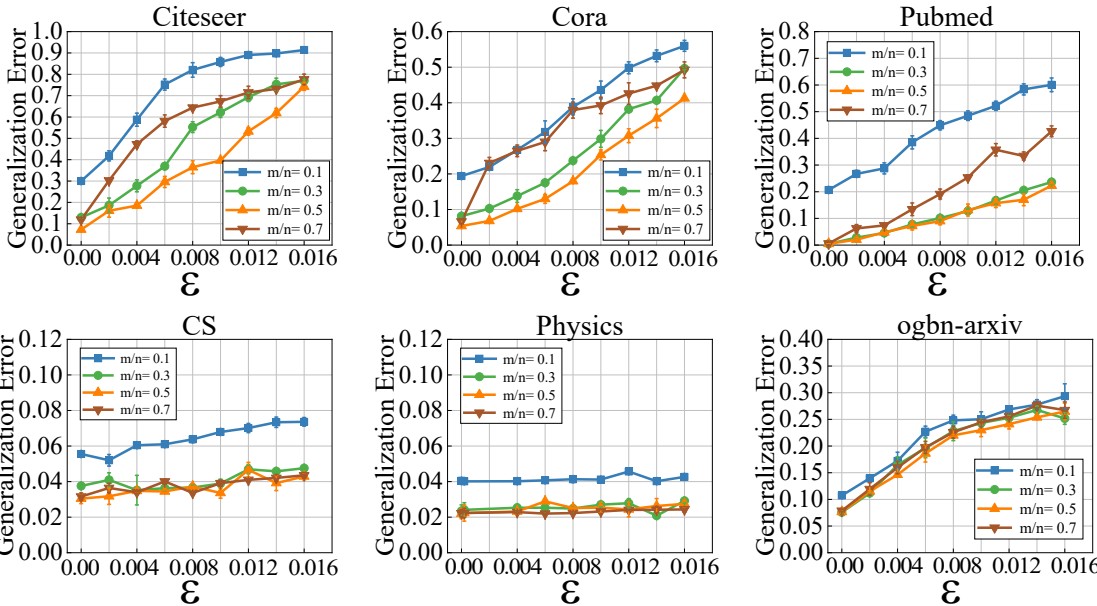

Figure 7: The empirical generalization error (mean value and standard deviation) with the label rate $m/n$. $\varepsilon$ denotes the maximum allowable perturbation.

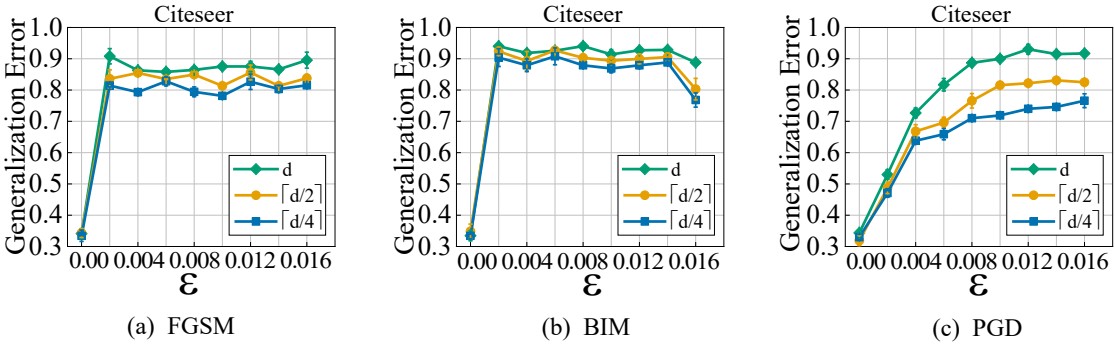

Figure 8: The empirical generalization error (mean value and standard deviation) for the Citeseer dataset under different attacks. $\varepsilon$ denotes the maximum allowable perturbation.

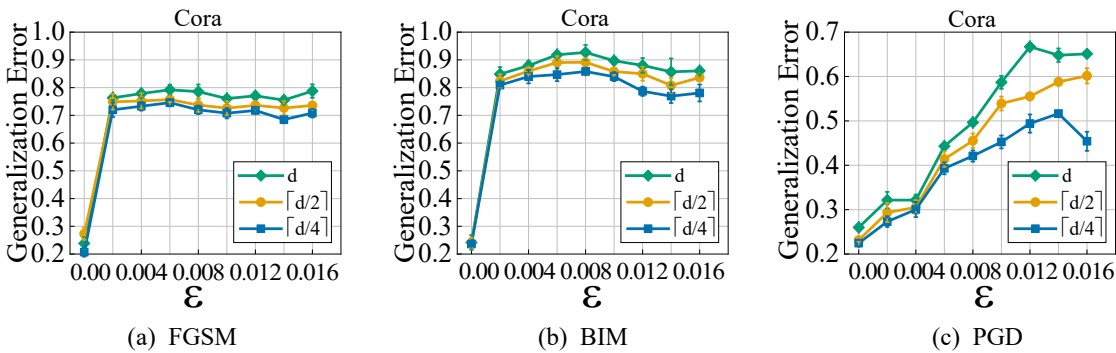

Figure 9: The empirical generalization error (mean value and standard deviation) for the Cora dataset under different attacks. $\varepsilon$ denotes the maximum allowable perturbation.

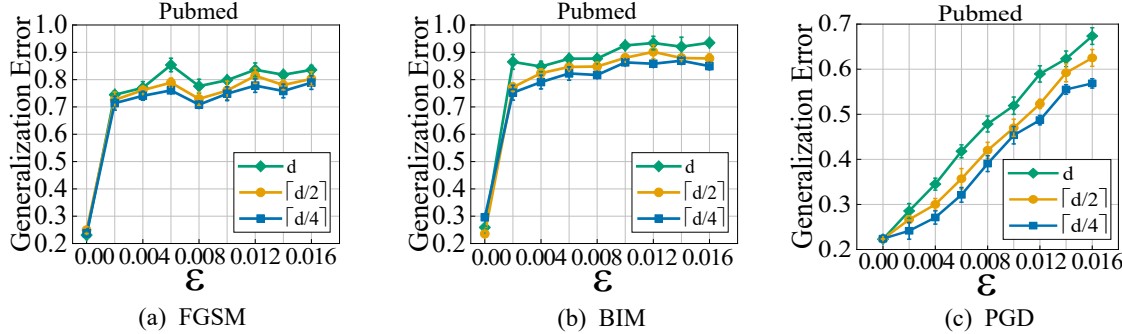

Figure 10: The empirical generalization error (mean value and standard deviation) for the Pubmed dataset under different attacks. $\varepsilon$ denotes the maximum allowable perturbation.

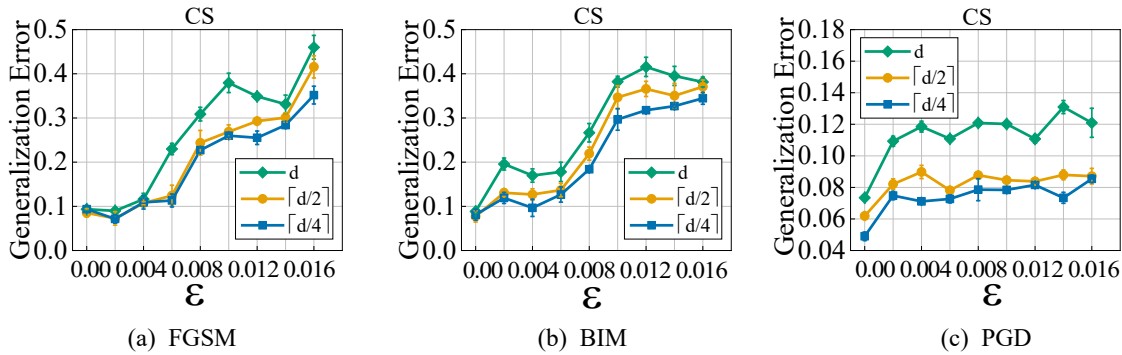

Figure 11: The empirical generalization error (mean value and standard deviation) for the CS dataset under different attacks. $\varepsilon$ denotes the maximum allowable perturbation.

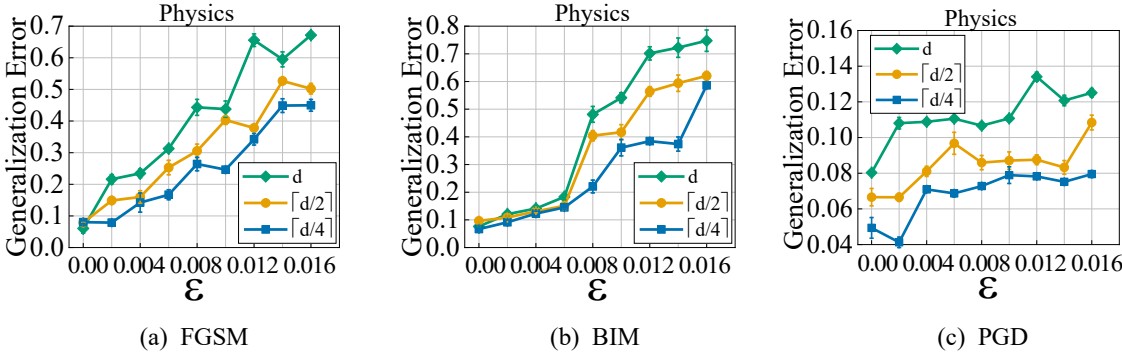

Figure 12: The empirical generalization error (mean value and standard deviation) for the Physics dataset under different attacks. $\varepsilon$ denotes the maximum allowable perturbation.

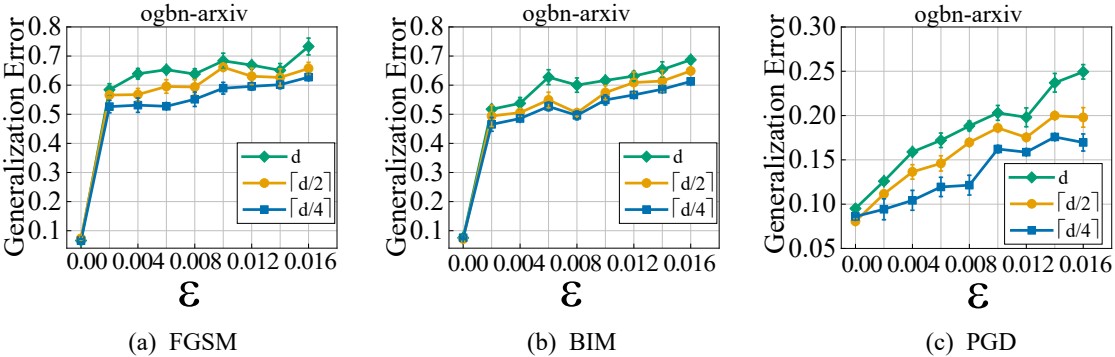

Figure 13: The empirical generalization error (mean value and standard deviation) for the ogbn-arxiv dataset under different attacks. $\varepsilon$ denotes the maximum allowable perturbation.

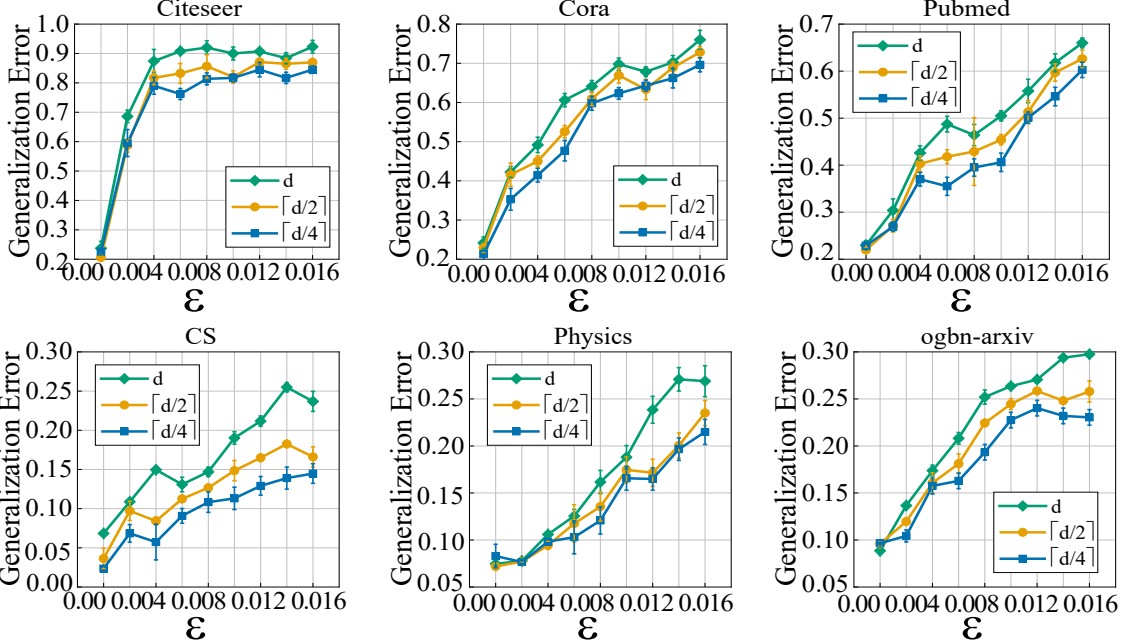

Figure 14: The empirical generalization error (mean value and standard deviation) with different feature dimensions for SGC. $\varepsilon$ denotes the maximum allowable perturbation.

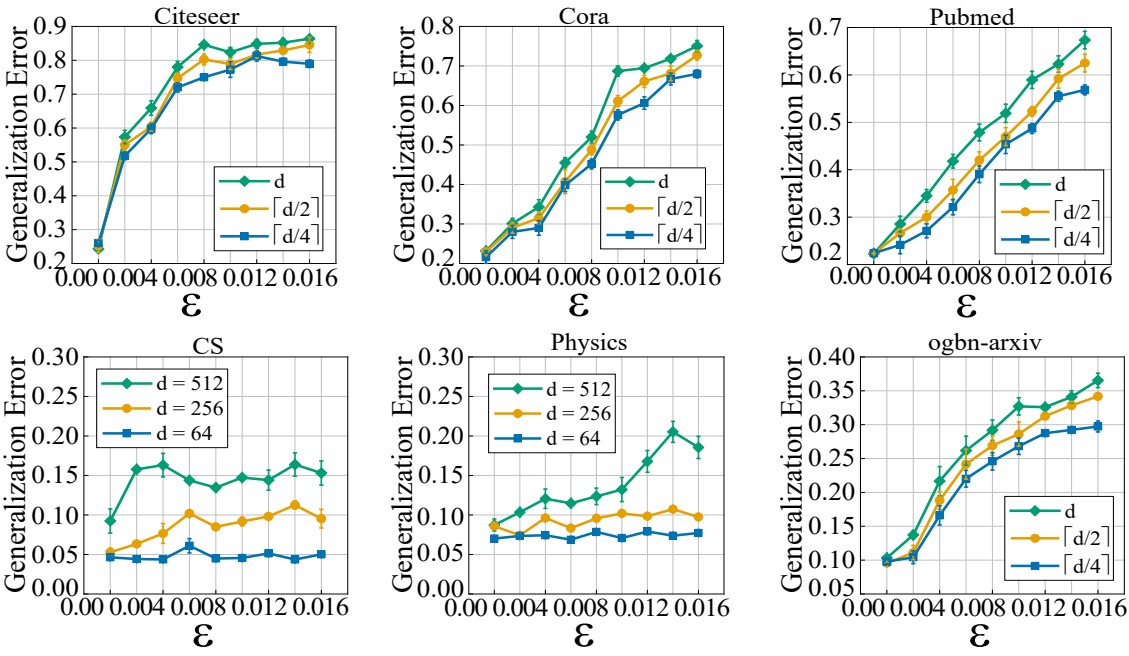

Figure 15: The empirical generalization error (mean value and standard deviation) with different feature dimensions for GCNII. $\varepsilon$ denotes the maximum allowable perturbation.

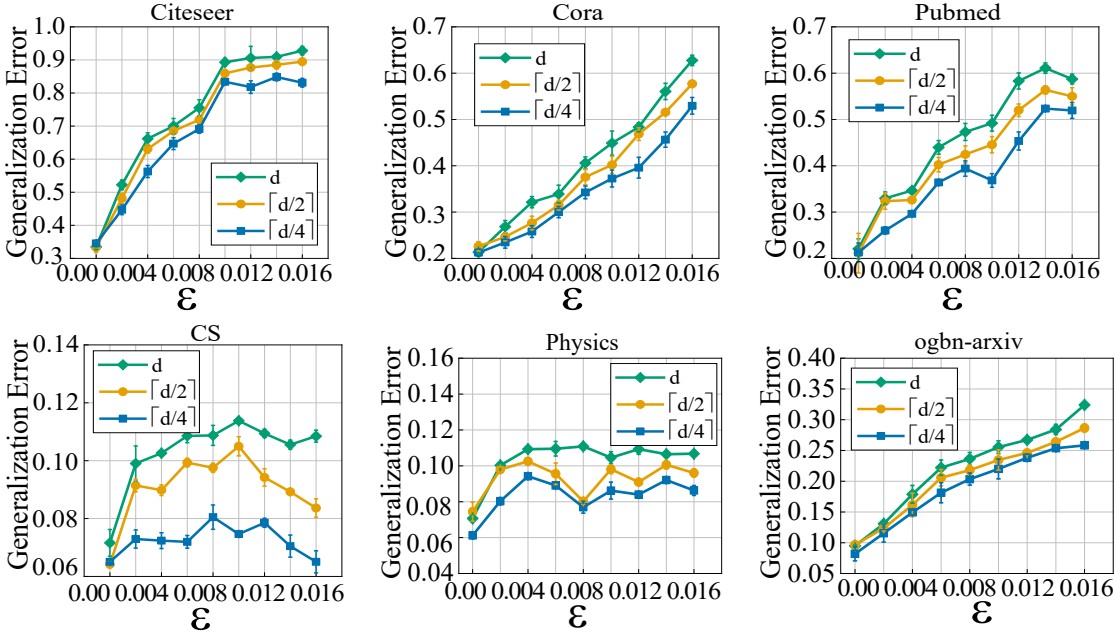

Figure 16: The empirical generalization error (mean value and standard deviation) with different feature dimensions for Residual GCN. $\varepsilon$ denotes the maximum allowable perturbation.

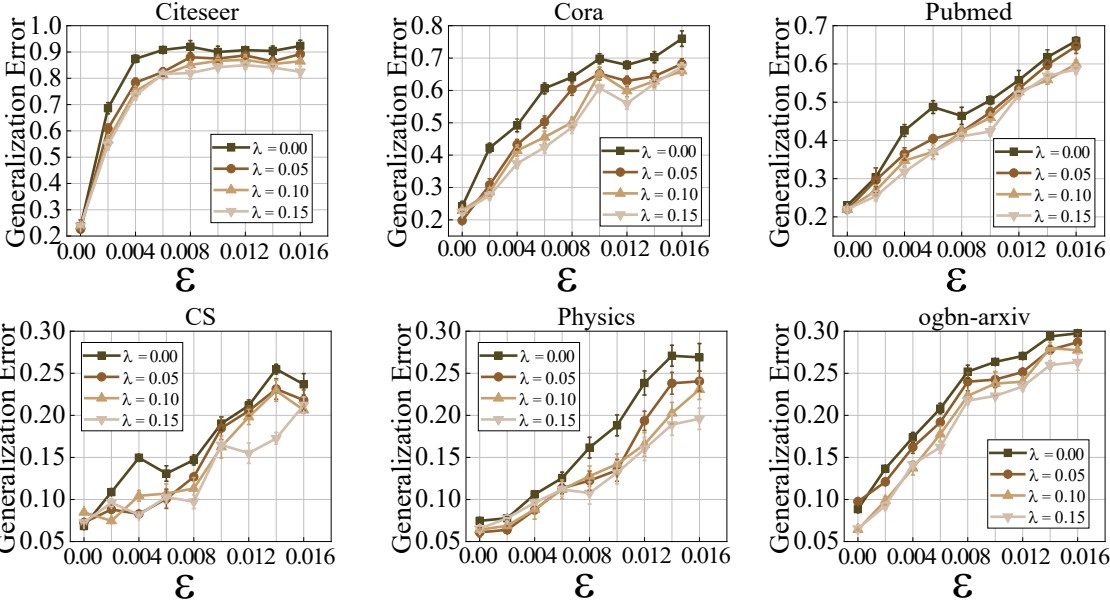

Figure 17: The empirical generalization error (mean value and standard deviation) of SGC trained with $\ell_1$ regularization for different regularization parameters (i.e., $\lambda$). $\varepsilon$ denotes the maximum allowable perturbation.

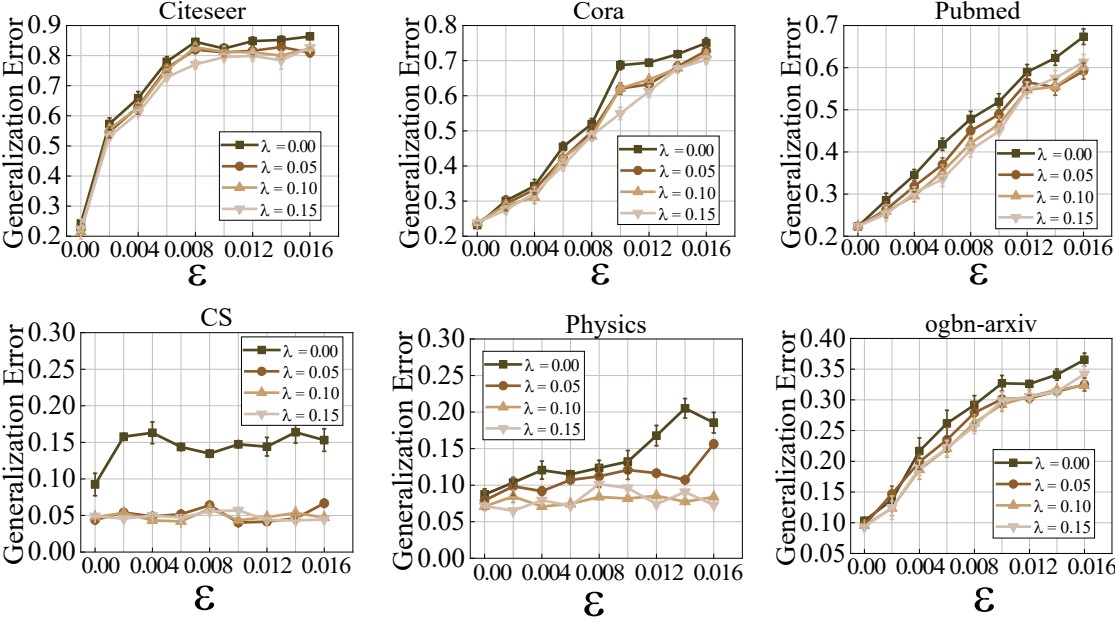

Figure 18: The empirical generalization error (mean value and standard deviation) of GCNII trained with $\ell_1$ regularization for different regularization parameters (i.e., $\lambda$). $\varepsilon$ denotes the maximum allowable perturbation.

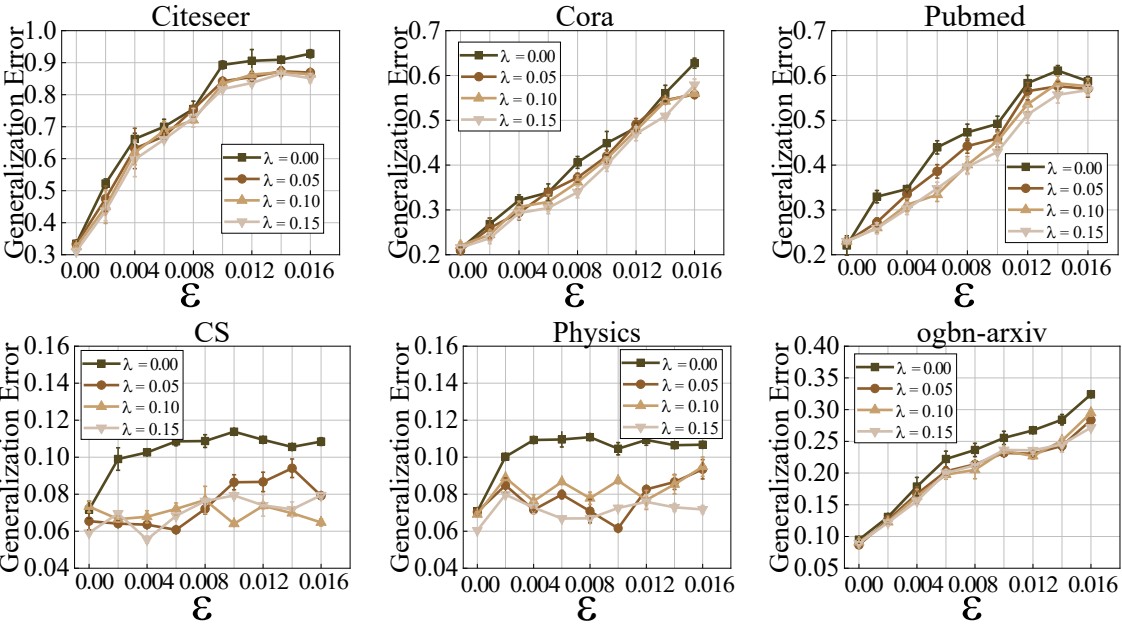

Figure 19: The empirical generalization error (mean value and standard deviation) of Residual GCN trained with $\ell_1$ regularization for different regularization parameters (i.e., $\lambda$). $\varepsilon$ denotes the maximum allowable perturbation.

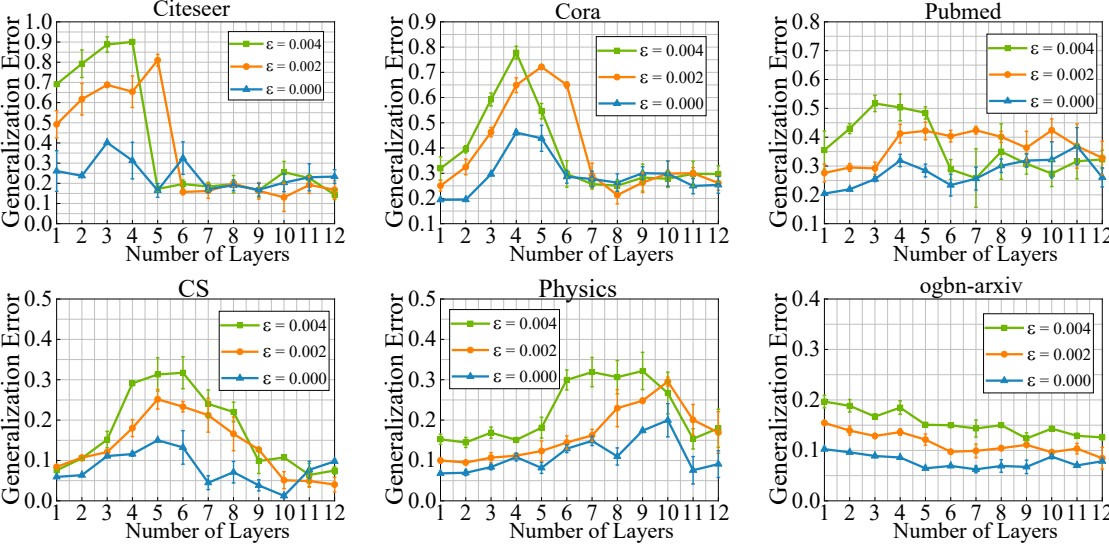

Figure 20: The empirical generalization error (mean value and standard deviation) with different depths for SGC. $\varepsilon$ denotes the maximum allowable perturbation.

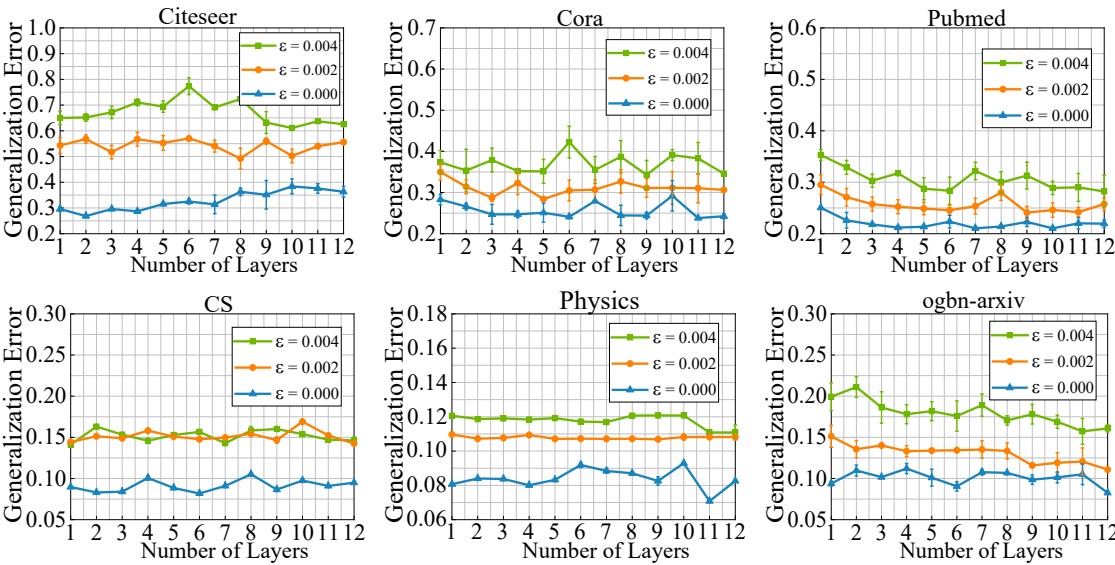

Figure 21: The empirical generalization error (mean value and standard deviation) with different depths for GCNII. $\varepsilon$ denotes the maximum allowable perturbation.

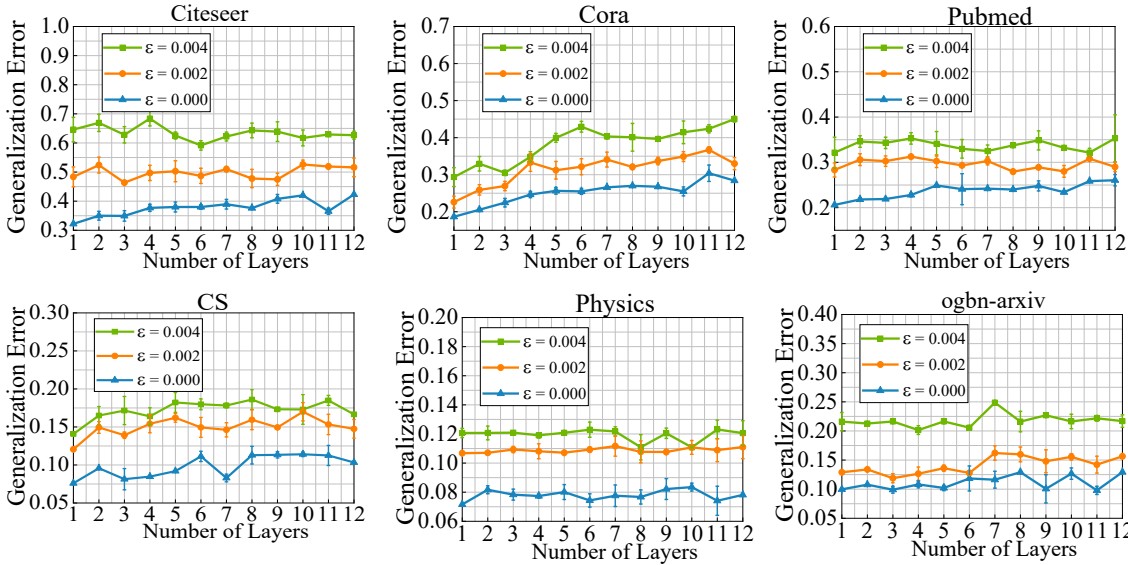

Figure 22: The empirical generalization error (mean value and standard deviation) with different depths for Residual GCN. $\varepsilon$ denotes the maximum allowable perturbation.

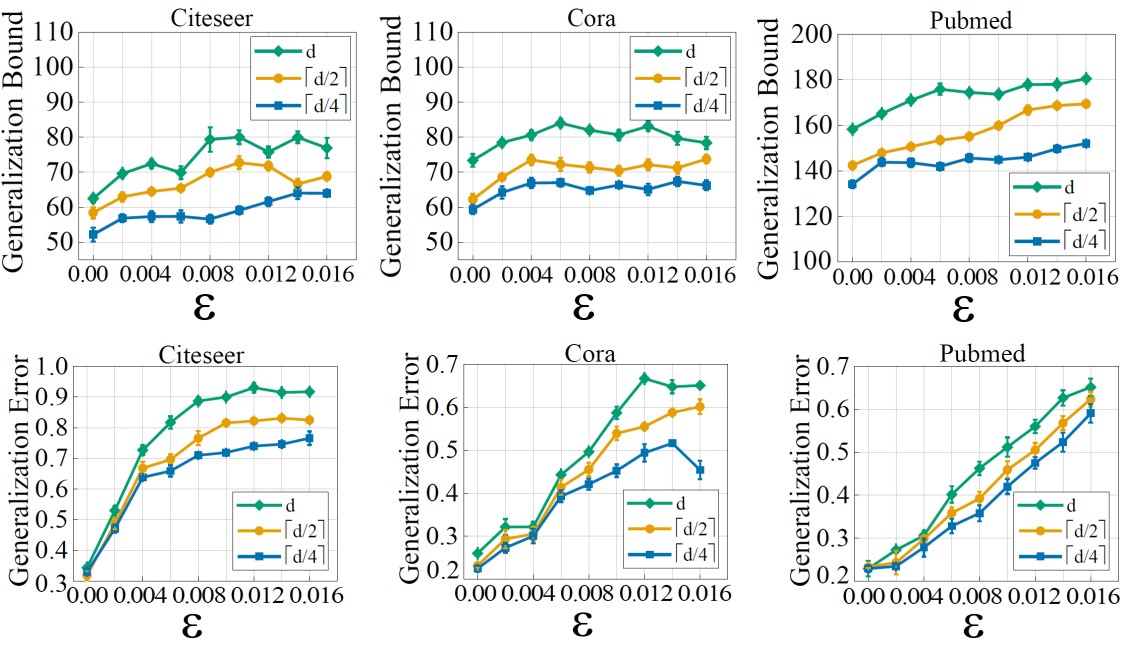

Figure 23: Theoretical generalization bounds and empirical generalization errors of the adopted datasets w.r.t. feature dimensions. $\varepsilon$ denotes the maximum allowable perturbation.

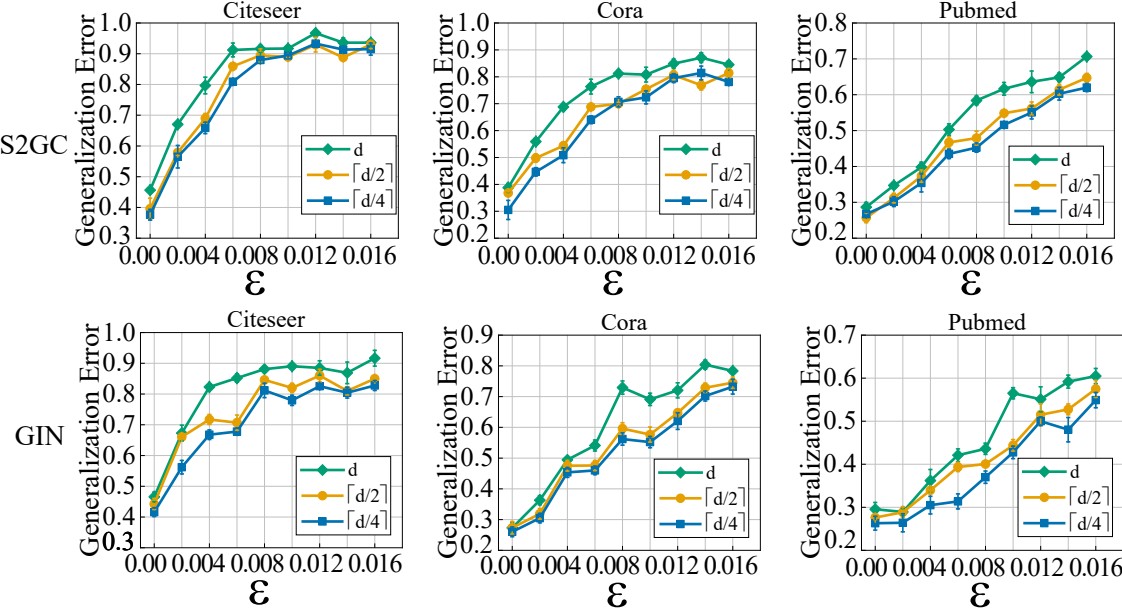

Figure 24: The empirical generalization error (mean value and standard deviation) with different dimensions for S2GC and GIN. $\varepsilon$ denotes the maximum allowable perturbation.

