# OpenReview forum: "Towards Generalization Bounds of GCNs for Adversarially Robust Node Classification"
_ICLR.cc/2025/Conference — ICLR 2025 Poster_

### Official Review · Reviewer_6SrP · 2024-11-03

**Soundness:** 3
**Presentation:** 3
**Contribution:** 3
**Rating:** 8
**Confidence:** 3

**Summary:**

The work tackles the problem of generalization of node classification when subject to adversarial attacks. Specifically, the study delves into the theoretical side, which is missing from the literature, by providing generalization bounds based on the Transductive Rademacher Complexity.

**Strengths:**

- The paper’s aim and perspective is well motivated and overall well written and presented.
- The presented theoretical study is very interesting and clearly useful.
    - Specifically, the connection between generalization and the dimension provided in Theorem 4.1 is very relevant. This can also serve as motivation to techniques that considered Weight regularisation as a defense approach while ensuring a good clean accuracy.
- The experimental setting seems to validate the theoretical insights.

**Weaknesses:**

- The analysis seems to rather focus on the GCN. While this is great, it would also be interesting to generalization to other more recently used models (Attention based GAT for instance).
- From my understanding of the problem setup, it seems that authors rather focus on adversarial perturbations targeting the node features. More clarification on this on the paper would be good.
- Additionally, possible insights on generalization to the adversarial structure perturbations could be useful.


- In the hypothesis class of GCNs (Eq. 1), the authors assume that the model’s weight norms are smaller or equal to $w$. Is there a specific motivation there? While I would agree that the existence of $w$ is clear (simply by taking the max over the norms), doesn’t that affect the tightness of the bound provided in Theorem 4.1 ?

- While the proposed experimental results confirms some of the concepts that were derived from the theoretical insights. It would also also be great to delve in the tightness of the proposed bounds.

**Questions:**

- Could you please provide insights on the possible extension to structural perturbations?
- It would be great to clarify some of the hypothesis provided in the paper and their relation to the tightness of the provided bound.
- How generalisable is your study to other models such as GIN, GAT or spectral models?
- On the experimental results, could you also results on other models such as to understand if empirically the results holds ?

---

> ### Author Response · Authors · 2024-11-16
> **Response to Reviewer 6SrP (Part 1)**
>
> We thank Reviewer 6SrP for the valuable comments and positive feedback! We will answer your questions as follows: (the modifications in the revised manuscript are marked in blue for easy reading)
>
> **A1 Extension to topology attacks.** Given the similar adversarial generation settings and adversarial loss definitions for topological attacks and node attacks, we suggest that the methodology (e.g., Lemma B.2) developed in this paper could be expanded upon the topology attack to address the outer maximization of the adversarial loss and tighten the generalization bounds.
>
> To be more specific, let the adversarial graph be generated from $\{\tilde{A}: \Vert A - \tilde{A}\Vert_\infty \leq \epsilon \}$, where $A$ and $\tilde{A}$ denote the original graph matrix and its adversarial counterpart. The adversarial loss w.r.t. adversarial graph is defined by
> $\max_{\Vert A - \tilde{A}\Vert_\infty \leq \epsilon} \ell(f(\tilde{A},X)_i,y_i) =  \hat{\ell} (\min _ {\Vert A - \tilde{A}\Vert _ \infty \leq \epsilon} y_if(\tilde{A},X)_i)$,
>
> where $\hat{\ell}:\mathbb{R}\rightarrow\mathbb{R} _ +$ is monotonically increasing. Analyzing analogously to the node attacks and applying the contraction technique in Lemma B.2, we obtain
>
> $Q \mathbb{E} _ {\sigma}\Big[\sup _ {f\in\mathcal{F}}\sum _ {i=1}^n\sigma _ i \inf _ {\Vert A - \tilde{A}\Vert_\infty \leq \epsilon} y_i f(\tilde{A},X) _ i \Big]  \leq Q \mathbb{E} _ {\sigma} \Big[\sup_{f\in \mathcal{F}} \sum _ {i=1}^n \sigma _ i f(\hat{A},X) _ i  \Big],$
>
> where $\hat{A} = \arg\inf _ {\tilde{A}:\Vert A - \tilde{A}\Vert _ \infty \leq \epsilon} y_i f(\tilde{A},X)_i$. If perturbing the graph topology at the last layer, then
>
> $Q \mathbb{E} _ {\sigma} \Big[\sup_{f\in \mathcal{F}} \sum_{i=1}^n \sigma_i f(\hat{A},X)_i  \Big]$
>
> $= Q \mathbb{E} _ {\sigma}\sup _ {\Vert W^{(L)}\Vert _ 2\leq \omega} \bigg[\sum _ {i=1}^n \sigma_i  \Big(\sum _ {j\in[n]} g(\hat{A}) _ {i,j} f^{(L-1)}(A,X) _ {j*} W^{(L)}\Big)\bigg]$
>
> $\leq Q\frac{1}{\lambda} \log \Bigg( 2 \Vert g(\hat{A})\Vert_\infty \omega \mathbb{E} _ {\sigma}\sup _ { j\in[n]} \exp \lambda  \bigg\Vert\sum _ {i=1}^n \sigma _ i   f^{(L-1)}(A,X) _ {j*} \bigg\Vert \Bigg)$
>
> $\cdots$
>
> $\leq Q\frac{1}{\lambda} \log \Bigg(2^L \Vert g(\hat{A})\Vert_\infty \Vert g(A)\Vert^{L-1}_\infty \omega^L \mathbb{E} _ {\sigma} \sup _ {j\in[n]} \exp \bigg(\lambda  \Big\Vert\sum _ {i=1}^n \sigma _ i  x _ {j}\Big\Vert _ {p*} \bigg)\Bigg)$
>
> Subsequent proof is similar to Eq (9) in Page 17. Notably, the upper bound of $\Vert g(\hat{A})\Vert^L_\infty$ for specific adversarial graph filters needs to be proved. Moreover, it remains to analyze the effect of perturbing the graph topology at the first layer, any intermediate layers, and all layers on generalization. We will leave this interesting study as future work. To refine this paper, we have added the discussion about topology attacks in Appendix G. Please refer to Page 30 in the revised paper.
>
> **A2. Connection between empirical results and theoretical bounds.** To further illustrate the tightness of the derived bounds to empirical observations, we quantitatively compute the theoretical generalization errors of the GCN model on the adopted dataset. As shown in Figure 23 (Page 41 of the revised paper), theoretical generalization errors decrease as the feature dimension decreases, which has the same overall trend as empirical observations. Hence, we suggest that these theoretical findings can provide effective guidance for improving the adversarial generalization of GCN models.
>
> **A3. Weight norm.** We clarify that for simplicity of notation, denote by $\omega$ the max bound over the norms, where $\omega=\max\{\omega_2,\omega_p \}$, $\Vert W^{(l)}\Vert_2 \leq \omega_2$, and $\Vert W^{(l)}\Vert_p \leq  \omega_p$. It is noteworthy that as discussed in Remark 4.2 (Page 5 of the revised paper), this weight norm constraint can help weaken the negative impact of adversarial perturbations, yielding tighter generalization bounds. Furthermore, the weight norm constraint is typically required to be a small constant, thereby rarely affecting the tightness of the bound. Inspired by the theoretical results, we adversarially train a robust model by leveraging the regularized adversarial training objective. Our experimental results further demonstrate the importance of an appropriate weight regularizer for achieving good generalization performance (Page 9 of the revised paper), and show that the norm of the weights could be less than or equal to 1. For clear reading, we have added the definition of $\omega$ below Eq. (2) in the revised paper.

---

> > ### Author Response · Authors · 2024-11-16
> > **Response to Reviewer 6SrP (Part 2)**
> >
> > **A4. Generalize to other models.** We highlight the scalability and universality of the generalization analysis developed in this paper from both theoretical and experimental perspectives:
> >
> > * Theoretically, we provide a unified methodology for deriving the generalization error bounds of GCNs in the adversarial setting. Our analysis is modular and applicable to a wide range of GCN models, including S2GC [1] and GIN [2]. Specifically, one could reduce the risk analysis of adversarial learning settings to that of standard learning settings by leveraging a tighter surrogate upper bound of the adversarial loss (e.g., Lemma 4.5). Then, using the contraction technique (Lemma A.2) and analyzing the TRC of the hypothesis can yield a tight generalization bound.
> >
> > * Empirically, we have conducted additional experiments on the S2GC [1] and GIN [2] models to validate our theoretical findings. We present the empirical errors with the dimension for the adopted datasets under adversarial attacks in Figure 24 (Page 41 of the revised paper). The experimental results show that the smaller the feature dimension, the smaller the generalization error, which is consistent with the theoretical findings.
> >
> > Reference:
> > 1. Zhu, Koniusz, et al. Simple spectral graph convolution. ICLR. 2021.
> >
> > 2. Xu, Keyulu, et al. How powerful are graph neural networks? ICML. 2018.

---

### Official Review · Reviewer_2X9z · 2024-11-04

**Soundness:** 4
**Presentation:** 3
**Contribution:** 3
**Rating:** 6
**Confidence:** 2

**Summary:**

The authors study adversarial generalization bounds for feature perturbations for GNNs. For this, the authors employ the Transductive Radamacher Complexity (TRC) to quantify the expressivity of hypothesis classes of GNNs and thereafter use this to derive generalization bounds. The authors instantiate their results for various GCN variants and empirically verify their results.

**Strengths:**

1. First to study adversarial generalization bounds for GNNs
1. A plethora of interesting theoretical insights that are verified empirically

**Weaknesses:**

1. This work solely studies feature perturbations and could elaborate on this earlier in the document
1. The authors could elaborate a bit on the specifics of how perturbations are introduced in a transductive setting. I.e., are perturbations of test data applied after training or before? A change after training is arguably inductive (e.g. see cited work of Gosch et al. 2024).

Minor:
- Equation 1 is hard to parse

**Questions:**

1. Why do the authors bound the l2 and lp norm of $W^{(l)}$ with the same constants? Aren't those norms in practice often of different magnitude?
1. How tight/useful are the derived bounds? While I appreciate the analysis of the relationship between, e.g., dimensions and generalization, it would also be nice to understand how well the theoretical generalization errors relate to the practically observed ones.

---

> ### Author Response · Authors · 2024-11-16
> **Response to Reviewer 2X9z**
>
> We thank Reviewer 2X9z for the insightful comments and positive feedback! We will answer your questions as follows: (the modifications in the revised manuscript are marked in blue for easy reading)
>
> **A1. Mention feature perturbations earlier.** Following the reviewer's suggestion, we have introduced the concept of the node perturbation attack at the beginning of the Introduction and elucidated the focus of this paper. Please refer to Page 1 of the revised paper.
>
> **A2. Experimental setup of perturbations.** We clarify that adversarial perturbations are added to test nodes after training, which avoids a biased evaluation due to memorization, as mentioned in [1]. We have added this description in Experimental Setup (Page 9 of the revised paper).
>
> **A3. Same norm constant.** We clarify that for simplicity of notation, denote by $\omega$ the max bound over the $\Vert \cdot\Vert_2, \Vert \cdot\Vert_p$-norms, where $\omega=\max\{\omega_2,\omega_p \}$,  $\Vert W^{(l)}\Vert_2 \leq \omega_2$, and $\Vert W^{(l)}\Vert_p \leq  \omega_p$. For clear reading, we have added the definition of $\omega$ below Eq. (2) in the revised paper.
>
> **A4. Correlation between theoretical and observed generalization errors.** To address your concerns, we compare both the theoretical and empirical generalization errors w.r.t. the feature dimension for the adopted datasets in Figure 23 (Page 41 of the revised paper), where the theoretical generalization error is computed based on the adopted dataset and the derived bounds.
> In Figure 23, the top three sub-figures present the theoretical generalization error, while the bottom three sub-figures present the experimental generalization error. Our experimental results show that both theoretical and empirical generalization errors decrease as the feature dimension decreases, with the same overall trend, which is consistent with our theoretical results. Therefore, we believe that our results can provide effective theoretical guidance for enhancing the adversarially robust generalization of GCNs.
>
> Reference:
> 1. Gosch et al. Adversarial training for graph neural networks: Pitfalls, solutions, and new directions. NIPS. 2024.

---

> ### Comment · Reviewer_2X9z · 2024-11-22
> **I thank the authors for their rebuttal**
>
> I thank the authors for clarifying most of my points and adjusting the paper accordingly!
>
> Regarding the theoretical bounds. While I do understand that it might be challenging, if not impossible, to derive simple bounds that are tight, is it correct that the theoretical bounds are about 2 orders of magnitude looser than the empirically observed ones?
>
> Nevertheless, I want to emphasize that if think that the paper is interesting regardless of how tight the bounds are.

---

> > ### Author Response · Authors · 2024-11-22
> > **Official Comment by Authors**
> >
> > Thank you once again for your thorough and insightful feedback. We are glad to address all your concerns in our response. Regarding your concerns, we clarify that the dimension size of the adopted datasets potentially causes the calculated theoretical generalization errors to be larger than the experimentally observed generalization errors. For this observation, in future work, we will reduce dimension dependency and develop tighter bounds. Your support and recognition are very encouraging.

---

### Official Review · Reviewer_7Djx · 2024-11-06

**Soundness:** 2
**Presentation:** 3
**Contribution:** 2
**Rating:** 6
**Confidence:** 3

**Summary:**

The paper introduce some theoritical bounds of the generalization of GCN and its variants. Through these bounds, the authors revealed the impact of mode parameters on the generalization behavior using the theory of Radamacher Complexity.

**Strengths:**

The theoritical backgroud sounds interesting. The paper introduce some theoritical bounds of the generalization of GCN and its variants. Through these bounds, the authors revealed the impact of mode parameters on the generalization behavior using the theory of Radamacher Complexity.

**Weaknesses:**

I think you can still developp other aspects in the papers. For example, the empirical evaluation is very limited. See the points below in details.

**Questions:**

1/ I cannot see how the generalization gap $Gen(f)$ in an indicator of the the generalization performance: 1-  $Gen(f)$ is the absolute difference of the adversarial error in the training set and the test set. 2- Since you re  considering the advesarial training, the adversarial training loss should converge to very low value compared to the adversarial test loss. 3- For me a good quantification of the generalization is $|\mathcal{L}_u - \tilde{\mathcal{L}}_u |$ to consider in order to quantify the impact of the adversarial training.

2/ In the Related Work section, you forgot to mention other type of Defence technique: GNNs thant enhace the robusntess by adapting the model architecture and seems to works well, For example, you can cite GCORN and Parseval networks [1,2].

3/  For me the empirical evaluation is too limited, ploting the generalization gap is not enough. You need to consider at least the test accuracy. And you can for example do an ablation study on different adversarial training strategies.

4/ This is just a suggestion: I suggest generalizing the theoritical findings to the graph classification tasks, and compare the adversarial training to other generalization enhacement frameworks such data augmentation which is also linked to Rademacher Complexity. Ex of data aug techniques for graphs : G-Mixup [3]

5/ The assumption you made about the loss function in Section 4.1 are not usually used in practice. For classification task, we usually use Cross entropy (which you also used in your experiments). Your assumption could work in node regression task perhaps.

6/ This is just a suggestion:For the ablation study on the graph filter, you can try others such as Random-walk Normalised Laplacian, Unnormalised Laplacian matrix.

4/ I recommend to evaluate the robustness against adversariat attacks using the Attack Success Rate (ASR), the percentage of attack attempts that produce successful adversarial examples, which implies that close input data should be predicted similarly [4]

[1] ABBAHADDOU, Y., ENNADIR, S., Lutzeyer, J. F., Vazirgiannis, M., & Boström, H. Bounding the Expected Robustness of Graph Neural Networks Subject to Node Feature Attacks. In The Twelfth International Conference on Learning Representations.

[2] Cisse, M., Bojanowski, P., Grave, E., Dauphin, Y., & Usunier, N. (2017, July). Parseval networks: Improving robustness to adversarial examples. In International conference on machine learning (pp. 854-863). PMLR.

[3] Han, Xiaotian, et al. "G-mixup: Graph data augmentation for graph classification." International Conference on Machine Learning. PMLR, 2022.

[4] Jing Wu, Mingyi Zhou, Ce Zhu, Yipeng Liu, Mehrtash Harandi, and Li Li. Performance evaluation of adversarial attacks: Discrepancies and solutions. arXiv preprint arXiv:2104.11103, 2021.

---

> ### Author Response · Authors · 2024-11-16
> **Response to Reviewer 7Djx**
>
> We thank Reviewer 7Djx for the insightful feedback! We will answer your questions as follows: (the modifications in the revised manuscript are marked in blue for easy reading）
>
> **A1. Evaluation criteria about generalization gap.** The difference between the population risk $\tilde{\mathcal{L}}_u(f)$ and the empirical risk $\tilde{\mathcal{L}}_m(f)$ is known as the generalization gap of $f$, denoted by $\mathrm{Gen}(f) = \tilde{\mathcal{L}}_u(f) - \tilde{\mathcal{L}}_m(f)$. A small generalization gap $\mathrm{Gen}(f)$ indicates that on average or high probability, the empirical performance of $f$ on the training dataset can be taken as a reliable measure of generalization to unknown samples. A large amount of work [1-6] aims at establishing the generalization error bound from the lens of statistical learning theory to guarantee the generalization of learning models.
>
> Notably, the generalization gap in theory usually cannot be calculated due to the unknown data distribution. Extensive work [1-4] thus considers an empirical proxy of the generalization gap:
> $$\vert \textrm{Adversarial Training Accuracy} - \textrm{Adversarial Test Accuracy}\vert,$$
> and then validate the theoretical findings by evaluating the empirical generalization errors with relevant factors. Accordingly, we utilize the empirical generalization error rather than ASR or other metrics to evaluate the adversarially robust generalization of the models, which is reasonable and realistic for verifying our theoretical findings.
>
> **A2. Related work.** Thanks for the recommended literature. We have cited the work of GCORN and Parseval networks in Page 2 of the revised paper.
>
> **A3. Assumption.** We clarify that the cross-entropy loss, defined by $L_{CE} = -\sum_{i=1}^n y_i\log(\hat{y}_i)$, where $y_i$ is the true label and $\hat{y}_i\in[0,1]$ is the probability of the predicted label, is indeed monotonically non-increasing loss function, which satisfies the monotonicity assumption considered in this paper (Page 4). It is noteworthy that this assumption is mild and realistic, encompassing commonly used losses such as the hinge loss, logistic loss, and exponential loss [1, 4], and has been widely used in adversarial learning literature [1, 4-6] to derive the non-trivial bounds. For clear reading, we have added a more detailed discussion about the assumption in Page 4 of the revised paper.
>
> **A4. Experiments about graph filters.** In Figure 3 (Page 32), we indeed compare the empirical generalization error with different graph filters, including the unnormalized Laplacian graph, symmetric normalized graph, and the random-walk graph. The experimental results show that the unnormalized graph has larger empirical generalization errors than the normalized graphs. Hence, we argue that normalizing the graph matrix can facilitate the adversarial generalization of GCNs. This also validates our theoretical findings.
>
> **A5. Experiments about adversarial training strategies.** In Appendix I (Page 31 of the revised paper), we also have compared the empirical generalization errors with feature dimensions for different adversarial training strategies, where the adversarial examples are generated by FGSM and BIM attack algorithms. The experimental results in Figures 8-13 (Pages 34-36) show that the low-dimensional feature mapping helps enhance the generalization performance of adversarial training models. This further verifies the effectiveness of our theoretical findings.
>
> Reference:
>
> 1. Yin, Ramchandran, Bartlett. Rademacher complexity for adversarially robust generalization. ICML. 2019.
>
> 2. Verma and Zhang. Stability and generalization of graph convolutional neural networks. KDD. 2019.
>
> 3. Deng et al. Graph convolution network based recommender systems: Learning guarantee and item mixture powered strategy. NIPS. 2022.
>
> 4. Xiao et al. Stability analysis and generalization bounds of adversarial training. NIPS. 2022.
>
> 5. Khim and Loh. Adversarial risk bounds via function transformation. 2018.
>
> 6. Awasthi et al. Adversarial learning guarantees for linear hypotheses and neural networks. ICML. 2020.

---

> > ### Comment · Reviewer_7Djx · 2024-11-22
> >
> > I will raise my score. I recommend however to do more experiments to make the work more solid.

---

> > > ### Author Response · Authors · 2024-11-22
> > > **Official Comment by Authors**
> > >
> > > Thank you for acknowledging our clarification and for raising your score. We greatly appreciate your constructive feedback, which has been invaluable in strengthening our paper. Your support and recognition are very encouraging. In further work, we will design theory-driven algorithms and conduct rich experiments to improve our work.

---

### Official Review · Reviewer_HnSA · 2024-11-10

**Soundness:** 3
**Presentation:** 3
**Contribution:** 2
**Rating:** 6
**Confidence:** 3

**Summary:**

This paper presents1 a theoretical study on the generalization capabilities of Graph Convolutional Networks (GCNs) under adversarial conditions. The research aims to establish high-probability generalization bounds for GCNs in node classification tasks when exposed to adversarial perturbations. Using Transductive Rademacher Complexity (TRC) and a novel contraction technique, the authors examine the relationship between generalization error and adversarial robustness, accounting for key factors such as feature dimension, network depth, and graph normalization. Their theoretical contributions are validated with empirical experiments on benchmark datasets, highlighting the efficacy of different GCN architectures, including SGC, Residual GCN, and GCNII, under adversarial training.

**Strengths:**

1. A comprehensive theoretical framework to analyze adversarial robustness in GCNs is provided.
2. The proposed method can be applied to a wide range of models and loss functions, including some deeper GNNs.
3. The experiment results demonstrate the effectiveness of the proposed method in reducing generalization error.

**Weaknesses:**

1. Only an $l_{\infty}$-PGD attack is considered in the evaluation. Including additional state-of-the-art graph adversarial attacks would provide a more comprehensive evaluation of the proposed method’s robustness. For instance, it would be beneficial to evaluate against attacks such as [1, 2] to give clearer insights into performance under diverse adversarial scenarios.
2. The proposed method is currently limited to feature perturbation attacks, while structural attacks are often more powerful and prevalent. Could the authors discuss the possibility and challenges of extending the method to structural perturbation attacks? Specifically, it would be helpful to address: 1) the key differences between feature and structural attacks that would need to be considered, 2) which parts of the current analysis might be adaptable to structural attacks, and 3) any new theoretical tools required for handling these types of attacks.

[1] Adversarial Attacks on Neural Networks for Graph Data.

[2] Adversarial attacks on node embeddings via graph poisoning.

**Questions:**

1. Comparisons to non-adversarial generalization bounds could benefit from further discussion on the trade-offs between robustness and model capacity in adversarial versus standard training settings. Can you add a specific subsection or paragraph that explicitly compares the derived bounds to non-adversarial bounds? It would clarify how model capacity and robustness trade off differently in these two contexts, strengthening the implications of the theoretical results.

---

> ### Author Response · Authors · 2024-11-16
> **Response to Reviewer HnSA**
>
> We thank Reviewer HnSA for the valuable comments and positive feedback! We will answer your questions as follows: (the modifications in the revised manuscript are marked in blue for easy reading)
>
> **A1. Diverse attacks.** We have conducted additional experiments to evaluate the generalization performance under FGSM and BIM node attack methods. The empirical generalization errors with dimensions of adopted datasets under attacks are plotted in Figures 8-13 (Pages 34-36 of the revised paper). The experimental results show that the low-dimensional feature mapping can help improve the generalization ability of the learning models against adversarial attacks, which is consistent with our theoretical findings.
>
> **A2. Discussion about topology attacks.** In the revised paper, we have added more discussion about topology attacks in Appendix G (Page 30 of the revised paper). We emphasize that given the similar adversarial generation settings and adversarial loss definitions for topological attacks and node attacks, the methodology (e.g., Lemma B.2) developed in this paper could be expanded upon the topology attack to address the outer maximization of the adversarial loss and tighten the generalization bounds. However, further investigation is needed regarding the norm bounds of adversarial graph filters and the impact of perturbing the graph topology at different network layers (including the first, last, intermediate, and all layers).
>
> To be more specific, let the adversarial graph be generated from $\{\tilde{A}: \Vert A - \tilde{A}\Vert_\infty \leq \epsilon \}$, where $A$ and $\tilde{A}$ denote the original graph matrix and its adversarial counterpart. The adversarial loss w.r.t. adversarial graph is defined by
>
> $\max_{\Vert A - \tilde{A}\Vert_\infty \leq \epsilon} \ell(f(\tilde{A},X)_i,y_i) =  \hat{\ell} (\min _ {\Vert A - \tilde{A}\Vert _ \infty \leq \epsilon} y_if(\tilde{A},X)_i)$,
>
> where $\hat{\ell}:\mathbb{R}\rightarrow\mathbb{R} _ +$ is monotonically non-increasing. Analyzing analogously to the node attacks and applying the contraction technique in Lemma B.2, we obtain
>
> $Q \mathbb{E} _ {\sigma}\Big[\sup _ {f\in\mathcal{F}}\sum _ {i=1}^n\sigma _ i \inf _ {\Vert A - \tilde{A}\Vert_\infty \leq \epsilon} y_i f(\tilde{A},X) _ i \Big]  \leq Q \mathbb{E} _ {\sigma} \Big[\sup_{f\in \mathcal{F}} \sum _ {i=1}^n \sigma _ i f(\hat{A},X) _ i  \Big],$
>
> where $\hat{A} = \arg\inf _ {\tilde{A}:\Vert A - \tilde{A}\Vert _ \infty \leq \epsilon} y_i f(\tilde{A},X)_i$. If perturbing the graph topology at the last layer, then
>
> $Q \mathbb{E} _ {\sigma} \Big[\sup_{f\in \mathcal{F}} \sum_{i=1}^n \sigma_i f(\hat{A},X)_i  \Big]$
>
> $= Q \mathbb{E} _ {\sigma}\sup _ {\Vert W^{(L)}\Vert _ 2\leq \omega} \bigg[\sum _ {i=1}^n \sigma_i  \Big(\sum _ {j\in[n]} g(\hat{A}) _ {i,j} f^{(L-1)}(A,X) _ {j*} W^{(L)}\Big)\bigg]$
>
> $\leq Q\frac{1}{\lambda} \log \Bigg( 2 \Vert g(\hat{A})\Vert_\infty \omega \mathbb{E} _ {\sigma}\sup _ { j\in[n]} \exp \lambda  \bigg\Vert\sum _ {i=1}^n \sigma _ i   f^{(L-1)}(A,X) _ {j*} \bigg\Vert \Bigg)$
>
> $\cdots$
>
> $\leq Q\frac{1}{\lambda} \log \Bigg(2^L \Vert g(\hat{A})\Vert_\infty \Vert g(A)\Vert^{L-1}_\infty \omega^L \mathbb{E} _ {\sigma} \sup _ {j\in[n]} \exp \bigg(\lambda  \Big\Vert\sum _ {i=1}^n \sigma _ i  x _ {j}\Big\Vert _ {p*} \bigg)\Bigg)$
>
> Subsequent proof is similar to Eq (9) in Page 17. Notably, the upper bound of $\Vert g(\hat{A})\Vert^L_\infty$ for specific adversarial graph filters needs to be proved. In addition, it remains to analyze the effect of perturbing the graph topology at the first layer, any intermediate layers, and all layers on generalization.
>
> **A3. Comparison with non-adversarial bounds.** Following the reviewer's suggestion, we have provided a detailed comparison in Remark 4.4 (Page 5 of the revised paper). "Taking $\varepsilon=0$ and applying Lemma 4.5 yield the upper bound of the generalization gap in the non-adversarial setting:
>
> $\mathcal{O}\Big(\max \big( \frac{1}{\sqrt{m}},\frac{1}{\sqrt{n-m}} \big) \times \big(\sqrt{2\log(2)L}+1 \big) \Vert g(A)\Vert^L_\infty \omega^L B_{p^*}\Vert X\Vert_{2,p^*} \Big),$
>
> which has a comparable convergence rate of $\mathcal{O}(\max(\frac{1}{\sqrt{m}},\frac{1}{\sqrt{n-m}}))$ to the existing TRC-based bound [1-3]. Notably, our bound improves the existing exponential dependency of the number of layers to a logarithmic term $\mathcal{O}(\sqrt{2\log(2)L})$, facilitating the tighter bound than [1, 2], which benefit from the usage of the contraction technique. "
>
> Reference:
>
> 1. Esser et al. Learning theory can (sometimes) explain generalisation in graph neural networks. NIPS. 2021.
>
> 2. Tang and Liu. Towards understanding generalization of graph neural networks. ICML. 2023.
>
> 3. Deng et al. Graph convolution network based recommender systems: Learning guarantee and item mixture powered strategy. NIPS. 2022.

---

> > ### Comment · Reviewer_HnSA · 2024-11-21
> >
> > Thanks for your response. I will keep my score.

---

> > > ### Author Response · Authors · 2024-11-22
> > > **Official Comment by Authors**
> > >
> > > We greatly appreciate your positive feedback and support for our work. Your support and recognition are very encouraging.

---

### Author Response · Authors · 2024-11-21
**Would you mind confirming if you have further questions? Thanks!**

Dear Reviewer HnSA, Reviewer 7Djx, Reviewer 2X9z, Reviewer 6SrP

We appreciate reviewers for their efforts and time in providing constructive comments. We have carefully considered the reviewers' concerns/advice and provided as much refinement and experiments as possible to address the issues regarding the submission. Would you mind checking our response and confirming whether we have addressed your concerns? If there are any further questions or suggestions, we will try our best to solve them to improve our work.

Sincerely,

The authors of Paper 5326

---

### Comment · Area_Chair_bm7b · 2024-11-28

I would like to encourage the reviewers to engage with the author's replies if they have not already done so. At the very least, please
acknowledge that you have read the rebuttal.

---

### Meta-Review · Area_Chair_bm7b · 2024-12-19

**Metareview:**

The paper propose generalisation bounds for GCN under adversarial attacks based on the transductive rademacher complexity. They study the effect of feature dimension, network depth, and graph normalization. The work tackles only feature perturbations. Some reviewers mentioned that this fact should be better highlighted and the authors concur. While some reviewers found the experimental evaluation to be limited other reviewers stated that "The experiment results demonstrate the effectiveness of the proposed method in reducing generalization error.". Overall, given the theoretical focus of the work I deem the experiments to be sufficient. The insight can provide guidance for improving GNNs' adversarial generalization.

**Additional Comments On Reviewer Discussion:**

The reviewers all agree that the theory is interesting and useful. Two reviewers highlighted the feature-only aspect of the work, and in the response the authors provided a partial extension to topology attacks (albeit under somewhat unrealistic threat model). The authors conduct additional experiments under FGSM and BIM node attack methods which while not SOTA do provide a clearer picture. Overall, the authors could address the majority of concerns and the reviewers positively evaluated the paper.

---

### Decision · Program_Chairs · 2025-01-22

Accept (Poster)